# Functional Bilevel Optimization for Machine Learning

**Ieva Petrulionyte, Julien Mairal, Michael Arbel**
Univ. Grenoble Alpes, Inria, CNRS, Grenoble INP, LJK, 38000 Grenoble, France
`firstname.lastname@inria.fr`

## Abstract

In this paper, we introduce a new functional point of view on bilevel optimization problems for machine learning, where the inner objective is minimized over a function space. These types of problems are most often solved by using methods developed in the parametric setting, where the inner objective is strongly convex with respect to the parameters of the prediction function. The functional point of view does not rely on this assumption and notably allows using over-parameterized neural networks as the inner prediction function. We propose scalable and efficient algorithms for the functional bilevel optimization problem and illustrate the benefits of our approach on instrumental regression and reinforcement learning tasks.

## 1 Introduction

Bilevel optimization methods solve problems with hierarchical structures, optimizing two interdependent objectives: an *inner-level* objective and an *outer-level* one. Initially used in machine learning for model selection [Bennett et al., 2006] and sparse feature learning [Mairal et al., 2012], these methods gained popularity as efficient alternatives to grid search for hyper-parameter tuning [Feurer and Hutter, 2019, Lorraine et al., 2019, Franceschi et al., 2017]. Applications of bilevel optimization include meta-learning [Bertinetto et al., 2019], auxiliary task learning [Navon et al., 2021], reinforcement learning [Hong et al., 2023, Liu et al., 2021a, Nikishin et al., 2022], inverse problems [Holler et al., 2018] and invariant risk minimization [Arjovsky et al., 2019, Ahuja et al., 2020].

Bilevel problems are challenging to solve, even in the *well-defined bilevel* setting with a unique inner-level solution. This difficulty stems from approximating both the inner-level solution and its sensitivity to the *outer-level* variable during gradient-based optimization. Methods like Iterative Differentiation (ITD, Baydin et al., 2017) and Approximate Implicit Differentiation (AID, Ghadimi and Wang, 2018) were designed to address these challenges in the well-defined setting, resulting in scalable algorithms with strong convergence guarantees [Domke, 2012, Gould et al., 2016, Ablin et al., 2020, Arbel and Mairal, 2022a, Blondel et al., 2022, Liao et al., 2018, Liu et al., 2022b, Shaban et al., 2019]. These guarantees usually require the *inner-level* objective to be strongly convex. However, when the inner-level variables are neural network parameters, the lower-level problem becomes non-convex and may have multiple solutions due to over-parameterization. While non-convexity is considered "benign" in this setting [Allen-Zhu et al., 2019, Liu et al., 2022a], multiplicity of inner-level solutions makes their dependence on the outer-level variable ambiguous [Liu et al., 2021b], posing a major challenge in bilevel optimization for modern machine learning applications.

We identify a common *functional structure* in bilevel machine learning problems to address the ambiguity challenge that arises with flexible models like neural networks. Specifically, we consider a *prediction function* $h$ optimized by the inner-level problem over a Hilbert space $\mathcal{H}$. This space consists of functions defined over an input space $\mathcal{X}$ and taking values in a finite dimensional vector space $\mathcal{V}$. The optimal *prediction function* is then evaluated in the outer-level to optimize an outer-level parameter $\omega$ in a finite dimensional space $\Omega = \mathbb{R}^d$, resulting in a *functional* bilevel problem:

$$\min_{\omega \in \Omega} \mathcal{F}(\omega) := L_{out}(\omega, h_\omega^\star) \quad \text{s.t.} \quad h_\omega^\star = \arg\min_{h \in \mathcal{H}} L_{in}(\omega, h). \tag{FBO}$$

38th Conference on Neural Information Processing Systems (NeurIPS 2024).

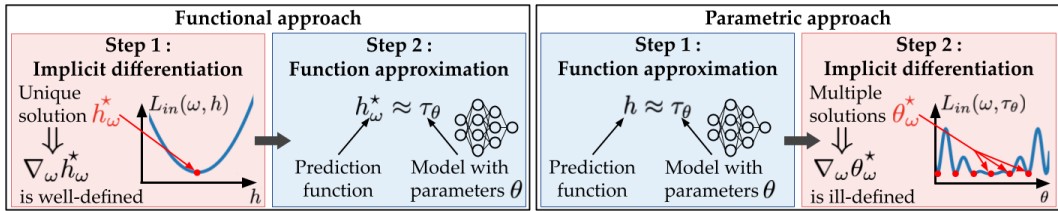

Figure 1: Parametric vs functional approaches for solving FBO by implicit differentiation.

In contrast to classical bilevel formulations involving neural networks, where the inner objective is non-convex with respect to the network parameters, the inner objective in (FBO) defines an optimization problem over a prediction function $h$ in a functional vector space $\mathcal{H}$.

A crucial consequence of adopting this new viewpoint is that it renders the strong convexity of the inner objective with respect to $h$ a mild assumption, which ensures the uniqueness of the solution $h_\omega^\star$ for any outer parameter value $\omega$. Strong convexity with respect to the *prediction function* is indeed much weaker than strong convexity with respect to model parameters and often holds in practice. For instance, a supervised prediction task with pairs of features/labels $(x, y)$ drawn from some training data distribution is formulated as a regularized empirical minimization problem:

$$\min_{h \in \mathcal{H}} L_{in}(\omega, h) := \mathbb{E}_{x,y} \left[ \|y - h(x)\|_2^2 \right] + \frac{\omega}{2} \|h\|_{\mathcal{H}}^2, \tag{1}$$

where $\mathcal{H}$ is the $L_2$ space of square integrable functions w.r.t. the distribution of $x$, and $\omega > 0$ controls the amount of regularization. The inner objective is strongly convex in $h$, even though the optimal prediction function $h_\omega^\star$ can be highly nonlinear in $x$. The function $h_\omega^\star$ may then be approximated, *e.g.*, by an overparameterized deep neural network, used here as a function approximation tool.

Although appealing, the (FBO) formulation necessitates the development of corresponding theory and algorithms, which is the aim of our paper. To the best of our knowledge, this is the first work to propose a functional bilevel point of view that can leverage deep networks for function approximation. The closest works are either restricted to kernel methods [Rosset, 2008, Kunapuli et al., 2008] and thus cannot be used for deep learning models, or propose abstract algorithms that can only be implemented for finite Hilbert spaces [Suonperä and Valkonen, 2024].

We introduce in Section 2 a theoretical framework for functional implicit differentiation in an abstract Hilbert space $\mathcal{H}$ that allows computing the *total gradient* $\nabla \mathcal{F}(\omega)$ using a functional version of the *implicit function theorem* [Ioffe and Tihomirov, 1979] and the *adjoint sensitivity method* [Pontryagin, 2018]. This involves solving a well-conditioned functional linear system, equivalently formulated as a regression problem in $\mathcal{H}$, to find an adjoint function $a_\omega^\star$ used for computing the *total gradient* $\nabla \mathcal{F}(\omega)$. We then specialize this framework to the common scenario where $\mathcal{H}$ is an $L_2$ space and objectives are expectations of point-wise losses. This setting covers many machine learning problems (see Sections 4.1, 4.2, and Appendix A). In Section 3, we propose an efficient algorithm where the prediction and adjoint functions can be approximated using parametric models, like neural networks, learned with standard stochastic optimization tools. We further study its convergence using analysis for biased stochastic gradient descent [Demidovich et al., 2024].

The proposed method, called *functional implicit differentiation* (*FuncID*), adopts a novel "differentiate implicitly, then parameterize" approach (left Figure 1): functional strong convexity is first exploited to derive an **unambiguous** implicit gradient in function space using a well-defined adjoint function. Then, both the lower-level solution and adjoint function are approximated using neural networks. This contrasts with traditional AID/ITD approaches (right Figure 1), which parameterize the inner-level solution as a neural network, leading to a non-convex 'parametric' bilevel problem in the network's parameters. An **ambiguous** implicit gradient is then computed by approximately solving an unstable or ill-posed linear system [Arbel and Mairal, 2022b]. Consequently, *FuncID* addresses the ambiguity challenge by exploiting the functional perspective and results in a stable algorithm with reduced time and memory costs. In Section 4, we demonstrate the benefits of our approach in instrumental regression and reinforcement learning tasks, which admit a natural functional bilevel structure.

**Related work on bilevel optimization with non-convex inner objectives.** In principle, considering amended versions of the bilevel problem can resolve the ambiguity arising from non-convex inner

objectives. This is the case of optimistic/pessimistic versions of the problem, often considered in the literature on mathematical optimization, where the outer-level objective is optimized over both outer and inner variables, under the optimality constraint of the inner-level variable [Dempe et al., 2007, Ye and Ye, 1997, Ye and Zhu, 1995, Ye et al., 1997]. While tractable methods were recently proposed to solve them [Liu et al., 2021a,b, 2023, Kwon et al., 2024, Shen and Chen, 2023], it is unclear how well the resulting solutions behave on unseen data in the context of machine learning. For instance, when using over-parameterized models for the inner-level problem, their parameters must be further optimized for the outer-level objective, possibly resulting in over-fitting [Vicol et al., 2022]. More recently, Arbel and Mairal [2022b] proposed a game formulation involving a *selection map* to deal with multiple inner-level solutions. Such a formulation justifies the use of ITD/AID outside the well-defined bilevel setting, by viewing those methods as approximating the Jacobian of the selection map. However, the justification only hold under rather strong geometric assumptions. Additional related work is discussed in Appendix B on bilevel optimization with strongly-convex inner objectives, the adjoint sensitivity method that is often used in the context of ordinary differential equations, and amortization techniques [Amos et al., 2023] that have been also exploited for approximately solving bilevel optimization problems [MacKay et al., 2019, Bae and Grosse, 2020].

## 2 A Theoretical Framework for Functional Bilevel Optimization

The functional bilevel problem (FBO) involves an optimal prediction function $h_\omega^\star$ for each value of the outer-level parameter $\omega$. Solving (FBO) by using a first-order method then requires characterizing the implicit dependence of $h_\omega^\star$ on the outer-level parameter $\omega$ to evaluate the total gradient $\nabla \mathcal{F}(\omega)$ in $\mathbb{R}^d$. Indeed, assuming that $h_\omega^\star$ and $L_{out}$ are Fréchet differentiable (this assumption will be discussed later), the gradient $\nabla \mathcal{F}(\omega)$ may be obtained by an application of the chain rule:

$$\nabla \mathcal{F}(\omega) = g_\omega + \partial_\omega h_\omega^\star d_\omega, \quad \text{with} \quad g_\omega := \partial_\omega L_{out}(\omega, h_\omega^\star) \quad \text{and} \quad d_\omega := \partial_h L_{out}(\omega, h_\omega^\star). \quad (2)$$

The Fréchet derivative $\partial_\omega h_\omega^\star : \mathcal{H} \to \mathbb{R}^d$ is a linear operator acting on functions in $\mathcal{H}$ and measures the sensitivity of the optimal solution on the outer variable. We will refer to this quantity as the "Jacobian" in the rest of the paper. While the expression of the gradient in Equation (2) might seem intractable in general, we will see in Section 3 a class of practical algorithms to estimate it.

### 2.1 Functional implicit differentiation

Our starting point is to characterize the dependence of $h_\omega^\star$ on the outer variable. To this end, we rely on the following implicit differentiation theorem (proven in Appendix C) which can be seen as a functional version of the one used in AID [Domke, 2012, Pedregosa, 2016], albeit, under a much weaker *strong convexity assumption* that holds in most practical cases of interest.

**Theorem 2.1 (Functional implicit differentiation).** *Consider problem (FBO) and assume that:*

- *For any $\omega \in \Omega$, there exists $\mu > 0$ such that $h \mapsto L_{in}(\omega', h)$ is $\mu$-strongly convex for any $\omega'$ near $\omega$.*

- *$h \mapsto L_{in}(\omega, h)$ has finite values and is Fréchet differentiable on $\mathcal{H}$ for all $\omega \in \Omega$.*

- *$\partial_h L_{in}$ is Hadamard differentiable on $\Omega \times \mathcal{H}$ (in the sense of Definition C.1 in Appendix C).*

*Then, $\omega \mapsto h_\omega^\star$ is uniquely defined and is Fréchet differentiable with a Jacobian $\partial_\omega h_\omega^\star$ given by:*

$$B_\omega + \partial_\omega h_\omega^\star C_\omega = 0, \qquad \text{with} \quad B_\omega := \partial_{\omega,h} L_{in}(\omega, h_\omega^\star), \quad \text{and} \quad C_\omega := \partial_h^2 L_{in}(\omega, h_\omega^\star). \quad (3)$$

The strong convexity assumption on the inner-level objective ensures the existence and uniqueness of the solution $h_\omega^\star$, while differentiability assumptions on $L_{in}$ and $\partial_h L_{in}$ ensure Fréchet differentiability of the map $\omega \mapsto h_\omega^\star$. Though the implicit function theorem for Banach spaces [Ioffe and Tihomirov, 1979] could yield similar conclusions, it demands the stronger assumption that $\partial_h L_{in}$ is continuously Fréchet differentiable, which is quite restrictive in our setting of interest: for instance, when $\mathcal{H}$ is an $L_2$-space and $L_{in}$ is an integral functional of the form $L_{in}(\omega, h) = \int \ell_{in}(w, h(x)) \, dx$, with $\ell_{in}$ defined on $\Omega \times \mathcal{V}$ and satisfying mild smoothness and growth assumptions on $\ell_{in}$, then $h \mapsto \partial_h L_{in}(\omega, h)$ cannot be Fréchet differentiable with uniformly continuous differential on bounded sets, unless $v \mapsto \ell_{in}(\omega, v)$ is a polynomial of degree at most 2 (see [Nemirovski and Semenov, 1973, Corollary 2, p 276] and discussions in [Noll, 1993, Goodman, 1971]). Instead, Theorem 2.1 employs the weaker notion of Hadamard differentiability for $\partial_h L_{in}$, widely used in statistics, particularly for

deriving the *delta-method* [van der Vaart and Wellner, 1996, Chapter 3.9]. Consequently, Theorem 2.1 allows us to cover a broader class of functional bilevel problems, as we see in Section 2.2.

Similarly to AID, only a Jacobian-vector product is needed when computing the total gradient $\nabla \mathcal{F}(\omega)$. The result in Proposition 2.2 below, relies on the *adjoint sensitivity method* [Pontryagin, 2018] to provide a more convenient expression for $\nabla \mathcal{F}(\omega)$ and is proven in Appendix C.2.

**Proposition 2.2 (Functional adjoint sensitivity).** *Under the same assumption on $L_{in}$ as in Theorem 2.1 and further assuming that $L_{out}$ is jointly differentiable in $\omega$ and $h$, the total objective $\mathcal{F}$ is differentiable with $\nabla \mathcal{F}(\omega)$ given by:*

$$\nabla \mathcal{F}(\omega) = g_\omega + B_\omega a_\omega^\star, \tag{4}$$

*where the adjoint function $a_\omega^\star := -C_\omega^{-1} d_\omega$ is an element of $\mathcal{H}$ that minimizes the quadratic objective:*

$$a_\omega^\star = \arg\min_{a \in \mathcal{H}} L_{adj}(\omega, a) := \tfrac{1}{2} \langle a, C_\omega a \rangle_{\mathcal{H}} + \langle a, d_\omega \rangle_{\mathcal{H}}. \tag{5}$$

Equation (4) indicates that computing the total gradient requires optimizing the quadratic objective (5) to find the adjoint function $a_\omega^\star$. The strong convexity of the adjoint objective $L_{adj}$ ensures the existence of a unique minimizer, and stems from the positive definiteness of its Hessian operator $C_\omega$ due to the inner-objective's strong convexity. Both adjoint and inner-level problems occur in the same function space $\mathcal{H}$ and are equivalent in terms of conditioning, as the adjoint Hessian operator equals the inner-level Hessian at optimum. The *strong convexity in $\mathcal{H}$* of the adjoint objective guarantees well-defined solutions and holds in many practical cases, as opposed to classical parametric bilevel formulations which require the more restrictive *strong convexity condition in the model's parameters*, and without which instabilities may arise due to ill-conditioned linear systems (see Appendix F.1).

## 2.2 Functional bilevel optimization in $L_2$ spaces

Specializing the abstract results from Section 2.1 to a common scenario in machine learning, we consider both inner and outer level objectives of (FBO) as expectations of point-wise functions over observed data. Specifically, we have two data distributions $\mathbb{P}$ and $\mathbb{Q}$ defined over a product space $\mathcal{X} \times \mathcal{Y} \subset \mathbb{R}^{d_x} \times \mathbb{R}^{d_y}$, and denote by $\mathcal{H}$ the Hilbert space of functions $h : \mathcal{X} \to \mathcal{V}$, where $\mathcal{V} = \mathbb{R}^{d_v}$. Given an outer parameter space $\Omega$, we address the following functional bilevel problem:

$$\begin{aligned} \min_{\omega \in \Omega} \ & L_{out}\left(\omega, h_\omega^\star\right) := \mathbb{E}_{\mathbb{Q}}\left[\ell_{out}\left(\omega, h_\omega^\star(x), x, y\right)\right] \\ & \text{s.t. } h_\omega^\star = \arg\min_{h \in \mathcal{H}} \ L_{in}\left(\omega, h\right) := \mathbb{E}_{\mathbb{P}}\left[\ell_{in}\left(\omega, h(x), x, y\right)\right], \end{aligned} \tag{6}$$

where $\ell_{out}, \ell_{in}$ are point-wise loss functions defined on $\Omega \times \mathcal{V} \times \mathcal{X} \times \mathcal{Y}$. This setting encompasses various deep learning problems discussed in Sections 4.1 and 4.2, and in Appendix A, representing a specific instance of FBO. The Hilbert space $\mathcal{H}$ of square-integrable functions not only models a broad range of prediction functions but also facilitates obtaining concrete expressions for the total gradient $\nabla \mathcal{F}(\omega)$, enabling the derivation of practical algorithms in Section 3.

The following proposition, proved in Appendix D, makes mild technical assumptions on probability distributions $\mathbb{P}, \mathbb{Q}$ and the point-wise losses $\ell_{in}, \ell_{out}$. It gives an expression for the total gradient in the form of expectations under $\mathbb{P}$ and $\mathbb{Q}$.

**Proposition 2.3 (Functional Adjoint sensitivity in $L_2$ spaces.).** *Under the technical Assumptions (A) to (L) stated in Appendix D.1, the conditions on $L_{in}$ and $L_{out}$ in Proposition 2.2 hold and the total gradient $\nabla \mathcal{F}(\omega)$ of $\mathcal{F}$ is expressed as $\nabla \mathcal{F}(\omega) = g_\omega + B_\omega a_\omega^\star$, with $a_\omega^\star \in \mathcal{H}$ being the minimizer of the objective $L_{adj}$ in Equation (5). Moreover, $L_{adj}$, $g_\omega$ and $B_\omega a_\omega^\star$ are given by:*

$$\begin{aligned} L_{adj}(\omega, a) = \ & \tfrac{1}{2} \, \mathbb{E}_{\mathbb{P}}\left[a(x)^\top \partial_v^2 \ell_{in}\left(\omega, h_\omega^\star(x), x, y\right) a(x)\right] \\ & + \mathbb{E}_{\mathbb{Q}}\left[a(x)^\top \partial_v \ell_{out}\left(\omega, h_\omega^\star(x), x, y\right)\right], \end{aligned} \tag{7}$$

$$g_\omega = \mathbb{E}_{\mathbb{Q}}\left[\partial_\omega \ell_{out}\left(\omega, h_\omega^\star(x), x, y\right)\right], \quad B_\omega a_\omega^\star = \mathbb{E}_{\mathbb{P}}\left[\partial_{\omega, v} \ell_{in}\left(\omega, h_\omega^\star(x), x, y\right) a_\omega^\star(x)\right], \tag{8}$$

*where $\partial_\omega \ell_{out}, \partial_v \ell_{out}, \partial_{\omega, v} \ell_{in}$, and $\partial_v^2 \ell_{in}$ are partial first and second order derivatives of $\ell_{out}$ and $\ell_{in}$ with respect to their first and second arguments $\omega$ and $v$.*

Assumptions (A) and (B) on $\mathbb{P}$ and $\mathbb{Q}$ ensure finite second moments and bounded Radon-Nikodym derivatives, maintaining square integrability under both distributions in Equation (6). Assumptions (C) to (L) on $\ell_{in}$ and $\ell_{out}$ primarily involve integrability, differentiability, Lipschitz continuity, and strong convexity of $\ell_{in}$ in its second argument, typically satisfied by objectives like mean squared error or cross entropy (see Proposition D.1 in Appendix D.1). Next, by leveraging Proposition 2.3, we derive practical algorithms for solving FBO using function approximation tools like neural networks.

## 3 Methods for Functional Bilevel Optimization in $L_2$ Spaces

We propose *Functional Implicit Differentiation (FuncID)*, a flexible class of algorithms for solving the functional bilevel problem in $L_2$ spaces described in Section 2.2 when samples from distributions $\mathbb{P}$ and $\mathbb{Q}$ are available.

*FuncID* relies on three main components detailed in the next subsections:

1. **Empirical objectives.** These approximate the objectives $L_{out}$, $L_{in}$ and $L_{adj}$ as empirical expectations over samples from inner and outer datasets $\mathcal{D}_{in}$ and $\mathcal{D}_{out}$.
2. **Function approximation.** The search space for both the prediction and adjoint functions is restricted to parametric spaces with finite-dimensional parameters $\theta$ and $\xi$. Approximate solutions $\hat{h}_\omega$ and $\hat{a}_\omega$ to the optimal functions $h_\omega^\star$ and $a_\omega^\star$ are obtained by minimizing the empirical objectives.
3. **Total gradient approximation.** *FuncID* estimates the total gradient $\nabla\mathcal{F}(\omega)$ using the empirical objectives, and the approximations $\hat{h}_\omega$ and $\hat{a}_\omega$ of the prediction and adjoint functions.

Algorithm 1 provides an outlines of *FuncID* which has a nested structure similar to AID: (1) inner-level optimizations (InnerOpt and AdjointOpt) to update the prediction and adjoint models using scalable algorithms such as stochastic gradient descent [Robbins and Monro, 1951], and (2) an outer-level optimization to update the parameter $\omega$ using a total gradient approximation TotalGrad. An optional *warm-start* allows initializing the parameters of both the prediction and adjoint models for the current outer-level iteration with those obtained from the previous one.

---

**Algorithm 1** *FuncID*

---

**Input:** initial outer, inner, and adjoint parameter $\omega_0, \theta_0, \xi_0$; warm-start option WS.
**for** $n = 0, \ldots, N-1$ **do**
    # *Optional warm-start*
    **if** WS=True **then** $(\theta_0, \xi_0) \leftarrow (\theta_n, \xi_n)$ **end if**
    # *Inner-level optimization*
    $\hat{h}_{\omega_n}, \theta_{n+1} \leftarrow$ InnerOpt$(\omega_n, \theta_0, \mathcal{D}_{in})$
    # *Adjoint optimization*
    $\hat{a}_{\omega_n}, \xi_{n+1} \leftarrow$ AdjointOpt$(\omega_n, \xi_0, \hat{h}_{\omega_n}, \mathcal{D})$
    # *Outer gradient estimation*
    Sample a mini-batch $\mathcal{B} = (\mathcal{B}_{out}, \mathcal{B}_{in})$ from $\mathcal{D} = (\mathcal{D}_{out}, \mathcal{D}_{in})$
    $g_{out} \leftarrow$ TotalGrad$(\omega_n, \hat{h}_{\omega_n}, \hat{a}_{\omega_n}, \mathcal{B})$
    $\omega_{n+1} \leftarrow$ update $\omega_n$ using $g_{out}$;
**end for**

---

### 3.1 From population losses to empirical objectives

We assume access to two datasets $\mathcal{D}_{in}$ and $\mathcal{D}_{out}$, comprising i.i.d. samples from $\mathbb{P}$ and $\mathbb{Q}$, respectively. This assumption can be relaxed, such as when using samples from a Markov chain or a Markov Decision Process to approximate population objectives. For scalability, we operate in a mini-batch setting, where batches $\mathcal{B} = (\mathcal{B}_{out}, \mathcal{B}_{in})$ are sub-sampled from datasets $\mathcal{D} := (\mathcal{D}_{out}, \mathcal{D}_{in})$. Approximating both inner and outer level objectives in (6) can be achieved using empirical versions:

$$\hat{L}_{out}(\omega, h, \mathcal{B}_{out}) := \frac{1}{|\mathcal{B}_{out}|} \sum_{(\tilde{x}, \tilde{y}) \in \mathcal{B}_{out}} \ell_{out}(\omega, h(\tilde{x}), \tilde{x}, \tilde{y}),$$

$$\hat{L}_{in}(\omega, h, \mathcal{B}_{in}) := \frac{1}{|\mathcal{B}_{in}|} \sum_{(x, y) \in \mathcal{B}_{in}} \ell_{in}(\omega, h(x), x, y).$$

**Adjoint objective.** Using the expression of $L_{adj}$ from Proposition 2.3, we derive a finite-sample approximation of the adjoint loss by replacing the population expectations by their empirical counterparts. More precisely, assuming we have access to an approximation $\hat{h}_\omega$ to the inner-level prediction function, we consider the following empirical version of the adjoint objective:

$$\hat{L}_{adj}\left(\omega, a, \hat{h}_\omega, \mathcal{B}\right) := \tfrac{1}{2}\tfrac{1}{|\mathcal{B}_{in}|} \sum_{(x,y)\in\mathcal{B}_{in}} a(x)^\top \partial_v^2 \ell_{in}(\omega, \hat{h}_\omega(x), x, y)\, a(x)$$
$$+ \tfrac{1}{|\mathcal{B}_{out}|} \sum_{(x,y)\in\mathcal{B}_{out}} a(x)^\top \partial_v \ell_{out}\left(\omega, \hat{h}_\omega(x), x, y\right). \quad (9)$$

The adjoint objective in Equation (9) requires computing a Hessian-vector product with respect to the output $v$ in $\mathbb{R}^{d_v}$ of the prediction function $\hat{h}_\omega$, which is typically of reasonably small dimension, unlike traditional AID methods that necessitate a Hessian-vector product with respect to some model parameters. Importantly, compared to AID, *FuncID* does not requires differentiating twice w.r.t the model's parameters $\tau(\theta)$ which results in memory and time savings as discussed in Appendix F.2.

## 3.2 Approximate prediction and adjoint functions

To find approximate solutions to the prediction and adjoint functions we rely on three steps: 1) specifying parametric search spaces for both functions, 2) introducing optional regularization to prevent overfitting and, 3) defining a gradient-based optimization procedure on the empirical objectives.

**Parametric search spaces.** We approximate both prediction and adjoint functions using parametric search spaces. A parametric family of functions defined by a map $\tau : \Theta \to \mathcal{H}$ over parameters $\Theta \subseteq \mathbb{R}^{p_{in}}$ constrains the prediction function $h$ as $h(x) = \tau(\theta)(x)$. We only require $\tau$ to be continuous and differentiable almost everywhere such that back-propagation can be applied [Bolte et al., 2021]. Notably, unlike AID, we do not need $\tau$ to be twice differentiable, as functional implicit differentiation computes the Hessian w.r.t. the output of $\tau$, not w.r.t. its parameters $\theta$. For flexibility, we can consider a different parameterized model $\nu : \Xi \to \mathcal{H}$ for approximating the adjoint function, defined over parameters $\Xi \subseteq \mathbb{R}^{p_{adj}}$, constraining the adjoint similarly to $\tau$. In practice, we often use the same parameterization, typically a neural network, for both the inner-level and the adjoint models.

**Regularization.** With empirical objectives and parametric search spaces, we can directly optimize parameters of both the inner-level model $\tau$ and the adjoint model $\nu$. However, to address finite sample issues, regularization may be introduced to these empirical objectives for better generalization. The method allows flexibility in regularization choice, accommodating functions $\theta \mapsto R_{in}(\theta)$ and $\xi \mapsto R_{adj}(\xi)$, such as ridge penalty or other commonly used regularization techniques

**Optimization.** The function `InnerOpt` (defined in Algorithm 2) optimizes inner model parameters for a given $\omega$, initialization $\theta_0$, and data $\mathcal{D}_{in}$, using $M$ gradient updates. It returns optimized parameters $\theta_M$ and the corresponding inner model $\hat{h}_\omega = \tau(\theta_M)$, approximating the inner-level solution. Similarly, `AdjointOpt` (defined in Algorithm 3) optimizes adjoint model parameters with $K$ gradient updates, producing the approximate adjoint function $\hat{a}_\omega = \nu(\xi_K)$. Other optimization procedures may also be used, especially when closed-form solutions are available, as exploited in some experiments in Section 4. Operations requiring differentiation can be implemented using standard optimization procedures with automatic differentiation packages like PyTorch [Paszke et al., 2019] or Jax [Bradbury et al., 2018].

---

| **Algorithm 2** `InnerOpt`$(\omega, \theta_0, \mathcal{D}_{in})$ | **Algorithm 3** `AdjointOpt`$(\omega, \xi_0, \hat{h}_\omega, \mathcal{D})$ |
|---|---|
| **for** $m = 0, \ldots, M-1$ **do** | **for** $k = 0, \ldots, K-1$ **do** |
|     Sample batch $\mathcal{B}_{in}$ from $\mathcal{D}_{in}$ |     Sample batch $\mathcal{B}$ from $\mathcal{D}$ |
|     $g_{in} \leftarrow \nabla_\theta[\hat{L}_{in}\left(\omega, \tau(\theta_m), \mathcal{B}_{in}\right) + R_{in}(\theta_m)]$ |     $g_{adj} \leftarrow \nabla_\xi[\hat{L}_{adj}(\omega, \nu(\xi_t), \hat{h}_\omega, \mathcal{B}) + R_{adj}(\xi_k)]$ |
|     $\theta_{m+1} \leftarrow$ Update $\theta_m$ using $g_{in}$ |     $\xi_{k+1} \leftarrow$ Update $\xi_k$ using $g_{adj}$ |
| **end for** | **end for** |
| **Return** $\tau(\theta_M), \theta_M$ | **Return** $\nu(\xi_K), \xi_K$ |

## 3.3 Total gradient estimation

We exploit Proposition 2.3 to derive Algorithm 4, which allows us to approximate the total gradient $\nabla \mathcal{F}(\omega)$ after computing the approximate solutions $\hat{h}_\omega$ and $\hat{a}_\omega$. There, we decompose the gradient into two terms: $g_{Exp}$, an empirical approximation of $g_\omega$ in Equation (8) representing the explicit dependence of $L_{out}$ on the outer variable $\omega$, and $g_{Imp}$, an approximation to the implicit gradient term $B_\omega a_\omega^\star$ in Equation (8). Both terms are obtained by replacing the expectations by empirical averages batches $\mathcal{B}_{in}$ and $\mathcal{B}_{out}$, and using the approximations $\hat{h}_\omega$ and $\hat{a}_\omega$ instead of the exact solutions.

---

**Algorithm 4** TotalGrad($\omega, \hat{h}_\omega, \hat{a}_\omega, \mathcal{B}$)

$g_{Exp} \leftarrow \partial_\omega \hat{L}_{out}(\omega, \hat{h}_\omega, \mathcal{B}_{out})$

$g_{Imp} \leftarrow \frac{1}{|\mathcal{B}_{in}|} \sum_{(x,y) \in \mathcal{B}_{in}} \partial_{\omega,v} \ell_{in}(\omega, \hat{h}_\omega(x), x, y) \, \hat{a}_\omega(x)$

**Return** $g_{Exp} + g_{Imp}$

---

## 3.4 Convergence Guarantees

Convergence of Algorithm 1 to stationary points of $\mathcal{F}$ depends on approximation errors, which result from sub-optimal inner and adjoint solutions, as shown by the convergence result below.

**Theorem 3.1.** *Assume that $\mathcal{F}$ is $\mathcal{L}$-smooth and admits a finite lower bound $\mathcal{F}^\star$. Use an update rule $\omega_{n+1} = \omega_n - \eta g_{out}$ with step size $0 < \eta \leq \frac{1}{4\mathcal{L}_\omega}$ in Algorithm 1. Under Assumption (a) on sub-optimality of $\hat{h}_\omega$ and $\hat{a}_\omega$, Assumptions (b) to (h) on the smoothness of $\ell_{in}$ and $\ell_{out}$, all stated in Appendix E.1, and the assumptions in Proposition 2.3, the iterates $\{\omega_n\}_{n \geq 0}$ of Algorithm 1 satisfy:*

$$\min_{0 \leq n \leq N-1} \mathbb{E} \left[ \|\nabla \mathcal{F}(\omega_n)\|^2 \right] \leq \frac{4 \left( \mathcal{F}(\omega_0) - \mathcal{F}^\star \right)}{\eta N} + 2\eta \mathcal{L} \sigma_{eff}^2 + (c_1 \epsilon_{in} + c_2 \epsilon_{adj}),$$

*where $c_1, c_2, \sigma_{eff}^2$ are positive constants, and $\epsilon_{in}, \epsilon_{adj}$ are sub-optimality errors that result from the inner and adjoint optimization procedures.*

Theorem 3.1 is proven in Appendix E and relies on the general convergence result in Demidovich et al. [2024, Theorem 3] for stochastic biased gradient methods. The key idea is to control both bias and variance of the gradient estimator in terms of generalization errors $\epsilon_{in}$ and $\epsilon_{adj}$ when approximating the inner and adjoint solutions. These generalization errors can, in turn, be made smaller in the case of over-parameterized networks, by increasing network capacity, number of steps and sample size [Allen-Zhu et al., 2019, Du et al., 2019, Zou et al., 2020].

## 4 Applications

We consider two applications of the functional bilevel optimization problem: Two-stage least squares regression (2SLS) and Model-based reinforcement learning. To illustrate its effectiveness we compare it with other bilevel optimization approaches like AID or ITD, as well as state-of-the-art methods for each application. We provide a versatile implementation of *FuncID* (https://github.com/inria-thoth/funcBO) in PyTorch [Paszke et al., 2019], compatible with standard optimizers (*e.g.*, Adam [Kingma and Ba, 2015]), and supports common regularization techniques. For the reinforcement learning application, we extend an existing JAX [Bradbury et al., 2018] implementation of model-based RL from Nikishin et al. [2022] to apply *FuncID*. To ensure fairness, experiments are conducted with comparable computational budgets for hyperparameter tuning using the MLXP experiment management tool [Arbel and Zouaoui, 2024]. Additionally, we maintain consistency by employing identical neural network architectures across all methods and repeating experiments multiple times with different random seeds.

### 4.1 Two-stage least squares regression (2SLS)

Two-stage least squares regression (2SLS) is commonly used in causal representation learning, including instrumental regression or proxy causal learning [Stock and Trebbi, 2003]. Recent studies have applied bilevel optimization approaches to address 2SLS, yielding promising results [Xu et al.,

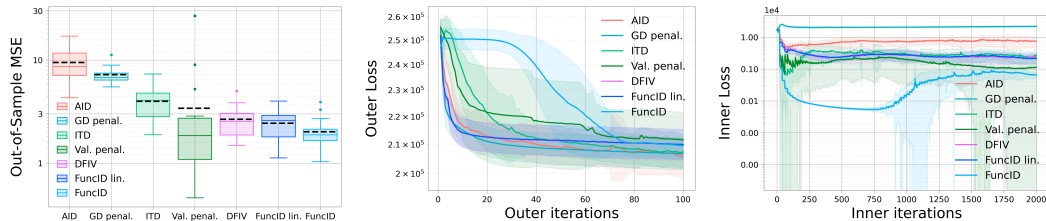

Figure 2: Performance metrics for Instrumental Variable (IV) regression. All results are averaged over 20 runs with 5000 training samples and 588 test samples. (**Left**) box plot of the test loss, with the dashed black line indicating the mean test error. (**Middle**) outer loss vs training iterations, (**Right**) inner loss vs training iterations. The bold lines in the middle and right plots indicate the mean loss, the shaded area corresponds to standard deviation.

2021b,a, Hu et al., 2023]. We particularly focus on 2SLS for Instrumental Variable (IV) regression, a widely-used statistical framework for mitigating endogeneity in econometrics [Blundell et al., 2007, 2012], medical economics [Cawley and Meyerhoefer, 2012], sociology [Bollen, 2012], and more recently, for handling confounders in off-line reinforcement learning [Fu et al., 2022].

**Problem formulation.**    In an IV problem, the objective is to model $f_\omega : t \mapsto o$ that approximates the structural function $f_{struct}$ using independent samples $(o, t, x)$ from a data distribution $\mathbb{P}$, where $x$ is an instrumental variable. The structural function $f_{struct}$ delineates the true effect of a treatment $t$ on an outcome $o$. A significant challenge in IV is the presence of an unobserved confounder $\epsilon$, which influences both $t$ and $o$ additively, rendering standard regression ineffective for recovering $f_\omega$. However, if the instrumental variable $x$ solely impacts the outcome $o$ through the treatment $t$ and is independent from the confounder $\epsilon$, it can be employed to elucidate the direct relationship between the treatment $t$ and the outcome $o$ using the 2SLS framework, under mild assumptions on the confounder [Singh et al., 2019]. This adaptation replaces the regression problem with a variant that averages the effect of the treatment $t$ conditionally on $x$

$$\min_{\omega \in \Omega} \mathbb{E}_\mathbb{P} \left[ \| o - \mathbb{E}_\mathbb{P} \left[ f_\omega(t) | x \right] \|^2 \right]. \tag{10}$$

Directly estimating the conditional expectation $\mathbb{E}_\mathbb{P} [f_\omega(t)|x]$ is hard in general. Instead, it is easier to express it, equivalently, as the solution of another regression problem predicting $f_\omega(t)$ from $x$:

$$h_\omega^\star := \mathbb{E}_\mathbb{P} [f_\omega(t)|x] = \arg \min_{h \in \mathcal{H}} \mathbb{E}_\mathbb{P} \left[ \| f_\omega(t) - h(x) \|^2 \right]. \tag{11}$$

Both equations result in the bilevel formulation in Equation (6) with $y = (t, o)$, $\mathbb{Q} = \mathbb{P}$ and the point-wise losses $\ell_{in}$ and $\ell_{out}$ given by $\ell_{in}(\omega, v, x, y) = \ell_{in}(\omega, v, x, (t, o)) = \| f_\omega(t) - v \|^2$ and $\ell_{out}(\omega, v, x, y) = \ell_{out}(\omega, v, x, (t, o)) = \| o - v \|^2$. It is, therefore, possible to directly apply Algorithm 1 to learn $f_\omega$ as we illustrate below.

**Experimental setup.**    We study the IV problem using the *dsprites* dataset [Matthey et al., 2017], comprising synthetic images representing single objects generated from five latent parameters: *shape, scale, rotation*, and *posX, posY* positions on image coordinates. Here, the treatment variable $t$ is the images, the hidden confounder $\epsilon$ is the *posY* coordinate, and the other four latent variables form the instrumental variable $x$. The outcome $o$ is an unknown structural function $f_{struct}$ of $t$, contaminated by confounder $\epsilon$ as detailed in Appendix G.1. We follow the setup of the Deep Feature Instrumental Variable Regression (DFIV) *dsprites* experiment by Xu et al. [2021a, Section 4.2], which achieves state-of-the-art performance. In this setup, neural networks serve as the prediction function and structural model, optimized to solve the bilevel problem in Equations (10) and (11). We explore two versions of our method: *FuncID*, which optimizes all adjoint network parameters, and *FuncID linear*, which learns only the last layer in closed-form while inheriting hidden layer parameters from the inner prediction function. We compare our method with DFIV, AID, ITD, and Penalty-based methods: gradient penalty (GD penal.) and value function penalty (Val penal.) [Shen and Chen, 2023], using identical network architectures and computational budgets for hyperparameter selection. Full details on network architectures, hyperparameters, and training settings are provided in Appendix G.2.

**Results.** Figure 2 compares structural models learned by different methods using 5K training samples (refer to Figure 6 in Appendix G.3 for 10K sample results). The left subplot illustrates out-of-sample mean squared error of learned structural models compared to ground truth outcomes (uncontaminated by noise $\epsilon$), while the middle and right subplots show the evolution of outer and inner objectives over iterations. For the 5K dataset, *FuncID* outperforms DFIV (*p-value*=0.003, one-sided paired t-test), while showing comparable performance on the 10K dataset (Figure 6). AID and ITD perform notably worse, indicating their parametric approach fails to fully leverage the functional structure. *FuncID* outperforms the gradient penalty method and performs either better or comparably to the value function penalty method, though the latter shows higher variance with some particularly bad outliers. While all methods achieve similar outer losses, this criterion alone is only reliable as an indicator of convergence when evaluated near the 'exact' inner-level solution corresponding to the lowest inner-loss values. Interestingly, FuncID obtains the lowest inner-loss values, suggesting its outer-loss is a more reliable indicator of convergence.

## 4.2 Model-based reinforcement learning

Model-based reinforcement learning (RL) naturally yields bilevel optimization formulations, since several components of an RL agent need to be learned using different objectives. Recently, Nikishin et al. [2022] showed that casting model-based RL as a bilevel problem can result in better tolerance to model-misspecification. Our experiments show that the functional bilevel framework yields improved results even when the model is well-specified, suggesting a broader use of the bilevel formulation.

**Problem formulation.** In model-based RL, the Markov Decision Process (MDP) is approximated by a probabilistic model $q_\omega$ with parameters $\omega$ that can predict the next state $s_\omega(x)$ and reward $r_\omega(x)$, given a pair $x := (s, a)$ where $s$ is the current environment state and $a$ is the action of an agent. A second model $h$ can be used to approximate the action-value function $h(x)$ that computes the expected cumulative reward given the current state-action pair. Traditionally, the action-value function is learned using the current MDP model, while the latter is learned independently from the action-value function using Maximum Likelihood Estimation (MLE) [Sutton, 1991].

In the bilevel formulation of model-based RL by Nikishin et al. [2022], the inner-level problem involves learning the optimal action-value function $h_\omega^\star$ with the current MDP model $q_\omega$ and minimizing the Bellman error. The inner-level objective can be expressed as an expectation of a point-wise loss $f$ with samples $(x, r', s') \sim \mathbb{P}$, derived from the agent-environment interaction:

$$h_\omega^\star = \arg \min_{h \in \mathcal{H}} \mathbb{E}_\mathbb{P} \left[ f(h(x), r_\omega(x), s_\omega(x)) \right]. \tag{12}$$

Here, the future state and reward $(r', s')$ are replaced by the MDP model predictions $r_\omega(x)$ and $s_\omega(x)$. In practice, samples from $\mathbb{P}$ are obtained using a replay buffer. The buffer accumulates data over several episodes of interactions with the environment, and can therefore be considered independent of the agent's policy. The point-wise loss function $f$ represents the error between the action-value function prediction and the expected cumulative reward given the current state-action pair:

$$f(v, r', s') := \frac{1}{2} \left\| v - r' - \gamma \log \sum_{a'} e^{\bar{h}(s', a')} \right\|^2,$$

with $\bar{h}$ a lagged version of $h$ (exponentially averaged network) and $\gamma$ a discount factor. The MDP model is learned implicitly using the optimal function $h_\omega^\star$, by minimizing the Bellman error w.r.t. $\omega$:

$$\min_{\omega \in \Omega} \mathbb{E}_\mathbb{P} \left[ f(h_\omega^\star(x), r', s') \right]. \tag{13}$$

Equations (12) and (13) define a bilevel problem as in Equation (6), where $\mathbb{Q} = \mathbb{P}$, $y = (r', s')$, and the point-wise losses $\ell_{in}$ and $\ell_{out}$ are given by: $\ell_{in}(\omega, v, x, y) = f(v, r_\omega(x), s_\omega(x))$ and $\ell_{out}(\omega, v, x, y) = f(v, r', s')$. Therefore, we can directly apply Algorithm 1 to learn both the MDP model $q_\omega$ and the optimal action-value function $h_\omega^\star$.

**Experimental setup.** We apply *FuncID* to the *CartPole* control problem, a classic benchmark in reinforcement learning [Brockman et al., 2016, Nagendra et al., 2017]. The goal is to balance a pole attached to a cart by moving the cart horizontally. Following Nikishin et al. [2022], we use a model-based approach and consider two scenarios: one with a well-specified network accurately representing the MDP, and another with a misspecified model having fewer hidden layer units, limiting its capacity.

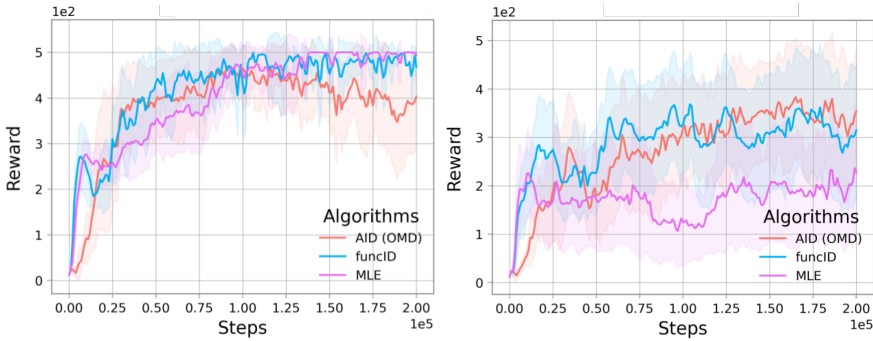

Figure 3: Average reward on an evaluation environment vs. training iterations on the *CartPole* task. (**Left**) Well-specified model with 32 hidden units. (**Right**) Misspecified model with 3 hidden units. Both plots show mean reward over 10 runs where the shaded region is the 95% confidence interval.

Using the bilevel formulation in Equations (12) and (13), we compare *FuncID* with the Optimal Model Design (OMD) algorithm [Nikishin et al., 2022], a variant of AID. Additionally, we compare against a standard single-level model-based RL formulation using Maximum Likelihood Estimation (MLE) [Sutton, 1991]. For the adjoint function in *FuncID*, we derive a simple closed-form expression based on the structure of the adjoint objective (see Appendix H.1). We follow the experimental setup of Nikishin et al. [2022], providing full details and hyperparameters in Appendix H.2.

**Results.** Figure 3 illustrates the training reward evolution for *FuncID*, OMD, and MLE in both well-specified and misspecified scenarios. *FuncID* consistently performs well across settings. In the well-specified case, where OMD achieves a reward of 4, *FuncID* reaches the maximum reward of 5, matching MLE (left Figure 3). In the misspecified scenario, *FuncID* performs comparably to OMD and significantly outperforms MLE (right Figure 3). Moreover, *FuncID* tends to converge faster than MLE (see Figure 7 in Appendix H.3) and yields consistently better prediction error than OMD (see Figure 8 in Appendix H.3). These findings align with Nikishin et al. [2022], suggesting that MLE may prioritize minimizing prediction errors, potentially leading to overfitting irrelevant features. In contrast, OMD and *FuncID* focus on maximizing expected returns, especially in the presence of model misspecification. Our results highlight the effectiveness of (FBO) even in well-specified settings, suggesting, for future work, further investigations for more general RL tasks.

## 5  Discussion and concluding remarks

This paper introduced a functional paradigm for bilevel optimization in machine learning, shifting focus from parameter space to function space. The proposed approach specifically addresses the ambiguity challenge arising from using deep networks in bilevel optimization. The paper establishes the validity of the functional framework by developing a theory of functional implicit differentiation, proving convergence for the proposed *FuncID* method, and numerically comparing it with other bilevel optimization methods.

The theoretical foundations of our work rely on several key assumptions worth examining. While our convergence guarantees assume both inner and adjoint optimization problems are solved to some optimality, this assumption is supported by recent results on global convergence in over-parameterized networks [Allen-Zhu et al., 2019, Liu et al., 2022a]. However, quantifying these optimality errors more precisely and understanding their relationship to optimization procedures remains an open challenge. Additionally, like other bilevel methods, our approach requires careful hyperparameter selection, which can impact practical implementation.

Several promising directions emerge for future research. While we focus on $L_2$ spaces, exploring alternative function spaces (such as Reproducing Kernel Hilbert Spaces or Sobolev spaces) could reveal additional advantages for specific applications. Furthermore, extending our framework to non-smooth objectives or constrained optimization problems, potentially building on existing work in non-smooth implicit differentiation [Bolte et al., 2022], would broaden its applicability.

## Acknowledgments

This work was supported by the ERC grant number 101087696 (APHELAIA project) and by ANR 3IA MIAI@Grenoble Alpes (ANR-19-P3IA-0003) and the ANR project BONSAI (grant ANR-23-CE23-0012-01). We thank Edouard Pauwels and Samuel Vaiter for their insightful discussions.

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

# A    Examples of FBO formulations

The functional bilevel setting applies to various practical bilevel problems where objectives depend solely on model predictions, not their parameterization. Below, we discuss a few examples.

**Auxiliary task learning.**    As in Equation (1), consider a *main* prediction task with features $x$ and labels $y$, equipped with a loss function $f(y, h(x))$. The goal of auxiliary task learning is to learn how a set of auxiliary tasks represented by a vector $f_{aux}(y, h(x))$ could help solve the main task. This problem is formulated by Navon et al. [2021] as a bilevel problem, which can be written as (FBO) with

$$L_{out}(\omega, h) = \mathbb{E}_{(y,x) \sim \mathcal{D}_{val}} \left[ f(y, h(x)) \right],$$

where the loss is evaluated over a validation dataset $\mathcal{D}_{val}$, and

$$L_{in}(\omega, h) = \mathbb{E}_{(y,x) \sim \mathcal{D}_{train}} \left[ f(y, h(x)) + g_{\omega}(f_{aux}(y, h(x))) \right],$$

where an independent training dataset $\mathcal{D}_{train}$ is used, and $g_{\omega}$ is a function that combines the auxiliary losses into a scalar value.

**Task-driven metric learning.**    Considering now a regression problem with features $x$ and labels $y$, the goal of task-driven metric learning formulated by Bansal et al. [2023] is to learn a metric parameterized by $\omega$ for the regression task such that the corresponding predictor $h_{\omega}^{\star}$ performs well on a downstream task $L_{task}$. This can be formulated as (FBO) with $L_{out}(\omega, h) = L_{task}(h)$ and

$$L_{in}(\omega, h) = \mathbb{E}_{(y,x)} \left[ \|y - h(x)\|_{A_{\omega}(x)}^2 \right],$$

where $\| \cdot \|_{\omega}^2$ is the squared Mahalanobis norm with parameters $\omega$ and $A_{\omega}(x)$ is a data-dependent metric that allows emphasizing features that are more important for the downstream task.

# B    Additional Related Work

**Bilevel optimization in machine learning.**    Two families of bilevel methods are prevalent in machine learning due to their scalability: iterative (or 'unrolled') differentiation (ITD, Baydin et al., 2017) and Approximate Implicit Differentiation (AID, Ghadimi and Wang, 2018). ITD approximates the optimal inner-level solution using an 'unrolled' function from a sequence of differentiable optimization steps, optimizing the outer variable via back-propagation [Shaban et al., 2019, Bolte et al., 2024]. The gradient approximation error decreases linearly with the number of steps when the inner-level is strongly convex, though at increased computational and memory costs [Grazzi et al., 2020, Theorem 2.1]. ITD is popular for its simplicity and availability in major deep learning libraries [Bradbury et al., 2018], but can be unstable, especially with non-convex inner objectives [Pascanu et al., 2013, Bengio et al., 1994, Arbel and Mairal, 2022b]. AID uses the Implicit Function Theorem (IFT) to derive the Jacobian of the inner-level solution with respect to the outer variable [Lorraine et al., 2019, Pedregosa, 2016], solving a finite-dimensional linear system for an adjoint vector representing optimality constraints. AID offers strong convergence guarantees for smooth, strongly convex inner objectives [Ji et al., 2021, Arbel and Mairal, 2022a]. However, without strong convexity, the linear system can become ill-posed due to a degenerate Hessian, leading to instabilities, especially with overparameterized deep neural networks. Our proposed approach avoids this issue, even when using deep networks for function approximation.

**Adjoint sensitivity method.**    The adjoint sensitivity method [Pontryagin, 2018] efficiently differentiates a controlled variable with respect to a control parameter. In bilevel optimization, AID applies a finite-dimensional version of this method [Margossian and Betancourt, 2021, Section 2]. Infinite-dimensional versions have differentiated solutions of ordinary differential equations (ODEs) with respect to defining parameters [Margossian and Betancourt, 2021, Section 3], and have been used in machine learning to optimize parameters of a vector field describing an ODE [Chen et al., 2018]. Here, the ODE's vector field, parameterized by a neural network, is optimized to match observations. The adjoint sensitivity method offers an efficient alternative to the unstable process of back-propagation through ODE solvers, requiring only the solution of an adjoint ODE to compute gradient updates, improving performance [Jia and Benson, 2019, Zhong et al., 2019, Li et al., 2020]. This method has been adapted to meta-learning [Li et al., 2023], viewing the inner optimization as an

ODE evolution with gradients obtained via the adjoint ODE. Recently, Marion et al. [2024] employ the adjoint sensitivity method for optimizing diffusion models, where an adjoint SDE is solved to compute the total gradient. Unlike these works, which use the adjoint method for ODE/SDE solutions as functions of time, our work applies an infinite-dimensional version of the adjoint sensitivity method to general learning problems, where solutions are functions of input data.

**Amortization.**    Recently, several methods have used amortization to approximately solve bilevel problems [MacKay et al., 2019, Bae and Grosse, 2020]. These methods employ a parametric model called a *hypernetwork* [Ha et al., 2017, Brock et al., 2018, Zhang et al., 2019], optimized to directly predict the inner-level solution given the outer-level parameter $\omega$. Amortized methods treat the two levels as independent optimization problems: (1) learning the hypernetwork for a range of $\omega$, and (2) performing first-order descent on $\omega$ using the hypernetwork as a proxy for the inner-level solution. Unlike ITD, AID, or our functional implicit differentiation method, amortized methods do not fully exploit the implicit dependence between the two levels. They are similar to amortized variational inference [Kingma and Welling, 2014, Rezende et al., 2014], where a parametric model produces approximate samples from a posterior distribution. Amortization methods perform well when the inner solution's dependence on $\omega$ is simple but may fail otherwise [Amos et al., 2023, pages 71-72]. In contrast, our functional implicit differentiation framework adapts to complex implicit dependencies between the inner solution and $\omega$.

# C    Theory for Functional Implicit Differentiation

## C.1    Preliminary results

We recall the definition of Hadamard differentiability and provide in Proposition C.2 a general property for Hadamard differentiable maps that we will exploit to prove Theorem 2.1 in Appendix C.2.

**Definition C.1. Hadamard differentiability.** Let $A$ and $B$ be two separable Banach spaces. A function $L : A \to B$ is said to be *Hadamard differentiable* [van der Vaart, 1991, Fang and Santos, 2018] if for any $a \in A$, there exist a continuous linear map $d_a L(a) : A \to B$ so that for any sequence $(u_n)_{n \geq 1}$ in $A$ converging to an element $u \in A$, and any real valued and non-vanishing sequence $(t_n)_{n \geq 1}$ converging to 0, it holds that:

$$\left\| \frac{1}{t_n} \left( L(a + t_n u_n) - L(a) \right) - d_a L(a) u \right\| \xrightarrow[n \to +\infty]{} 0. \tag{14}$$

**Proposition C.2.** *Let $A$ and $B$ be two separable Banach spaces. Let $L : A \to B$ be a Hadamard differentiable map with differential $d_a L$ at point $a$. Consider a bounded linear map defined over a euclidean space $\mathbb{R}^n$ of finite dimension $n$ and taking values in $A$, i.e $J : \mathbb{R}^n \to A$. Then, the following holds:*

$$L(a + Ju) = L(a) + d_a L(a) Ju + o(\|u\|).$$

*Proof.* Consider a sequence $(u_k)_{k \geq 1}$ in $\mathbb{R}^n$ so that $u_k$ converges to 0 with $\|u_k\| > 0$ for all $k \geq 1$ and define the first order error $E_k$ as follows:

$$E_k = \frac{1}{\|u_k\|} \|L(a + Ju_k) - L(a) - d_a L(a) u_k\|.$$

The goal is to show that $E_k$ converges to 0. We can write $u_k$ as $u_k = t_k \tilde{u}_k$ with $t_k = \|u_k\|$ and $\|\tilde{u}_k\| = 1$, so that:

$$E_k = \left\| \frac{1}{\|t_k\|} \left( L(a + t_k J\tilde{u}_k) - L(a) \right) - d_a L(a) \tilde{u}_k \right\|.$$

If $E_k$ were unbounded, then, by contradiction, there must exist a subsequence $(E_{\phi(k)})_{k \geq 1}$ converging to $+\infty$, with $\phi(k)$ increasing and $\phi(k) \to +\infty$. Moreover, since $\tilde{u}_k$ is bounded, one can further choose the subsequence $E_{\phi(k)}$ so that $\tilde{u}_{\phi(k)}$ converges to some element $\tilde{u}$. We can use the following upper-bound:

$$E_k \leq \underbrace{\left\| \frac{1}{\|t_k\|} \left( L(a + t_k J\tilde{u}_k) - L(a) \right) - d_a L(a) \tilde{u} \right\|}_{\tilde{E}_k} + \|d_a L(a)\| \|\tilde{u}_k - \tilde{u}\|, \tag{15}$$

where we used that $d_a L(a)$ is bounded. Since $L$ is Hadamard differentiable, $\tilde{E}_{\phi(k)}$ converges to 0. Moreover, $\left\| \tilde{u}_{\phi(k)} - \tilde{u} \right\|$ also converges to 0. Hence, $E_{\phi(k)}$ converges to 0 which contradicts $E_{\phi(k)} \to +\infty$. Therefore, $E_k$ is bounded.

Consider now any convergent subsequence of $(E_k)_{k \geq 1}$. Then, it can be written as $(E_{\phi(k)})_{k \geq 1}$ with $\phi(k)$ increasing and $\phi(k) \to +\infty$. We then have $E_{\phi(k)} \to e < +\infty$ by construction. Since $\tilde{u}_k$ is bounded, one can further choose the subsequence $E_{\phi(k)}$ so that $\tilde{u}_{\phi(k)}$ converges to some element $\tilde{u}$. Using again Equation (15) and the fact that $L$ is Hadamard differentiable, we deduce that $\tilde{E}_{\phi(k)}$ must converge to 0, and by definition of $\tilde{u}_{\phi(k)}$, that $\left\| \tilde{u}_{\phi(k)} - \tilde{u} \right\|$ converges to 0. Therefore, it follows that $E_{\phi(k)} \to 0$, so that $e = 0$. We then have shown that $(E_k)_{k \geq 1}$ is a bounded sequence and every subsequence of it converges to 0. Therefore, $E_k$ must converge to 0, which concludes the proof. $\square$

## C.2   Proof of the Functional implicit differentiation theorem

*Proof of Theorem 2.1.* The proof strategy consists in establishing the existence and uniqueness of the solution map $\omega \mapsto h_\omega^\star$, deriving a candidate Jacobian for it, then proving that $\omega \mapsto h_\omega^\star$ is differentiable.

**Existence and uniqueness of a solution map** $\omega \mapsto h_\omega^\star$**.** Let $\omega$ in $\Omega$ be fixed. The map $h \mapsto L_{in}(\omega, h)$ is lower semi-continuous since it is Fréchet differentiable by assumption. It is also strongly convex. Therefore, it admits a unique minimier $h_\omega^\star$ [Bauschke and Combettes, 2011, Corollary 11.17]. We then conclude that the map $\omega \mapsto h_\omega^\star$ is well-defined on $\Omega$.

**Strong convexity inequalities.** We provide two inequalities that will be used for proving differentiability of the map $\omega \mapsto h_\omega^\star$. The map $h \mapsto L_{in}(\omega, h)$ is Fréchet differentiable on $\mathcal{H}$ and $\mu$-strongly convex (with $\mu$ positive by assumption). Hence, for all $h_1, h_2$ in $\mathcal{H}$ the following quadratic lower-bound holds:

$$L_{in}(\omega, h_2) \geq L_{in}(\omega, h_1) + \langle \partial_h L_{in}(\omega, h_1), (h_2 - h_1) \rangle_{\mathcal{H}} + \frac{\mu}{2} \|h_2 - h_1\|_{\mathcal{H}}^2. \tag{16}$$

From the inequality above, we can also deduce that $h \mapsto \partial_h L_{in}(\omega, h)$ is a $\mu$-strongly monotone operator:

$$\langle \partial_h L_{in}(\omega, h_1) - \partial_h L_{in}(\omega, h_2), h_1 - h_2 \rangle_{\mathcal{H}} \geq \mu \|h_1 - h_2\|_{\mathcal{H}}^2. \tag{17}$$

Finally, note that, since $h \mapsto L_{in}(\omega, h)$ is Fréchet differentiable, its gradient must vanish at the optimum $h_\omega^\star$, i.e :

$$\partial_h L_{in}(\omega, h_\omega^\star) = 0. \tag{18}$$

**Candidate Jacobian for** $\omega \mapsto h_\omega^\star$**.** Let $\omega$ be in $\Omega$. Using Equation (17) with $h_1 = h + tv$ and $h_2 = h$ for some $h, v \in \mathcal{H}$, and a non-zeros real number $t$ we get:

$$\frac{1}{t} \langle \partial_h L_{in}(\omega, h + tv) - \partial_h L_{in}(\omega, h), v \rangle_{\mathcal{H}} \geq \mu \|v\|^2. \tag{19}$$

By assumption, $h \mapsto \partial_h L_{in}(\omega, h)$ is Hadamard differentiable and, a fortiori, directionally differentiable. Thus, by taking the limit when $t \to 0$, it follows that:

$$\langle \partial_h^2 L_{in}(\omega, h) v, v \rangle_{\mathcal{H}} \geq \mu \|v\|^2. \tag{20}$$

Hence, $\partial_h^2 L_{in}(\omega, h) : \mathcal{H} \to \mathcal{H}$ defines a coercive quadratic form. By definition of Hadamard differentiability, it is also bounded. Therefore, it follows from Lax-Milgram's theorem [Debnath and Mikusinski, 2005, Theorem 4.3.16], that $\partial_h^2 L_{in}(\omega, h)$ is invertible with a bounded inverse. Moreover, recalling that $B_\omega = \partial_{\omega, h} L_{in}(\omega, h_\omega^\star)$ is a bounded operator, its adjoint $(B_\omega)^\star$ is also a bounded operator from $\Omega$ to $\mathcal{H}$. Therefore, we can define $J = -C_\omega^{-1}(B_\omega)^\star$ which is a bounded linear map from $\Omega$ to $\mathcal{H}$ and will be our candidate Jacobian.

**Differentiability of** $\omega \mapsto h_\omega^\star$**.** By the strong convexity assumption (locally in $\omega$), there exists an open ball $\mathcal{B}$ centered at the origin 0 that is small enough so that we can ensure the existence of $\mu > 0$ for which $h \mapsto L_{in}(\omega + \epsilon, h)$ is $\mu$-strongly convex for all $\epsilon \in \mathcal{B}$. For a given $\epsilon \in \mathcal{B}$, we use the $\mu$-strong monotonicity of $h \mapsto \partial_h L_{in}(\omega + \epsilon, h)$ (17) at points $h_\omega^\star + J\epsilon$ and $h_{\omega+\epsilon}^\star$ to get:

$$\mu \left\| h_{\omega+\epsilon}^\star - h_\omega^\star - J\epsilon \right\|^2 \leq \langle (\partial_h L_{in}(\omega + \epsilon, h_{\omega+\epsilon}^\star) - \partial_h L_{in}(\omega + \epsilon, h_\omega^\star + J\epsilon)), (h_{\omega+\epsilon}^\star - h_\omega^\star - J\epsilon) \rangle_{\mathcal{H}}$$
$$= \langle -\partial_h L_{in}(\omega + \epsilon, h_\omega^\star + J\epsilon), (h_{\omega+\epsilon}^\star - h_\omega^\star - J\epsilon) \rangle_{\mathcal{H}}$$
$$\leq \|\partial_h L_{in}(\omega + \epsilon, h_\omega^\star + J\epsilon)\|_{\mathcal{H}} \|h_{\omega+\epsilon}^\star - h_\omega^\star - J\epsilon\|_{\mathcal{H}},$$

where the second line follows from optimality of $h_{\omega+\epsilon}^\star$ (Equation (18)), and the last line uses Cauchy-Schwarz's inequality. The above inequality allows us to deduce that:

$$\left\| h_{\omega+\epsilon}^\star - h_\omega^\star - J\epsilon \right\| \leq \frac{1}{\mu} \left\| \partial_h L_{in}\left(\omega + \epsilon, h_\omega^\star + J\epsilon\right) \right\|_{\mathcal{H}}. \tag{21}$$

Moreover, since $\partial_h L_{in}$ is Hadamard differentiable, by Proposition C.2 it follows that:

$$\partial_h L_{in}\left(\omega + \epsilon, h_\omega^\star + J\epsilon\right) = \underbrace{\partial_h L_{in}\left(\omega, h_\omega^\star\right)}_{=0} + d_{(\omega,h)}\partial_h L_{in}(\omega, h_\omega^\star)(\epsilon, J\epsilon) + o(\|\epsilon\|), \tag{22}$$

where the first term vanishes as a consequence of Equation (18), since $h_\omega^\star$ is a minimizer of $h \mapsto L_{in}(\omega, h)$. Additionally, note that the differential $d_{(\omega,h)}\partial_h L_{in}(\omega, h) : \Omega \times \mathcal{H} \to \mathcal{H}$ acts on elements $(\epsilon, g) \in \Omega \times \mathcal{H}$ as follows:

$$d_{(\omega,h)}\partial_h L_{in}(\omega, h)(\epsilon, g) = \partial_h^2 L_{in}(\omega, h)g + (\partial_{\omega,h} L_{in}(\omega, h))^\star \epsilon, \tag{23}$$

where $\partial_h^2 L_{in}(\omega, h) : \mathcal{H} \to \mathcal{H}$ and $\partial_{\omega,h} L_{in}(\omega, h) : \mathcal{H} \to \Omega$ are bounded operators and $(\partial_{\omega,h} L_{in}(\omega, h))^\star$ denotes the adjoint of $\partial_{\omega,h} L_{in}(\omega, h)$. By definition of $J$, and using Equation (23), it follows that:

$$d_{(\omega,h)}\partial_h L_{in}(\omega, h)(\epsilon, J\epsilon) = C_\omega J\epsilon + B_\omega \epsilon = 0.$$

Therefore, combining Equation (22) with the above equality yields:

$$\partial_h L_{in}\left(\omega + \epsilon, h_\omega^\star + J\epsilon\right) = o(\|\epsilon\|). \tag{24}$$

Finally, combining Equation (21) with the above equality directly shows that $\left\| h_{\omega+\epsilon}^\star - h_\omega^\star - J\epsilon \right\| \leq \frac{1}{\mu} o(\|\epsilon\|)$. We have shown that $\omega \mapsto h_\omega^\star$ is differentiable with a Jacobian map $\partial_\omega h_\omega^\star$ given by $J^\star = -B_\omega C_\omega^{-1}$. □

### C.3 Proof of the functional adjoint sensitivity in Proposition 2.2

*Proof of Proposition 2.2.* We use the assumptions and definitions from Proposition 2.2 and express the gradient $\nabla \mathcal{F}(\omega)$ using the chain rule:

$$\nabla \mathcal{F}(\omega) = \partial_\omega L_{out}(\omega, h_\omega^\star) + [\partial_\omega h_\omega^\star] \partial_h L_{out}(\omega, h_\omega^\star).$$

The Jacobian $\partial_\omega h_\omega^\star$ is the solution of a linear system obtained by applying Theorem 2.1 :

$$\partial_\omega h_\omega^\star = -B_\omega C_\omega^{-1}.$$

We note $g_\omega = \partial_\omega L_{out}(\omega, h_\omega^\star)$ and $d_\omega = \partial_h L_{out}(\omega, h_\omega^\star)$. It follows that the gradient $\nabla \mathcal{F}(\omega)$ can be expressed as:

$$\nabla \mathcal{F}(\omega) = g_\omega + [\partial_\omega h_\omega^\star] d_\omega = g_\omega + B_\omega a_\omega^\star$$
$$a_\omega^\star := -C_\omega^{-1} d_\omega.$$

In other words, the implicit gradient $\nabla \mathcal{F}(\omega)$ can be expressed using the adjoint function $a_\omega^\star$, which is an element of $\mathcal{H}$ and can be defined as the solution of the following functional regression problem:

$$a_\omega^\star = \arg\min_{a \in \mathcal{H}} L_{adj}(\omega, a) := \tfrac{1}{2} \langle a, C_\omega a \rangle_{\mathcal{H}} + \langle a, d_\omega \rangle_{\mathcal{H}}.$$

□

## D  Functional Adjoint Sensitivity Results in $L_2$ Spaces

In this section we provides full proofs of Proposition 2.3. We start by stating the assumptions needed on the data distributions and the point-wise losses in Appendix D.1, then provide some differentiation results in Appendix D.2 and conclude with the main proofs in Appendix D.3.

### D.1 Assumptions

**Assumption on $\mathbb{P}$ and $\mathbb{Q}$.**

- **(A)** $\mathbb{P}$ and $\mathbb{Q}$ admit finite second moments.
- **(B)** The marginal of $X$ w.r.t. $\mathbb{Q}$ admits a Radon-Nikodym derivative $r(x)$ w.r.t. the marginal of $X$ w.r.t. $\mathbb{P}$, i.e. $\mathrm{d}\mathbb{Q}(x, \mathcal{Y}) = r(x)\,\mathrm{d}\mathbb{P}(x, \mathcal{Y})$. Additionally, $r(x)$ is upper-bounded by a positive constant $M$.

**Assumptions on $\ell_{in}$.**

- **(C)** For any $\omega \in \Omega$, there exists a positive constant $\mu$ and a neighborhood $B$ of $\omega$ for which $\ell_{in}$ is $\mu$-strongly convex in its second argument for all $(\omega', x, y) \in B \times \mathcal{X} \times \mathcal{Y}$.
- **(D)** For any $\omega \in \Omega$, $\mathbb{E}_{\mathbb{P}}\left[ |\ell_{in}(\omega, 0, x, y)| + \|\partial_v \ell_{in}(\omega, 0, x, y)\|^2 \right] < +\infty$.
- **(E)** $v \mapsto \ell_{in}(\omega, v, x, y)$ is continuously differentiable for all $(\omega, x, y) \in \Omega \times \mathcal{X} \times \mathcal{Y}$.
- **(F)** For any fixed $\omega \in \Omega$, there exists a constant $L$ and a neighborhood $B$ of $\omega$ s.t. $v \mapsto \ell_{in}(\omega', v, x, y)$ is $L$-smooth for all $\omega', x, y \in B \times \mathcal{X} \times \mathcal{Y}$.
- **(G)** $(\omega, v) \mapsto \partial_v \ell_{in}(\omega, v, x, y)$ is continuously differentiable on $\Omega \times \mathcal{V}$ for all $x, y \in \mathcal{X} \times \mathcal{Y}$,
- **(H)** For any $\omega \in \Omega$, there exists a positive constant $C$ and a neighborhood $B$ of $\omega$ s.t. for all $(\omega', x, y) \in B \times \mathcal{X} \times \mathcal{Y}$:

$$\|\partial_{\omega, v} \ell_{in}(\omega', 0, x, y)\| \leq C(1 + \|x\| + \|y\|). \tag{25}$$

- **(I)** For any $\omega \in \Omega$, there exists a positive constant $C$ and a neighborhood $B$ of $\omega$ s.t. for all $(\omega', v_1, v_2, x, y) \in B \times \mathcal{V} \times \mathcal{V} \times \mathcal{X} \times \mathcal{Y}$ we have:

$$\|\partial_{\omega, v} \ell_{in}(\omega', v_1, x, y) - \partial_{\omega, v} \ell_{in}(\omega', v_2, x, y)\| \leq C\|v_1 - v_2\|. \tag{26}$$

**Assumptions on $\ell_{out}$.**

- **(J)** For any $\omega \in \Omega$, $\mathbb{E}_{\mathbb{Q}}\left[ |\ell_{out}(\omega, 0, x, y)| \right] < +\infty$.
- **(K)** $(\omega, v) \mapsto \ell_{out}(\omega, v, x, y)$ is jointly continuously differentiable on $\Omega \times \mathcal{V}$ for all $(x, y) \in \mathcal{X} \times \mathcal{Y}$.
- **(L)** For any $\omega \in \Omega$, there exits a neighborhood $B$ of $\omega$ and a positive constant $C$ s.t. for all $(\omega', v, v', x, y) \in B \times \mathcal{V} \times \mathcal{V} \times \mathcal{X} \times \mathcal{Y}$ we have:

$$\|\partial_\omega \ell_{out}(\omega', v, x, y) - \partial_\omega \ell_{out}(\omega', v', x, y)\| \leq C(1 + \|v\| + \|v'\| + \|x\| + \|y\|)\|v - v'\|,$$
$$\|\partial_v \ell_{out}(\omega', v, x, y) - \partial_v \ell_{out}(\omega', v', x, y)\| \leq C\|v - v'\|,$$
$$\|\partial_v \ell_{out}(\omega', v, x, y)\| \leq C(1 + \|v\| + \|x\| + \|y\|),$$
$$\|\partial_\omega \ell_{out}(\omega', v, x, y)\| \leq C\left(1 + \|v\|^2 + \|x\|^2 + \|y\|^2\right).$$

**Example.** Here we consider the squared error between two vectors $v, z$ in $\mathcal{V}$. Given a map $(\omega, x, y) \mapsto f_\omega(x, y)$ defined over $\Omega \times \mathcal{X} \times \mathcal{Y}$ and taking values in $\mathcal{V}$, we define the following point-wise objective:

$$\ell(\omega, v, x, y) := \frac{1}{2}\|v - z\|^2, \quad z = f_\omega(x, y). \tag{27}$$

We assume that for any $\omega \in \Omega$, there exists a constant $C > 0$ such that for all $\omega'$ in a neighborhood of $\omega$ and all $x, y \in \mathcal{X} \times \mathcal{Y}$, the following growth assumption holds:

$$\|f_{\omega'}(x, y)\| + \|\partial_\omega f_{\omega'}(x, y)\| \leq C(1 + \|x\| + \|y\|). \tag{28}$$

This growth assumption is weak in the context of neural networks with smooth activations as discussed by Bińkowski et al. [2018, Appendix C.4].

**Proposition D.1.** *Assume that the map $\omega \mapsto f_\omega(x, y)$ is continuously differentiable for any $x, y \in \mathcal{X} \times \mathcal{Y}$, and that Equation (28) holds. Additionally, assume that $\mathbb{P}$ and $\mathbb{Q}$ admit finite second order moments. Then the point-wise objective $\ell$ in Equation (27) satisfies Assumptions (C) to (L).*

*Proof.* We show that each of the assumptions are satisfied by the classical squared error objective.

- Assumption (**C**): the squared error is 1-strongly convex in $v$, since $\partial_v^2 \ell \succeq I$. Hence, the strong convexity assumption holds with $\mu = 1$.

- Assumption (**D**): For any $\omega \in \Omega$, we have

$$\mathbb{E}_\mathbb{P}\left[|\ell(\omega, 0, x, y)| + \|\partial_v \ell(\omega, 0, x, y)\|^2\right] = \mathbb{E}_\mathbb{P}\left[\frac{1}{2}\|f_\omega(x, y)\|^2 + \|f_\omega(x, y)\|^2\right] < +\infty,$$

which holds by the growth assumption on $f_\omega(x, y)$, and $\mathbb{P}$ having finite second moments.

- Assumption (**E**): With a perturbation $u \in \mathcal{V}$ we have:

$$\ell(\omega, v + u, x, y) = \frac{1}{2}\|v - z\|^2 + \langle v - z, u \rangle + o(\|u\|^2), \quad z = f_\omega(x, y),$$

with $o(\|u\|^2) = \frac{1}{2}\|u\|^2$. The mapping $v \mapsto v - z$ is continuous, thus the assumption holds.

- Assumption (**F**): For any two points $v_1, v_2 \in \mathcal{V}$ using the expression of $\partial_v \ell(\omega, v, x, y) = v - z$ with $z = f_\omega(x, y)$ we have:

$$\|\partial_v \ell(\omega, v_1, x, y) - \partial_v \ell(\omega, v_2, x, y)\| = \|(v_1 - z) - (v_2 - z)\| = \|v_1 - v_2\|,$$

We see that $\ell$ is $L$-smooth with $L = 1$ and the assumption holds.

- Assumption (**K**): By the differentiation assumption on $f_\omega(x, y)$, with a perturbation $\epsilon \in \Omega$ we can write:

$$f_{\omega+\epsilon}(x, y) = f_\omega(x, y) + \partial_\omega f_\omega(x, y)\epsilon + o(\epsilon).$$

With a perturbation $\epsilon \times u \in \Omega \times \mathcal{V}$ and substituting $f_{\omega+\epsilon}(x, y)$ with the expression above we have:

$$\begin{aligned}
\ell(\omega + \epsilon, v + u, x, y) =& \frac{1}{2}\|(v + u) - (f_\omega(x, y) + \partial_\omega f_\omega(x, y)\epsilon + o(\epsilon))\|^2 \\
=& \frac{1}{2}\|v - f_\omega(x, y)\|^2 + \langle \epsilon, \partial_\omega f_\omega(x, y)^\top (f_\omega(x, y) - v)\rangle \\
& + \langle u, v - f_\omega(x, y)\rangle + o(\|\epsilon\| + \|u\|),
\end{aligned}$$

which allows us to conclude that $(\omega, v) \mapsto \ell(\omega, v, x, y)$ is continuously differentiable on $\Omega \times \mathcal{V}$ for all $x, y \in \mathcal{X} \times \mathcal{Y}$ and the assumption holds.

- Assumption (**G**): With a perturbation $\epsilon \times u \in \Omega \times \mathcal{V}$ using the expression of $\partial_v \ell(\omega, v, x, y)$ we can write:

$$\begin{aligned}
\partial_v \ell(\omega + \epsilon, v + u, x, y) &= (v + u) - f_{\omega+\epsilon}(x, y) \\
&= (v + u) - (f_\omega(x, y) + \partial_\omega f_\omega(x, y)\epsilon + o(\epsilon)) \\
&= (v - f_\omega(x, y)) + u - \partial_\omega f_\omega(x, y)\epsilon + o(\epsilon),
\end{aligned}$$

by continuously differentiable $f_\omega(x, y)$, we have that the assumption holds.

- Assumptions (**H**) and (**I**): From the expression of $\partial_v \ell(\omega, v, x, y)$:

$$\partial_{\omega, v} \ell(\omega, v, x, y) = \partial_\omega (v - f_\omega(x, y)) = \partial_\omega f_\omega(x, y),$$

then using the expression above and the growth assumption on $f_{\omega'}(x, y)$ we have that the two assumptions hold.

- Assumption (**J**): For any $\omega \in \Omega$ we have:

$$\mathbb{E}_\mathbb{Q}\left[|\ell(\omega, 0, x, y)|\right] = \mathbb{E}_\mathbb{Q}\left[\frac{1}{2}\|f_\omega(x, y)\|^2\right] < +\infty,$$

by the growth assumption on $f_\omega(x, y)$, and $\mathbb{P}$ having finite second moments, thus the assumption is verified.

- Assumption (**L**): Using the growth assumption on $f_{\omega'}(x, y)$, we have the following inequalities:

$$\|\partial_\omega \ell\left(\omega', v, x, y\right) - \partial_\omega \ell\left(\omega', v', x, y\right)\| = \left\|f_{\omega'}(x, y)^\top (v' - v)\right\|$$

$$\leq \left\|f_{\omega'}(x, y)^\top\right\| + \|v' - v\|$$

$$\leq C\left(1 + \|v\| + \|v'\| + \|x\| + \|y\|\right)\|v - v'\|,$$

$$\|\partial_v \ell\left(\omega', v, x, y\right)\| = \|v - f_{\omega'}(x, y)\|$$

$$\leq \|v\| + \|f_{\omega'}(x, y)\|$$

$$\leq \|v\| + C\left(1 + \|x\| + \|y\|\right)$$

$$\leq C\left(1 + \|v\| + \|x\| + \|y\|\right)$$

$$\|\partial_\omega \ell\left(\omega', v, x, y\right)\| = \left\|\partial_\omega f_{\omega'}(x, y)^\top \left(f_{\omega'}(x, y) - v\right)\right\|$$

$$\leq \|\partial_\omega f_{\omega'}(x, y)\| \|(f_{\omega'}(x, y) - v)\|$$

$$\leq \|\partial_\omega f_{\omega'}(x, y)\| \left(\|f_{\omega'}(x, y)\| + \|v\|\right)$$

$$\leq C\left(1 + \|v\|^2 + \|x\|^2 + \|y\|^2\right),$$

combining the above with $L$-smoothness of $\ell$ we can conclude that the assumption holds.

$\square$

## D.2   Differentiability results

The next lemmas show differentiability of $L_{out}$, $L_{in}$ and $\partial_h L_{in}$ and will be used to prove Proposition 2.3.

**Lemma D.2** (Differentiability of $L_{in}$ in its second argument). *Under Assumptions (**D**) to (**F**), the function $h \mapsto L_{in}(\omega, h)$ is differentiable in $\mathcal{H}$ with partial derivative vector $\partial_h L_{in}(\omega, h) \in \mathcal{H}$ given by:*

$$\partial_h L_{in}(\omega, h) : \mathcal{X} \to \mathcal{V}$$
$$x \mapsto \mathbb{E}_{\mathbb{P}}\left[\partial_v \ell_{in}\left(\omega, h(x), x, y\right) | x\right].$$

*Proof.* We decompose the proof into three parts: verifying that $L_{in}$ is well-defined, identifying a bounded map as candidate for the differential and showing that it is the Fréchet differential of $L_{in}$.

**Well-defined objective.** Consider $(\omega, h)$ in $\Omega \times \mathcal{H}$. To show that $L_{in}(\omega, h)$ is well-defined, we need to prove that $\ell_{in}(\omega, h(x), x, y)$ is integrable under $\mathbb{P}$. We use the following inequalities to control $\ell_{in}(\omega, h(x), x, y)$:

$$|\ell_{in}(\omega, h(x), x, y)| \leq |\ell_{in}(\omega, h(x), x, y) - \ell_{in}(\omega, 0, x, y)| + |\ell_{in}(\omega, 0, x, y)|$$

$$= \left|\int_0^1 \mathrm{d}t\left(h(x)^\top \partial_v \ell_{in}(\omega, th(x), x, y)\right)\right| + |\ell_{in}(\omega, 0, x, y)|$$

$$\leq \|h(x)\| \int_0^1 \mathrm{d}t \|\partial_v \ell_{in}(\omega, th(x), x, y) - \partial_v \ell_{in}(\omega, 0, x, y)\|$$

$$+ \|h(x)\| \|\partial_v \ell_{in}(\omega, 0, x, y)\| + |\ell_{in}(\omega, 0, x, y)|$$

$$\leq \frac{L}{2}\|h(x)\|^2 + \frac{1}{2}\left(\|h(x)\|^2 + \|\partial_v \ell_{in}(\omega, 0, x, y)\|^2\right) + |\ell_{in}(\omega, 0, x, y)|,$$

where the first line follows by triangular inequality, the second follows by application of the fundamental theorem of calculus since $\ell_{in}$ is differentiable by Assumption (**E**). The third uses Cauchy-Schwarz inequality along with a triangular inequality. Finally, the last line follows using that $\ell_{in}$ is $L$-smooth in its second argument, locally in $\omega$ and uniformly in $x$ and $y$ by Assumption (**F**). Taking the expectation under $\mathbb{P}$ yields:

$$|L_{in}(\omega, h)| \leq \mathbb{E}_{\mathbb{P}}\left[|\ell_{in}\left(\omega, h(x), x, y\right)|\right]$$

$$\leq \frac{L+1}{2}\|h\|_{\mathcal{H}}^2 + \mathbb{E}_{\mathbb{P}}\left[\|\partial_v \ell_{in}(\omega, 0, x, y)\|^2 + |\ell_{in}(\omega, 0, x, y)|\right] < +\infty,$$

where $\|h\|_{\mathcal{H}}$ is finite since $h \in \mathcal{H}$ and expectations under $\mathbb{P}$ of $\|\partial_v \ell_{in}(\omega, 0, x, y)\|^2$ and $|\ell_{in}(\omega, 0, x, y)|$ are finite by Assumption (D). This shows that $L_{in}(\omega, h)$ is well defined on $\Omega \times \mathcal{H}$.

**Candidate differential.** Fix $(\omega, h)$ in $\Omega \times \mathcal{H}$ and consider the following linear form $d_{in}$ in $\mathcal{H}$:

$$d_{in}g := \mathbb{E}_{\mathbb{P}}\left[g(x)^\top \partial_v \ell_{in}(\omega, h(x), x, y)\right], \qquad \forall g \in \mathcal{H}.$$

We need to show that it is a bounded form. To this end, we will show that $d_{in}$ is a scalar product with some vector $D_{in}$ in $\mathcal{H}$. The following equalities hold:

$$\begin{aligned}
d_{in}g &= \mathbb{E}_{\mathbb{P}}\left[g(x)^\top \partial_v \ell_{in}(\omega, h(x), x, y)\right] \\
&= \mathbb{E}_{\mathbb{P}}\left[g(x)^\top \mathbb{E}_{\mathbb{P}}\left[\partial_v \ell_{in}(\omega, h(x), x, y) | x\right]\right] \\
&= \mathbb{E}_{\mathbb{P}}\left[g(x)^\top D_{in}(x)\right],
\end{aligned}$$

where the second line follows by the "tower" property for conditional expectations and where we define $D_{in}(x) := \mathbb{E}_{\mathbb{P}}\left[\partial_v \ell_{in}(\omega, h(x), x, y) | x\right]$ in the last line. $D_{in}$ is a the candidate representation of $d_{in}$ in $\mathcal{H}$. We simply need to check that $D_{in}$ is an element of $\mathcal{H}$. To see this, we use the following upper-bounds:

$$\begin{aligned}
\mathbb{E}_{\mathbb{P}}\left[\|D_{in}(x)\|^2\right] &\leq \mathbb{E}_{\mathbb{P}}\left[\mathbb{E}_{\mathbb{P}}\left[\|\partial_v \ell_{in}(\omega, h(x), x, y)\|^2 \Big| x\right]\right] \\
&= \mathbb{E}_{\mathbb{P}}\left[\|\partial_v \ell_{in}(\omega, h(x), x, y)\|^2\right] \\
&\leq 2\mathbb{E}_{\mathbb{P}}\left[\|\partial_v \ell_{in}(\omega, h(x), x, y) - \partial_v \ell_{in}(\omega, 0, x, y)\|^2\right] + 2\mathbb{E}_{\mathbb{P}}\left[\|\partial_v \ell_{in}(\omega, 0, x, y)\|^2\right] \\
&\leq 2L^2 \mathbb{E}_{\mathbb{P}}\left[\|h(x)\|^2\right] + 2\mathbb{E}_{\mathbb{P}}\left[\|\partial_v \ell_{in}(\omega, 0, x, y)\|^2\right] < +\infty.
\end{aligned}$$

The first inequality is an application of Jensen's inequality by convexity of the squared norm. The second line follows by the "tower" property for conditional probability distributions while the third follows by triangular inequality and Jensen's inequality applied to the square function. The last line uses that $\ell_{in}$ is $L$-smooth in its second argument, locally in $\omega$ and uniformly in $x, y$ by Assumption (F). Since $h$ is square integrable under $\mathbb{P}$ by construction and $\|\partial_v \ell_{in}(\omega, 0, x, y)\|$ is also square integrable by Assumption (D), we deduce from the above upper-bounds that $D_{in}(x)$ must also be square integrable and thus an element of $\mathcal{H}$. Therefore, we have shown that $d_{in}$ is a continuous linear form admitting the following representation:

$$d_{in}g = \langle D_{in}, g \rangle_{\mathcal{H}}. \tag{29}$$

**Differentiability of $h \mapsto L_{in}(\omega, h)$.** To prove differentiability, we simply control the first order error $E(g)$ defined as:

$$E(g) := |L_{in}(\omega, h + g) - L_{in}(\omega, h) - d_{in}g|. \tag{30}$$

For a given $g \in \mathcal{H}$, the following inequalities hold:

$$\begin{aligned}
E(g) &= \left|\mathbb{E}_{\mathbb{P}}\left[\int_0^1 \mathrm{d}t \left(g(x)^\top \left(\partial_v \ell_{in}(\omega, h(x) + tg(x), x, y) - \partial_v \ell_{in}(\omega, h(x), x, y)\right)\right)\right]\right| \\
&\leq \mathbb{E}_{\mathbb{P}}\left[\int_0^1 |g(x)^\top \left(\partial_v \ell_{in}(\omega, h(x) + tg(x), x, y) - \partial_v \ell_{in}(\omega, h(x), x, y)\right)| \, \mathrm{d}t\right] \\
&\leq \frac{L}{2}\mathbb{E}_{\mathbb{P}}\left[\|g(x)\|^2\right] = \frac{L}{2}\|g\|_{\mathcal{H}}^2,
\end{aligned}$$

where the first inequality follows by application of the fundamental theorem of calculus since $\ell_{in}$ is differentiable in its second argument by Assumption (E). The second line follows by Jensen's inequality while the last line uses that $v \mapsto \partial \ell_{in}(\omega, v, x, y)$ is $L$-Lipschitz locally in $\omega$ and uniformly in $x$ and $y$ by Assumption (F). Therefore, we have shown that $E(g) = o(\|g\|_{\mathcal{H}})$ which precisely means that $h \mapsto L_{in}(\omega, h)$ is differentiable with differential $d_{in}$. Moreover, $D_{in}$ is the partial gradient of $L_{in}(\omega, h)$ in the second variable:

$$\partial_h L_{in}(\omega, h) = D_{in} = x \mapsto \mathbb{E}_{\mathbb{P}}\left[\partial_v \ell_{in}(\omega, h(x), x, y) | x\right].$$

$\square$

**Lemma D.3** (Differentiability of $L_{out}$). *Under Assumptions (A), (B) and (J) to (L), $L_{out}$ is jointly differentiable in $\omega$ and $h$. Moreover, its partial derivatives $\partial_\omega L_{out}(\omega, h)$ and $\partial_h L_{out}(\omega, h)$ are elements in $\Omega$ and $\mathcal{H}$ given by:*

$$\partial_\omega L_{out}(\omega, h) = \mathbb{E}_\mathbb{Q}\left[\partial_\omega \ell_{out}(\omega, h(x), x, y)\right]$$
$$\partial_h L_{out}(\omega, h) = x \mapsto r(x)\mathbb{E}_\mathbb{Q}\left[\partial_v \ell_{out}(\omega, h(x), x, y)|x\right]. \tag{31}$$

*Proof.* We follow a similar procedure as in Lemma D.2, where we decompose the proof into three steps: verifying that the objective $L_{out}$ is well-defined, identifying a candidate for the differential and proving that it is the differential of $L_{out}$.

**Well-definiteness of the objective.** Let $(\omega, h)$ be in $\Omega \times \mathcal{H}$. First, note that by Assumption (B), we have that

$$\mathbb{E}_\mathbb{Q}\left[\|h(x)\|^2\right] = \mathbb{E}_\mathbb{P}\left[\|h(x)\|^2 r(x)\right] \leq M \|h\|_\mathcal{H}^2 < +\infty. \tag{32}$$

The next inequalities control the growth of $\ell_{out}$:

$$|\ell_{out}(\omega, h(x), x, y)| \leq |\ell_{out}(\omega, 0, x, y)| + |\ell_{out}(\omega, h(x), x, y) - \ell_{out}(\omega, 0, x, y)|$$

$$\leq |\ell_{out}(\omega, 0, x, y)| + \int_0^1 dt \left|h(x)^\top \partial_v \ell_{out}(\omega, th(x), x, y)\right|$$

$$\leq |\ell_{out}(\omega, 0, x, y)| + \|h(x)\| \int_0^1 dt \|\partial_v \ell_{out}(\omega, th(x), x, y)\|$$

$$\leq |\ell_{out}(\omega, 0, x, y)| + C \|h(x)\| (1 + \|h(x)\| + \|x\| + \|y\|)$$

$$\leq |\ell_{out}(\omega, 0, x, y)| + C \left(1 + 3\|h(x)\|^2 + \|x\|^2 + \|y\|^2\right).$$

The first line is due to the triangular inequality while the second line follows by differentiability of $\partial_v \ell_{out}$ in its second argument (Assumption (K)). The third line follows by Cauchy-Scwharz inequality wile the fourth line uses that $\ell_{out}$ has at most a linear growth in its last three arguments by Assumption (L). Using the above inequalities, we get the following upper-bound on $L_{out}$:

$$|L_{out}(\omega, h)| \leq \mathbb{E}_\mathbb{Q}\left[|\ell_{out}(\omega, 0, x, y)|\right] + C \left(1 + 3\mathbb{E}_\mathbb{Q}\left[\|h(x)\|^2\right] + \mathbb{E}_\mathbb{Q}\left[\|x\|^2 + \|y\|^2\right]\right) < +\infty. \tag{33}$$

In the above upper-bound, $\mathbb{E}_\mathbb{Q}\left[\|h(x)\|^2\right]$ is finite by Equation (32). Additionally, $\mathbb{E}_\mathbb{Q}\left[\|x\|^2 + \|y\|^2\right]$ is finite since $\mathbb{Q}$ has finite second moments by Assumption (A) while $\mathbb{E}_\mathbb{Q}\left[|\ell_{out}(\omega, 0, x, y)|\right]$ is also finite by Assumption (J). Therefore, $L_{out}$ is well defined over $\Omega \times \mathcal{H}$.

**Candidate differential.** Fix $(\omega, h)$ in $\Omega \times \mathcal{H}$ and define the following linear form:

$$d_{out}(\epsilon, g) := \epsilon^\top \mathbb{E}_\mathbb{Q}\left[\partial_\omega \ell_{out}(\omega, h(x), x, y)\right] + \mathbb{E}_\mathbb{Q}\left[g(x)^\top \partial_v \ell_{out}(\omega, h(x), x, y)\right]$$

Define $D_{out} = (D_\omega, D_h)$ to be:

$$D_\omega := \mathbb{E}_\mathbb{Q}\left[\partial_\omega \ell_{out}(\omega, h(x), x, y)\right]$$
$$D_h := x \mapsto r(x)\mathbb{E}_\mathbb{Q}\left[\partial_v \ell_{out}(\omega, h(x), x, y)|x\right].$$

By an argument similar to the one in Lemma D.2, we see that $d_{out}(\epsilon, g) = \langle g, D_h \rangle_\mathcal{H} + \epsilon^\top D_\omega$. We now need to show that $D_\omega$ and $D_h$ are well defined elements of $\Omega$ and $\mathcal{H}$.

**Square integrability of $D_h$.** We use the following upper-bounds:

$$\mathbb{E}_\mathbb{P}\left[\|D_h(x)\|^2\right] \leq \mathbb{E}_\mathbb{P}\left[r(x)^2 \mathbb{E}_\mathbb{Q}\left[\|\partial_v \ell_{out}(\omega, h(x), x, y)\| \, |x]^2\right]\right]$$

$$\leq \mathbb{E}_\mathbb{P}\left[r(x)^2 \mathbb{E}_\mathbb{Q}\left[\|\partial_v \ell_{out}(\omega, h(x), x, y)\|^2 \Big| x\right]\right]$$

$$\leq M\mathbb{E}_\mathbb{P}\left[r(x)\mathbb{E}_\mathbb{Q}\left[\|\partial_v \ell_{out}(\omega, h(x), x, y)\|^2 \Big| x\right]\right]$$

$$= M\mathbb{E}_\mathbb{Q}\left[\|\partial_v \ell_{out}(\omega, h(x), x, y)\|^2\right]$$

$$\leq 4MC \left(1 + \mathbb{E}_\mathbb{Q}\left[\|h(x)\|^2\right] + \mathbb{E}_\mathbb{Q}\left[\|x\|^2 + \|y\|^2\right]\right).$$

The first inequality is an application of Jensen's inequality by convexity of the norm, while the second one is an application of Cauchy-Schwarz inequality. The third line uses that $r(x)$ is upper-bounded by a constant $M$ by Assumption (**B**), and the fourth line follows from the "tower" property for conditional probability distributions. Finally, the last line follows by Assumption (**L**) which ensures that $\partial_v \ell_{out}$ has at most a linear growth in its last three arguments. By Equation (32), we have that $\mathbb{E}_{\mathbb{Q}}\left[\|h(x)\|^2\right] < +\infty$. Moreover, since $\mathbb{Q}$ has finite second order moment by Assumption (**A**), we also have that $\mathbb{E}_{\mathbb{Q}}\left[\|x\|^2 + \|y\|^2\right] < +\infty$. We therefore conclude that $\mathbb{E}_{\mathbb{P}}\left[\|D_h(x)\|^2\right]$ is finite which ensure that $D_h$ belongs to $\mathcal{H}$.

**Well-definiteness of $D_\omega$.** To show that $D_\omega$ is well defined, we need to prove that $(x, y) \mapsto \partial_\omega \ell_{out}(\omega, h(x), x, y)$ is integrable under $\mathbb{Q}$. By Assumption (**L**), we know that $\partial_\omega \ell_{out}$ has at most a quadratic growth in it last three arguments so that the following inequality holds.

$$\|\partial_\omega \ell_{out}(\omega, h(x), x, y)\| \leq C \left\|1 + \|h(x)\|^2 + \|x\|^2 + \|y\|^2\right\|.$$

We can directly conclude by taking the expectation under $\mathbb{Q}$ in the above inequality and recalling that $\mathbb{E}_{\mathbb{Q}}\left[\|h(x)\|^2\right]$ is finite by Equation (32), and that $\mathbb{Q}$ has finite second-order moments by Assumption (**A**).

**Differentiability of $L_{out}$.** Since differentiability is a local notion, we may assume without loss of generality that $\|\epsilon\|^2 + \|g\|_{\mathcal{H}}^2 \leq 1$. Introduce the functions $\Delta_1$ and $\Delta_2$ defined over $\Omega \times \mathcal{H}, \mathcal{X} \times \mathcal{Y} \times [0,1]$ as follows:

$$\Delta_1(\epsilon, g, x, y, t) := \partial_v \ell_{out}(\omega + t\epsilon, h(x) + tg(x), x, y) - \partial_v \ell_{out}(\omega + t\epsilon, h(x), x, y)$$
$$\Delta_1'(\epsilon, g, x, y, t) := \partial_v \ell_{out}(\omega + t\epsilon, h(x), x, y) - \partial_v \ell_{out}(\omega, h(x), x, y)$$
$$\Delta_2(\epsilon, g, x, y, t) := \partial_\omega \ell_{out}(\omega + t\epsilon, h(x) + tg(x), x, y) - \partial_\omega \ell_{out}(\omega + t\epsilon, h(x), x, y)$$
$$\Delta_2'(\epsilon, g, x, y, t) := \partial_\omega \ell_{out}(\omega + t\epsilon, h(x), x, y) - \partial_\omega \ell_{out}(\omega, h(x), x, y).$$

We consider the first-order error $E(\epsilon, g)$ which admits the following upper-bounds:

$$E(\epsilon, g) := |L_{out}(\omega + \epsilon, h + g) - L_{out}(\omega, h) - d_{out}(\epsilon, g)|$$
$$= \left|\mathbb{E}_{\mathbb{Q}}\left[\int_0^1 dt \left(g(x)^\top (\Delta_1 + \Delta_1')(\epsilon, g, x, y, t) + \epsilon^\top (\Delta_2 + \Delta_2')(\epsilon, g, x, y, t)\right)\right]\right|$$
$$\leq \mathbb{E}_{\mathbb{Q}}\left[\int_0^1 dt \, \|g(x)\| \left(\|\Delta_1(\epsilon, g, x, y, t)\| + \|\Delta_1'(\epsilon, g, x, y, t)\|\right)\right]$$
$$+ \mathbb{E}_{\mathbb{Q}}\left[\int_0^1 dt \, \|\epsilon\| \left(\|\Delta_2(\epsilon, g, x, y, t)\| + \|\Delta_2'(\epsilon, g, x, y, t)\|\right)\right]$$
$$\leq \mathbb{E}_{\mathbb{Q}}\left[\|g(x)\|^2\right]^{\frac{1}{2}} \left(\underbrace{\mathbb{E}_{\mathbb{Q}}\left[\int_0^1 dt \, \|\Delta_1(\epsilon, g, x, y, t)\|^2\right]^{\frac{1}{2}}}_{A_1(\epsilon, g)} + \underbrace{\mathbb{E}_{\mathbb{Q}}\left[\int_0^1 dt \, \|\Delta_1'(\epsilon, g, x, y, t)\|^2\right]^{\frac{1}{2}}}_{A_2(\epsilon, g)}\right)$$
$$+ \|\epsilon\| \left(\underbrace{\mathbb{E}_{\mathbb{Q}}\left[\int_0^1 dt \, \|\Delta_2(\epsilon, g, x, y, t)\|\right]}_{A_3(\epsilon, g)} + \underbrace{\mathbb{E}_{\mathbb{Q}}\left[\int_0^1 dt \, \|\Delta_2'(\epsilon, g, x, y, t)\|\right]}_{A_4(\epsilon, g)}\right)$$
$$\leq M \|g\|_{\mathcal{H}} (A_1(\epsilon, g) + A_2(\epsilon, g)) + \|\epsilon\| (A_3(\epsilon, g) + A_4(\epsilon, g)).$$

The second line uses differentiability of $\ell_{out}$ (Assumption (**K**)). The third uses the triangular inequality, while the fourth line uses Cauchy-Schwarz inequality. Finally, the last line uses Equation (32).

We simply need to show that each of the terms $A_1$, $A_2$, $A_3$ and $A_4$ converge to 0 as $\epsilon$ and $g$ converge to 0. We treat each term separately.

**Controlling $A_1$ and $A_3$.** For $\epsilon$ small enough so that Assumption (**L**) holds, the following upper-bounds on $A_1$ and $A_2$ hold:

$$
\begin{aligned}
A_1(\epsilon, g) \leq& C\mathbb{E}_{\mathbb{Q}}\left[\|g(x)\|^2\right]^{\frac{1}{2}} \\
\leq& CM^{\frac{1}{2}}\|g\|_{\mathcal{H}} \\
A_3(\epsilon, g) \leq& C\mathbb{E}_{\mathbb{Q}}\left[(1 + \|h(x) + tg(x)\| + \|h(x)\| + \|x\| + \|y\|)\|g(x)\|\right] \\
\leq& C\mathbb{E}_{\mathbb{Q}}\left[\|g(x)\|^2\right]^{\frac{1}{2}}\left(1 + 2\mathbb{E}_{\mathbb{Q}}\left[\|h(x)\|^2\right]^{\frac{1}{2}} + \mathbb{E}_{\mathbb{Q}}\left[\|g(x)\|^2\right]^{\frac{1}{2}} + \mathbb{E}_{\mathbb{Q}}\left[\|x\|^2 + \|y\|^2\right]^{\frac{1}{2}}\right) \\
\leq& CM^{\frac{1}{2}}\|g\|_{\mathcal{H}}\left(1 + M^{\frac{1}{2}} + 2\mathbb{E}_{\mathbb{Q}}\left[\|h(x)\|^2\right]^{\frac{1}{2}} + \mathbb{E}_{\mathbb{Q}}\left[\|x\|^2 + \|y\|^2\right]^{\frac{1}{2}}\right).
\end{aligned}
$$

For $A_1$, we used that $\partial_v \ell_o ut$ has is Lipschitz continuous in its second argument for any $x, y \in \mathcal{X} \times \mathcal{Y}$ and locally in $\omega$ by Assumption (**L**). The second upper-bound on $A_1$ uses Equation (32). For $A_2$, we used the locally Lipschitz property of $\partial_\omega \ell_{out}$ from Assumption (**L**), followed by Cauchy-Schwarz inequality and Equation (32). For the last line, we also used that $\|g\|_{\mathcal{H}} \leq 1$ by assumption. The above upper-bounds on $A_1$ and $A_3$ ensure that these quantities converge to 0 as $\epsilon$ and $g$ approach 0.

**Controlling $A_2$ and $A_4$.** To show that $A_2$ and $A_4$ converge to 0, we will use the dominated convergence theorem. It is easy to see that $\Delta_1'(\epsilon, g, x, y, t)$ and $\Delta_2'(\epsilon, g, x, y, t)$ converge point-wise to 0 when $\epsilon$ and $g$ converge to 0 since $(\omega, v) \mapsto \partial_v \ell_{out}(\omega, v, x, y)$ and $(\omega, v) \mapsto \partial_\omega \ell_{out}(\omega, v, x, y)$ are continuous by Assumption (**K**). It remains to dominate these functions. For $\epsilon$ small enough so that Assumption (**L**) holds, we have that:

$$
\begin{aligned}
\Delta_1'(\epsilon, g, x, y, t)^2 \leq& 16C^2\left(1 + \|h(x)\|^2 + \|x\|^2 + \|y\|^2\right) \\
\Delta_2'(\epsilon, g, x, y, t) \leq& 2C\left(1 + \|h(x)\|^2 + \|x\|^2 + \|y\|^2\right).
\end{aligned}
$$

Both upper-bounds are integrable under $\mathbb{Q}$ since $\mathbb{E}_{\mathbb{Q}}\left[\|h(x)\|^2\right] < +\infty$ by Equation (32) and $\mathbb{Q}$ has finite second-order moment by Assumption (**A**). Therefore, by the dominated convergence theorem, we deduce that $A_2$ and $A_4$ converge to 0 as $\epsilon$ and $g$ approach 0.

Finally, we have shown that $E(\epsilon, g) = o\left(\|\epsilon\| + \|g\|_{\mathcal{H}}\right)$ which allows to conclude that $L_{out}$ is differentiable with the partial derivatives given by Equation (31). $\square$

**Lemma D.4 (Differentiability of $\partial_h L_{in}$).** *Under Assumptions (**A**) and (**E**) to (**I**), the differential map $(\omega, h) \mapsto \partial_h L_{in}(\omega, h)$ defined in Lemma D.2 is differentiable on $\Omega \times \mathcal{H}$ in the sense of Definition C.1. Its differential $d_{(\omega, h)}\partial_h L_{in}(\omega, h) : \Omega \times \mathcal{H} \to \mathcal{H}$ acts on elements $(\epsilon, g) \in \Omega \times \mathcal{H}$ as follows:*

$$
d_{(\omega, h)}\partial_h L_{in}(\omega, h)(\epsilon, g) = \partial_h^2 L_{in}(\omega, h)g + (\partial_{\omega, h}L_{in}(\omega, h))^\star \epsilon, \tag{34}
$$

*where $\partial_h^2 L_{in}(\omega, h) : \mathcal{H} \to \mathcal{H}$ is a linear symmetric operator representing the partial derivative of $\partial_h L_{in}(\omega, h)$ w.r.t $h$ and $(\partial_{\omega, h}L_{in}(\omega, h))^\star$ is the adjoint of $\partial_{\omega, h}L_{in}(\omega, h) : \mathcal{H} \to \Omega$ which represents the partial derivative of $\partial_h L_{in}(\omega, h)$ w.r.t $\omega$. Moreover, $\partial_h^2 L_{in}(\omega, h)$ and $\partial_{\omega, h}L_{in}(\omega, h)$ are given by:*

$$
\partial_h^2 L_{in}(\omega, h)g = x \mapsto \mathbb{E}_{\mathbb{P}}\left[\partial_v^2 \ell_{in}(\omega, h(x), x, y)\big|x\right]g(x) \tag{35}
$$

$$
\partial_{\omega, h}L_{in}(\omega, h)g = \mathbb{E}_{\mathbb{P}}\left[\partial_{\omega, v}\ell_{in}(\omega, h(x), x, y)g(x)\right], \tag{36}
$$

*Proof.* Let $(\omega, h)$ be in $\Omega \times \mathcal{H}$. To show that $\partial_\omega L_{in}$ is Hadamard differentiable, we proceed in two steps: we first identify a candidate differential and show that it is a bounded operator, then we prove Hadamard differentiability.

**Candidate differential.** For a given $(\omega, h) \in \Omega \times \mathcal{H}$, we consider the following linear operators $C_{w,h} : \mathcal{H} \to \mathcal{H}$ and $B_{w,h} : \mathcal{H} \to \Omega$:

$$
C_{\omega, h}g = \mathbb{E}_{\mathbb{P}}\left[\partial_v^2 \ell_{in}(\omega, h(x), x, y)\big|x\right]g(x), \qquad B_{\omega, h}g = \mathbb{E}_{\mathbb{P}}\left[\partial_{\omega, v}\ell_{in}(\omega, h(x), x, y)g(x)\right],
$$

where the expectations are over $y$ conditionally on $x$. Next, we show that $C_{\omega, h}$ and $B_{\omega, h}$ are well-defined and bounded.

**Well-definiteness of the operator** $C_{\omega,h}$**.** The first step is to show that the image $C_{\omega,h}g$ of any element $g \in \mathcal{H}$ by $C_{\omega,h}$ is also an element in $\mathcal{H}$. To this end, we simply need to find a finite upper-bound on $\|C_{\omega,h}g\|_{\mathcal{H}}$ for a given $g \in \mathcal{H}$:

$$
\begin{aligned}
\|C_{\omega,h}g\|_{\mathcal{H}}^2 &= \mathbb{E}_{\mathbb{P}}\left[ \left\| \mathbb{E}_{\mathbb{P}}\left[ \partial_v^2 \ell_{in}(\omega, h(x), x, y) \big| x \right] g(x) \right\|^2 \right] \\
&\leq \mathbb{E}_{\mathbb{P}}\left[ \left\| \mathbb{E}_{\mathbb{P}}\left[ \partial_v^2 \ell_{in}(\omega, h(x), x, y) \big| x \right] \right\|_{op}^2 \|g(x)\|^2 \right] \\
&\leq \mathbb{E}_{\mathbb{P}}\left[ \mathbb{E}_{\mathbb{P}}\left[ \left\| \partial_v^2 \ell_{in}(\omega, h(x), x, y) \right\|_{op} \big| x \right]^2 \|g(x)\|^2 \right] \\
&\leq \mathbb{E}_{\mathbb{P}}\left[ \left\| \partial_v^2 \ell_{in}(\omega, h(x), x, y) \right\|_{op}^2 \|g(x)\|^2 \right] \\
&\leq L^2 \|g\|_{\mathcal{H}}^2 .
\end{aligned}
$$

The second line follows using the operator norm inequality, the third line follows by Jensen's inequality applied to the norm, while the fourth uses the "tower" property for conditional distributions. Finally, the last line uses that $\partial_v^2 \ell_{in}$ is upper-bounded uniformly in $x$ and $y$ by Assumption **(F)**. Therefore, we conclude that $C_{\omega,h}g$ belongs to $\mathcal{H}$. Moreover, the inequality $\|C_{\omega,h}g\|_{\mathcal{H}} \leq L \|g\|_{\mathcal{H}}$ also establishes the continuity of the operator $C_{\omega,h}$.

**Well-definiteness of the operator** $B_{\omega,h}$**.** We first show that the image $B_{\omega,h}$ is bounded. For a given $g$ in $\mathcal{H}$, we write:

$$
\begin{aligned}
\|B_{\omega,h}g\| &= \left\| \mathbb{E}_{\mathbb{P}}\left[ \partial_{\omega,v}\ell_{in}(\omega, h(x), x, y)g(x) \right] \right\| \\
&\leq \mathbb{E}_{\mathbb{P}}\left[ \left\| \partial_{\omega,v}\ell_{in}(\omega, h(x), x, y)g(x) \right\| \right] \\
&\leq \mathbb{E}_{\mathbb{P}}\left[ \left\| \partial_{\omega,v}\ell_{in}(\omega, h(x), x, y) \right\|_{op} \|g(x)\| \right] \\
&\leq \|g\|_{\mathcal{H}} \, \mathbb{E}_{\mathbb{P}}\left[ \left\| \partial_{\omega,v}\ell_{in}(\omega, h(x), x, y) \right\|_{op}^2 \right]^{\frac{1}{2}} \\
&\leq \|g\|_{\mathcal{H}} \, \mathbb{E}_{\mathbb{P}}\left[ \left\| \partial_{\omega,v}\ell_{in}(\omega, h(x), x, y) - \partial_{\omega,v}\ell_{in}(\omega, 0, x, y) \right\|_{op}^2 \right]^{\frac{1}{2}} \\
&\quad + \|g\|_{\mathcal{H}} \, \mathbb{E}_{\mathbb{P}}\left[ \left\| \partial_{\omega,v}\ell_{in}(\omega, 0, x, y) \right\|_{op}^2 \right]^{\frac{1}{2}} \\
&\leq C \|g\|_{\mathcal{H}} \left( \mathbb{E}_{\mathbb{P}}\left[ \|h(x)\|^2 \right]^{\frac{1}{2}} + \mathbb{E}_{\mathbb{P}}\left[ (1 + \|x\| + \|y\|)^2 \right]^{\frac{1}{2}} \right) \\
&\leq C \|g\|_{\mathcal{H}} \left( \|h\|_{\mathcal{H}} + 2\mathbb{E}_{\mathbb{P}}\left[ 1 + \|x\|^2 + \|y\|^2 \right] \right) < +\infty.
\end{aligned}
$$

In the above expression, the second line is due to Jensen's inequality applied to the norm function, the third line follows from the operator norm inequality, while the fourth follows by Cauchy-Schwarz. The fifth line is due to the triangular inequality. Finally, the sixth line relies on two facts: 1) that $v \mapsto \partial_{\omega,v}\ell_{in}(\omega, v, x, y)$ is Lipschitz uniformly in $x$ and $y$ and locally in $\omega$ by Assumption **(I)**, and, 2) that $\|\partial_{\omega,v}\ell_{in}(\omega, 0, x, y)\|$ has at most a linear growth in $x$ and $y$ locally in $\omega$ by Assumption **(H)**. Since $\mathbb{P}$ has finite second order moments by Assumption **(A)** and both $h$ and $g$ are square integrable, we conclude that the constant $\|B_{\omega,h}\|$ is finite. Moreover, the last inequality establishes that $B_{\omega,h}$ is a continuous linear operator from $\mathcal{H}$ to $\Omega$. One can then see that the adjoint of $B_{\omega,h}$ admits a representation of the form:

$$
(B_{\omega,h})^{\star}\epsilon := (\partial_{\omega,h}L_{in}(\omega, h))^{\star}\epsilon = x \mapsto \mathbb{E}_{\mathbb{P}}\left[ (\partial_{\omega,v}\ell_{in}(\omega, h(x), x, y))^{\top} \big| x \right] \epsilon.
$$

Therefore, we can consider the following candidate operator $d_{in}^2$ for the differential of $\partial_h L_{in}$:

$$
d_{in}^2(\epsilon, g) := C_{\omega,h}g + (B_{\omega,h})^{\star}\epsilon.
$$

**Differentiablity of** $\partial_h L_{in}$**.** We will show that $\partial_h L_{in}$ is jointly Hadamard differentiable at $(\omega, h)$ with differential operator given by:

$$
d_{(\omega,h)}\partial_h L_{in}(\omega, h)(\epsilon, g) = C_{\omega,h}g + (B_{\omega,h})^{\star}\epsilon. \tag{37}
$$

To this end, we consider a sequence $(\epsilon_k, g_k)_{k \geq 1}$ converging in $\Omega \times \mathcal{H}$ towards an element $(\epsilon, g) \in \Omega \times \mathcal{H}$ and a non-vanishing real valued sequence $t_k$ converging to $0$. Define the first-order error $E_k$ as follows:

$$E_k := \left\| \frac{1}{t_k} \left( \partial_h L_{in}(\omega + t_k \epsilon_k, h + t_k g_k) - \partial_h L_{in}(\omega, h) \right) - C_{\omega,h} g - (B_{\omega,h})^\star \epsilon \right\|_{\mathcal{H}}^2.$$

Introduce the functions $P_1, P_2, \Delta_1$ and $\Delta_2$ defined over $\mathbb{N}^\star, \mathcal{X} \times \mathcal{Y} \times [0,1]$ as follows:

$$P_1(k, x, y, s) = \begin{cases} \partial_v^2 \ell_{in}(\omega + st_k \epsilon_k, h(x) + st_k g_k(x), x, y), & k \geq 1 \\ \partial_v^2 \ell_{in}(\omega, h(x), x, y), & k = 0 \end{cases}$$

$$P_2(k, x, y, s) = \begin{cases} (\partial_{\omega,v} \ell_{in}(\omega + st_k \epsilon_k, h(x) + st_k g_k(x), x, y))^\top & k \geq 1 \\ (\partial_{\omega,v} \ell_{in}(\omega, h(x), x, y))^\top, & k = 0. \end{cases}$$

$$\Delta_1(k, x, y, s) = P_1(k, x, y, s) - P_1(0, x, y, s),$$
$$\Delta_2(k, x, y, s) = P_2(k, x, y, s) - P_2(0, x, y, s).$$

By joint differentiability of $(\omega, v) \mapsto \partial_v \ell_{in}(\omega, v, x, y)$ (Assumption (E)), we use the fundamental theorem of calculus to express $E_k$ in terms of $\Delta_1$ and $\Delta_2$:

$$E_k = \mathbb{E}_{\mathbb{P}} \left[ \left\| \mathbb{E}_{\mathbb{P}} \left[ \int_0^1 \mathrm{d}t \, (P_1(k, x, y, s) g_k(x) - P_1(0, x, y, s) g(x) + P_2(k, x, y, s) \epsilon_k - P_2(0, x, y, s) \epsilon) \Big| x \right] \right\|^2 \right]$$

$$\leq \mathbb{E}_{\mathbb{P}} \left[ \mathbb{E}_{\mathbb{P}} \left[ \int_0^1 \mathrm{d}t \, \| P_1(k, x, y, s) g_k(x) - P_1(0, x, y, s) g(x) + P_2(k, x, y, s) \epsilon_k - P_2(0, x, y, s) \epsilon \|^2 \Big| x \right] \right]$$

$$= \mathbb{E}_{\mathbb{P}} \left[ \int_0^1 \mathrm{d}t \, \| P_1(k, x, y, s) g_k(x) - P_1(0, x, y, s) g(x) + P_2(k, x, y, s) \epsilon_k - P_2(0, x, y, s) \epsilon \|^2 \right]$$

$$\leq 4 \underbrace{\mathbb{E}_{\mathbb{P}} \left[ \int_0^1 \mathrm{d}t \, \| \Delta_1(k, x, y, t) \|_{op}^2 \| g(x) \|^2 \right]}_{A_k^{(1)}} + 4 \| \epsilon \|^2 \underbrace{\mathbb{E}_{\mathbb{P}} \left[ \int_0^1 \mathrm{d}t \, \| \Delta_2(k, x, y, t) \|_{op}^2 \right]}_{B_k^{(1)}}$$

$$+ 4 \underbrace{\mathbb{E}_{\mathbb{P}} \left[ \int_0^1 \mathrm{d}t \, \| P_1(k, x, y, t) \|_{op}^2 \| g(x) - g_k(x) \|^2 \right]}_{A_k^{(2)}} + 4 \| \epsilon - \epsilon_k \|^2 \underbrace{\mathbb{E}_{\mathbb{P}} \left[ \int_0^1 \mathrm{d}t \, \| P_2(k, x, y, t) \|_{op}^2 \right]}_{B_k^{(2)}}.$$

The second line uses Jensen's inequality applied to the squared norm, the fourth line results from the "tower" property of conditional distributions. The fifth line uses Jensen's inequality for the square function followed by the operator norm inequality. It remains to show that $A_k^{(1)}, B_k^{(1)}$ and $A_k^{(2)}$ converge to $0$ and that $B_k^{(2)}$ is bounded.

**Upper-bound on $A_k^{(1)}$.** We will use the dominated convergence theorem. Assumption (F) ensures the existence of a positive constant $L$ and a neighborhood $B$ of $\omega$ so that $v \mapsto \left\| \partial_v^2 \ell_{in}(\omega', v, x, y) \right\|_{op}$ is bounded by $L$ for any $\omega', x, y \in B \times \mathcal{X} \times \mathcal{Y}$. Since $\omega + t_k \epsilon_k \to \omega$, then there exists some $K_0$ so that, for any $k \geq K_0$, we can ensure that $\omega + t_k \epsilon_k \in B$. This allows us to deduce that:

$$\| \Delta_1(k, x, y, t) \|_{op}^2 \| g(x) \|^2 \leq 4L^2 \| g(x) \|^2, \tag{38}$$

for any $k \geq K_0$ and any $x, y \in \mathcal{X} \times \mathcal{Y}$, with $\| g(x) \|^2$ being integrable under $\mathbb{P}$.

Moreover, we also have the following point-wise convergence for $\mathbb{P}$-almost all $x \in \mathcal{X}$:

$$\| \Delta_1(k, x, y, t) \|_{op}^2 \| g(x) \|^2 \to 0. \tag{39}$$

Equation (39) follows by noting that $\omega + t_k \epsilon_k \to \omega$ and that $h(x) + t_k g_k(x) \to h(x)$ for $\mathbb{P}$-almost all $x \in \mathcal{X}$, since $t_k$ converges to $0$, $\epsilon_k$ converges to $\epsilon$ and $g_k$ converges to $g$ in $\mathcal{H}$ (a fortiori converges point-wise for $\mathbb{P}$-almost all $x \in \mathcal{X}$). Additionally, the map $(\omega, v) \mapsto \left\| \partial_v^2 \ell_{in}(\omega, v, x, y) \right\|_{op}$ is

continuous by Assumption **(G)**, which allows to establish Equation (39). From Equations (38) and (39) we can apply the dominated convergence theorem which allows to deduce that $A_k^{(1)} \to 0$.

**Upper-bound on $A_k^{(2)}$.** By a similar argument as for $A_k^{(1)}$ and using Assumption **(F)**, we know that there exists $K_0 > 0$ so that for any $k \geq K_0$:

$$\|P_1(k, x, y, t)\|_{op}^2 \leq L^2. \tag{40}$$

Therefore, we directly get that:

$$A_k^{(2)} \leq L^2 \|g(x) - g_k(x)\|_{\mathcal{H}}^2 \to 0, \tag{41}$$

where we used that $g_k \to g$ by construction.

**Upper-bound on $B_k^{(2)}$.** We will show that $\|P_2(k, x, y, t)\|_{op}$ is upper-bounded by a square integrable function under $\mathbb{P}$. By Assumptions **(H)** and **(I)**, there exists a neighborhood $B$ and a positive constant $C$ such that, for all $\omega', v_1, v_2, x, y \in B \times \mathcal{V} \times \mathcal{V} \times \mathcal{X} \times \mathcal{Y}$:

$$\|\partial_{\omega, v_1} \ell_{in}(\omega', 0, x, y)\| \leq C(1 + \|x\| + \|y\|) \tag{42}$$

$$\|\partial_{\omega, v} \ell_{in}(\omega', v_1, x, y) - \partial_{\omega, v} \ell_{in}(\omega', v_2, x, y)\| \leq C \|v_1 - v_2\| \tag{43}$$

By a similar argument as for $A_k^{(1)}$, there exists $K_0$ so that for any $k \geq K_0$, the above inequalities hold when choosing $\omega' = \omega + t_k \epsilon_k$. Using this fact, we obtain the following upper-bound on $\|P_2(k, x, y, t)\|_{op}$ for $k \geq K_0$:

$$
\begin{aligned}
\|P_2(k, x, y, t)\|_{op} &\leq \|\partial_{\omega, v} \ell_{in}(\omega + st_k\epsilon_k, h(x) + st_k g_k(x), x, y) - \partial_{\omega, v} \ell_{in}(\omega + st_k\epsilon_k, 0, x, y)\|_{op} \\
&\quad + \|\partial_{\omega, v} \ell_{in}(\omega + st_k\epsilon_k, 0, x, y)\|_{op} \\
&\leq C(1 + \|h(x) + st_k g_k(x)\| + \|x\| + \|y\|) \\
&\leq C(1 + \|h(x)\| + t_k \|g_k(x)\| + \|x\| + \|y\|)
\end{aligned}
$$

Therefore, by taking expectations and integrating over $t$, it follows:

$$
\begin{aligned}
B_k^{(2)} &\leq C^2 \mathbb{E}_{\mathbb{P}} \left[ (1 + \|h(x)\| + t_k \|g_k(x)\| + \|x\| + \|y\|)^2 \right] \\
&\leq 4C^2 \mathbb{E}_{\mathbb{P}} \left[ \left( 1 + \|h(x)\|^2 + t_k^2 \|g_k(x)\|^2 + \|x\|^2 + \|y\|^2 \right) \right].
\end{aligned}
$$

By construction $t_k^2 \|g_k(x)\|^2 \to 0$ and is therefore a bounded sequence. Moreover, $\mathbb{E}_{\mathbb{P}} \left[ \|h(x)\|^2 \right]$ is finite since $h$ belongs to $\mathcal{H}$. Finally, $\mathbb{E}_{\mathbb{P}} \left[ \|x\|^2 + \|y\|^2 \right] < +\infty$ by Assumption **(A)**. Therefore, we have shown that $B_k^{(2)}$ is bounded.

**Upper-bound on $B_k^{(1)}$.** By a similar argument as for $B_k^{(2)}$ and using again Assumptions **(H)** and **(I)**, there exists $K_0$ so that for any $k \geq K_0$:

$$
\begin{aligned}
\|\Delta_2(k, x, y, t)\|_{op} &\leq \|\partial_{\omega, v} \ell_{in}(\omega + st_k\epsilon_k, h(x) + st_k g_k(x), x, y) - \partial_{\omega, v} \ell_{in}(\omega + st_k\epsilon_k, h(x), x, y)\|_{op} \\
&\quad + \|\partial_{\omega, v} \ell_{in}(\omega + st_k\epsilon_k, h(x), x, y) - \partial_{\omega, v} \ell_{in}(\omega, h(x), x, y)\|_{op} \\
&\leq Ct_k \|g_k(x)\| + \|\partial_{\omega, v} \ell_{in}(\omega + st_k\epsilon_k, h(x), x, y) - \partial_{\omega, v} \ell_{in}(\omega, h(x), x, y)\|_{op},
\end{aligned}
$$

where we used Equation (43) to get an upper-bound on the first terms. By squaring the above inequality and taking the expectation under $\mathbb{P}$ we get:

$$B_k^{(1)} \leq 2Ct_k \|g_k\|_{\mathcal{H}}^2 + 2\mathbb{E}_{\mathbb{P}} \left[ \underbrace{\|\partial_{\omega, v} \ell_{in}(\omega + st_k\epsilon_k, h(x), x, y) - \partial_{\omega, v} \ell_{in}(\omega, h(x), x, y)\|_{op}^2}_{e_k(x, y)} \right]. \tag{44}$$

We only need to show that $\mathbb{E}_{\mathbb{P}} \left[ e_k(x, y) \right]$ converges to 0 since the first term $2Ct_k \|g_k\|_{\mathcal{H}}^2$ already converges to 0 by construction of $t_k$ and $g_k$. To achieve this, we will use the dominated convergence theorem. It is easy to see that $e_k(x, y)$ converges to 0 point-wise by continuity of $\omega \mapsto \partial_{\omega, v} \ell_{in}(\omega, v, x, y)$

(Assumption **(G)**). Therefore, we only need to show that $e_k(x, y)$ is dominated by an integrable function. Provided that $k \geq K_0$, we can use Equations (42) and (43) to get the following upper-bounds:

$$
\begin{aligned}
\frac{1}{4} e_k(x, y) \leq & \left\| \partial_{\omega,v} \ell_{in}(\omega + st_k \epsilon_k, h(x), x, y) - \partial_{\omega,v} \ell_{in}(\omega + st_k \epsilon_k, 0, x, y) \right\|_{op}^2 \\
& + \left\| \partial_{\omega,v} \ell_{in}(\omega, h(x), x, y) - \partial_{\omega,v} \ell_{in}(\omega, 0, x, y) \right\|_{op}^2 \\
& + \left\| \partial_{\omega,v} \ell_{in}(\omega, 0, x, y) \right\|_{op}^2 + \left\| \partial_{\omega,v} \ell_{in}(\omega + st_k \epsilon_k, 0, x, y) \right\|_{op}^2 \\
\leq & 2C^2 \left( 1 + \|h(x)\|^2 + \|x\|^2 + \|y\|^2 \right).
\end{aligned}
$$

The l.h.s. of the last line is an integrable function that is independent of $k$, since $h$ is square integrable by definition and $\|x\|^2 + \|y\|^2$ are integrable by Assumption **(A)**. Therefore, by application of the dominated convergence theorem, it follows that $\mathbb{E}_{\mathbb{P}}[e_k(x, y)] \to 0$, we have shown that $B_k^{(1)} \to 0$.

To conclude, we have shown that the first-order error $E_k$ converges to 0 which means that $(\omega, h) \mapsto \partial_h L_{in}(\omega, h)$ is jointly differentiable on $\Omega \times \mathcal{H}$, with differential given by Equations (34) and (35).

$\square$

### D.3  Proof of Proposition 2.3

*Proof.* The strategy is to show that the conditions on $L_{in}$ and $L_{out}$ stated in Proposition 2.2 hold. By Assumption **(C)**, for any $\omega \in \Omega$, there exists a positive constant $\mu$ and a neighborhood $B$ of $\omega$ on which the function $\ell_{in}(\omega', v, x, y)$ is $\mu$-strongly convex in $v$ for any $(\omega', x, y) \in B \times \mathcal{X} \times \mathcal{Y}$. Therefore, by integration, we directly deduce that $h \mapsto L_{in}(\omega', h)$ is $\mu$ strongly convex in $h$ for any $\omega' \in B$. By Lemmas D.2 and D.4, $h \mapsto L_{in}(\omega, h)$ is differentiable on $\mathcal{H}$ for all $\omega \in \Omega$ and $\partial_h L_{in}$ is Hadamard differentiable on $\Omega \times \mathcal{H}$. Additionally, $L_{out}$ is jointly differentiable in $\omega$ and $h$ by Lemma D.3. Therefore, the conditions on $L_{in}$ and $L_{out}$ for applying Proposition 2.2 hold. Using the notations from Proposition 2.2, we have that the total gradient $\nabla \mathcal{F}(\omega)$ can be expressed as:

$$
\nabla \mathcal{F}(\omega) = g_\omega + B_\omega a_\omega^\star \tag{45}
$$

where $g_\omega = \partial_\omega L_{out}(\omega, h_\omega^\star)$, $B_\omega = \partial_{\omega,h} L_{in}(\omega, h_\omega^\star)$ and where $a_\omega^\star$ is the minimizer of the adjoint objective $L_{adj}$:

$$
L_{adj}(\omega, a) := \tfrac{1}{2} a^\top C_\omega a + a^\top d_\omega,
$$

with $C_\omega = \partial_h^2 L_{in}(\omega, h_\omega^\star)$ and $d_\omega = \partial_h L_{out}(\omega, h_\omega^\star)$. Recalling the expressions of the first and second order differential operators from Lemmas D.2 and D.4, we deduce the expression of the adjoint objective as a sum of two expectations under $\mathbb{P}$ and $\mathbb{Q}$ given the optimal prediction function

$$
\begin{aligned}
L_{adj}(\omega, a) = & \tfrac{1}{2} \mathbb{E}_{(x,y) \sim \mathbb{P}} \left[ a(x)^\top \partial_v^2 \ell_{in}(\omega, h_\omega^\star(x), x, y) a(x) \right] \\
& + \mathbb{E}_{(x,y) \sim \mathbb{Q}} \left[ a(x)^\top \partial_v \ell_{out}(\omega, h_\omega^\star(x), x, y) \right].
\end{aligned}
$$

Furthermore, the vectors $g_\omega$ and $B_\omega a_\omega^\star$ appearing in Equation (45) can also be expressed as expectations:

$$
\begin{aligned}
g_\omega & = \mathbb{E}_{(x,y) \sim \mathbb{Q}} \left[ \partial_\omega \ell_{out}(\omega, h_\omega^\star(x), x, y) \right] \\
B_\omega a_\omega^\star & = \mathbb{E}_{(x,y) \sim \mathbb{P}} \left[ \partial_{\omega,v} \ell_{in}(\omega, h_\omega^\star(x), x, y) a_\omega^\star(x) \right].
\end{aligned}
$$

$\square$

## E   Convergence Analysis

We provide a convergence result of Algorithm 1 to stationary points of $\mathcal{F}$. Our analysis uses the framework of biased stochastic gradient descent (Biased SGD) [Demidovich et al., 2024] where the bias arises from suboptimality errors when solving the inner-level and adjoint problems.

### E.1  Setup and assumptions

**Gradient estimators.** Recall that the total gradient $\nabla \mathcal{F}(\omega)$ admits the following expression under Assumptions (A) to (L):

$$\nabla \mathcal{F}(\omega) = \mathbb{E}_{\mathbb{Q}} \left[ \partial_\omega \ell_{out} \left( \omega, h_\omega^\star(x), x, y \right) \right] + \mathbb{E}_{\mathbb{P}} \left[ \partial_{\omega,v} \ell_{in} \left( \omega, h_\omega^\star(x), x, y \right) a_\omega^\star(x) \right],$$

We denote by $\hat{g}(\omega)$ the gradient estimator, i.e., the mapping $\hat{g} : \Omega \to \Omega$ computed by Algorithm 4 and which admits the following expression:

$$\hat{g}(\omega) = \frac{1}{|\mathcal{B}_{out}|} \sum_{(\tilde{x},\tilde{y}) \in \mathcal{B}_{out}} \partial_\omega \ell_{out} \left( \omega, \hat{h}_\omega(\tilde{x}), \tilde{x}, \tilde{y} \right) + \frac{1}{|\mathcal{B}_{in}|} \sum_{(x,y) \in \mathcal{B}_{in}} \partial_{\omega,v} \ell_{in} \left( \omega, \hat{h}_\omega(x), x, y \right) \hat{a}_\omega(x),$$

where $\mathcal{B}_{in}$ and $\mathcal{B}_{out}$ are samples from $\mathbb{P}$ and $\mathbb{Q}$ independent from $\hat{h}_\omega$ and $\hat{a}_\omega$ and independent from each other (i.e. $\mathcal{B}_{in} \perp \mathcal{B}_{out}$). Here, there are three independent sources of randomness when computing $\hat{g}(\omega)$: estimation of $\hat{h}_\omega$ and $\hat{a}_\omega$ in Algorithms 2 and 3, as well as random batches $\mathcal{B}_{in}$ and $\mathcal{B}_{out}$. We denote by $\mathbb{E}[\cdot]$ the expectation with respect to all random variables appearing in the expression of $\hat{g}(\omega)$, and by $\mathbb{E}[\cdot|\hat{h}_\omega]$ and $\mathbb{E}[\cdot|\hat{h}_\omega, \hat{a}_\omega]$ the conditional expectations knowing $\hat{h}_\omega$ only and both $\hat{h}_\omega$ and $\hat{a}_\omega$. $\hat{g}(\omega)$ is a biased estimator of $\nabla \mathcal{F}(\omega)$ (i.e., $\mathbb{E}[\hat{g}(\omega)]$ is not equal to $\nabla \mathcal{F}(\omega)$), as the bias is due to using sub-optimal solutions $\hat{h}_\omega$ and $\hat{a}_\omega$ instead of $h_\omega^\star$ and $a_\omega^\star$ in the expression of $\hat{g}(\omega)$. Furthermore, we define $G(\omega) := \mathbb{E}\left[\hat{g}(\omega)|\hat{h}_\omega, \hat{a}_\omega\right]$ to be the conditional expectation of $\hat{g}(\omega)$ at $\omega$ given estimates $\hat{h}_\omega$ and $\hat{a}_\omega$ of $h_\omega^\star$ and $a_\omega^\star$. By independence of $\mathcal{B}_{in}$ and $\mathcal{B}_{out}$ from $\hat{h}_\omega$ and $\hat{a}_\omega$, $G(\omega)$ admits the following expression:

$$G(\omega) = \mathbb{E}_{\mathbb{Q}} \left[ \partial_\omega \ell_{out} \left( \omega, \hat{h}_\omega(x), x, y \right) \right] + \mathbb{E}_{\mathbb{P}} \left[ \partial_{\omega,v} \ell_{in} \left( \omega, \hat{h}_\omega(x), x, y \right) \hat{a}_\omega(x) \right],$$

where the expectation is taken w.r.t. $(x, y) \sim \mathbb{Q}$.

**Approximate adjoint objective.**

We introduce the approximate adjoint objective $\tilde{L}_{adj}(\omega, a)$ where $h_\omega^\star$ is replaced by $\hat{h}_\omega$:

$$\tilde{L}_{adj}(\omega, a) = \frac{1}{2} a^\top \partial_h^2 L_{in}(\omega, \hat{h}_\omega) a + a^T \partial_h L_{out}(\omega, \hat{h}_\omega). \tag{46}$$

By independence of the estimator $\hat{h}$ and the samples $\mathcal{B}$ used for computing $\hat{L}_{adj}(\omega, a, \hat{h}_\omega, \mathcal{B})$ it is easy to see that $\mathbb{E}\left[\hat{L}_{adj}(\omega, a, \hat{h}_\omega, \mathcal{B})\Big|\hat{h}_\omega\right] = \tilde{L}_{adj}(\omega, a)$. Hence, it is natural to think of $\hat{a}_\omega$ as an approximation to the minimizer $\tilde{a}_\omega$ of $\tilde{L}_{adj}(\omega, a)$ in $\mathcal{H}$.

**Sub-optimality assumption.**

The following assumption quantifies the sub-optimality errors made by $\hat{h}_\omega$ and $\hat{a}_\omega$.

    **(a)** For some positive constants $\epsilon_{in}$ and $\epsilon_{adj}$ and for all $\omega \in \Omega$, $\hat{h}_\omega$ and $\hat{a}_\omega$ satisfy:

$$\mathbb{E}\left[ L_{in}(\omega, \hat{h}_\omega) - L_{in}(\omega, h_\omega^\star) \right] \leq \epsilon_{in}, \qquad \mathbb{E}\left[ \tilde{L}_{adj}(\omega, \hat{a}_\omega) - \tilde{L}_{adj}(\omega, \tilde{a}_w) \right] \leq \epsilon_{adj}.$$

**Assumptions on $\ell_{in}$.**

    **(b)** **(Strong convexity)** $\ell_{in}(\omega, v, x, y)$ is $\mu$-strongly convex in $v \in \mathcal{V}$.

    **(c)** $v \mapsto \partial_v^2 \ell_{in}(\omega, v, x, y)$ is $C_1$-Lipschitz on $\Omega \times \mathcal{X} \times \mathcal{Y}$.

    **(d)** $v \mapsto \partial_{\omega,v} \ell_{in}(\omega, v, x, y)$ is differentiable and $C_2$-Lipschitz and bounded by $B_2 \in \mathbb{R}$.

**Assumptions on $\ell_{out}$.**

    **(e)** $\|\partial_h L_{out}(\omega, h_\omega^\star)\|_{\mathcal{H}}$ is bounded by a positive constant $B_3$.

    **(f)** $v \mapsto \partial_v \ell_{out}(\omega, v, x, y)$ is $C_4$-Lipschitz for all $(\omega, x, y) \in \Omega \times \mathcal{X} \times \mathcal{Y}$.

    **(g)** $v \mapsto \partial_\omega \ell_{out}(\omega, v, x, y)$ is differentiable and $C_3$-Lipschitz for all $(\omega, x, y) \in \Omega \times \mathcal{X} \times \mathcal{Y}$.

**(h)** $(x, y) \mapsto \partial_\omega \ell_{out}(\omega, h_\omega^\star(x), x, y)$ has a variance bounded by $\sigma_{out}^2$ for all $\omega \in \Omega$.

**Smoothness of the total objective.**

**(i)** Function $\mathcal{F}(\omega)$ is $\mathcal{L}$-smooth for all $\omega \in \Omega$, and bounded from below by $\mathcal{F}^\star \in \mathbb{R}$.

*Remark* E.1. Assumption **(a)** reflects the generalization errors made when optimizing $L_{in}$ and $\tilde{L}_{adj}$ using Algorithms 2 and 3. In the case of over-parameterized networks, these errors can be made smaller by increasing network capacity, number of steps and the number of samples [Allen-Zhu et al., 2019, Du et al., 2019, Zou et al., 2020].

*Remark* E.2. Assumptions **(b)** to **(h)** are similar in spirit to those used for analyzing bi-level optimization algorithms (ex: [Arbel and Mairal, 2022a, Assumptions 1 to 5]). In particular, Assumption **(e)** is even weaker than [Arbel and Mairal, 2022a, Assumptions 2] where $\partial_h L_{out}(\omega, h)$ needs to be bounded for all $\omega, h$ in $\Omega \times \mathcal{H}$. For instance, Assumption **(e)** trivially holds when $\partial_h L_{out}(\omega, h)$ is a linear transformation of $\partial_h L_{in}(\omega, h)$, as is the case for min-max problems. There $\partial_h L_{in}(\omega, h_\omega^\star) = 0$ so that $\partial_h L_{out}(\omega, h_\omega^\star) = 0$ is bounded, while $\partial_h L_{out}(\omega, h)$ might not be bounded in general.

## E.2 Proof of the main result

The general strategy of the proof is to first show that the conditions for applying the general convergence result for biased SGD of Demidovich et al. [2024, Theorem 3] hold. We start with Proposition E.3 which shows that the biased gradient estimate $\hat{g}(\omega)$ satisfies the conditions of Demidovich et al. [2024, Assumption 9].

**Proposition E.3.** *Let Assumptions (a) to (h) and Assumptions (A) to (L) hold.*

$$\nabla\mathcal{F}(\omega)^\top \mathbb{E}[\hat{g}(\omega)] \geq \frac{1}{2}\left\|\nabla\mathcal{F}(\omega)\right\|^2 - \frac{1}{2}\left(c_1\epsilon_{in} + c_2\epsilon_{out}\right) \tag{47}$$

$$\mathbb{E}\left[\left\|\hat{g}(\omega)\right\|^2\right] \leq \underbrace{\sigma_0^2 + 2\left(c_1\epsilon_{in} + c_2\epsilon_{adj}\right)}_{\sigma_{eff}^2} + 2\left\|\nabla\mathcal{F}(\omega)\right\|^2, \tag{48}$$

*where $c_1$ and $c_2$ are constants defined in Equation (50) and $\sigma_0$ is a positive constant defined in Equation (51).*

*Proof.* We prove each bound separately.

**Lower-bound on $\nabla\mathcal{F}(\omega)^\top \mathbb{E}[\hat{g}(\omega)]$.** Fist note that $\mathbb{E}[\hat{g}(\omega)] = \mathbb{E}[G(\omega)]$. Hence, by direct calculation, we have that:

$$\nabla\mathcal{F}(\omega)^\top \mathbb{E}[\hat{g}(\omega)] = \nabla\mathcal{F}(\omega)^\top \mathbb{E}[G(\omega)]$$
$$= \frac{1}{2}\left(\left\|\nabla\mathcal{F}(\omega)\right\|^2 + \mathbb{E}\left[\left\|G(\omega)\right\|^2\right]\right) - \frac{1}{2}\mathbb{E}\left[\left\|G(\omega) - \nabla\mathcal{F}(\omega)\right\|^2\right]$$
$$\geq \frac{1}{2}\left\|\nabla\mathcal{F}(\omega)\right\|^2 - \frac{1}{2}\left(c_1\epsilon_{in} + c_2\epsilon_{out}\right),$$

where we use Lemma E.4 to get the last lower-bound.

**Upper-bound on $\mathbb{E}[\|\hat{g}(\omega)\|^2]$.**

$$\mathbb{E}\left[\left\|\hat{g}(\omega)\right\|^2\right] = \mathbb{E}\left[\left\|\hat{g}(\omega) - G(\omega)\right\|^2\right] + \mathbb{E}\left[\left\|G(\omega) - \nabla\mathcal{F}(\omega) + \nabla\mathcal{F}(\omega)\right\|^2\right]$$
$$\leq \mathbb{E}\left[\left\|\hat{g}(\omega) - G(\omega)\right\|^2\right] + 2\mathbb{E}\left[\left\|G(\omega) - \nabla\mathcal{F}(\omega)\right\|^2\right] + 2\left\|\nabla\mathcal{F}(\omega)\right\|^2$$
$$\leq \sigma_0^2 + 2\left(c_1\epsilon_{in} + c_2\epsilon_{adj}\right) + 2\left\|\nabla\mathcal{F}(\omega)\right\|^2,$$

where the first line uses that $\mathbb{E}[\hat{g}(\omega)] = \mathbb{E}[G(\omega)]$ and the last line uses Lemmas E.4 and E.5.

$\square$

We can now directly use Proposition E.3 and Assumption **(i)** on $\mathcal{F}$ to prove the Theorem 3.1 using the biased SGD convergence result in [Demidovich et al., 2024, Theorem 3].

*Proof of Theorem 3.1.* The proof is a direct application of [Demidovich et al., 2024, Theorem 3] given that the variance and bias conditions on the estimator $\hat{g}(\omega)$ are satisfied by Proposition E.3 and that $\mathcal{F}$ is $\mathcal{L}$-smooth and has a finite lower-bound $\mathcal{F}^\star \in \mathbb{R}$ by Assumption **(i)**. $\square$

### E.3 Bias-variance decomposition.

**Lemma E.4** (Bias control). *Let Assumptions (a) to (g) and Assumptions (A) to (L) hold.*

$$\mathbb{E}\left[\|G(\omega) - \nabla\mathcal{F}(\omega)\|^2\right] \leq c_1\epsilon_{in} + c_2\epsilon_{adj}, \tag{49}$$

*where $c_1$ and $c_2$ are non-negative constants defined in Equation (50).*

*Proof.*

$$G(\omega) - \nabla\mathcal{F}(\omega) = \underbrace{\mathbb{E}_{\mathbb{Q}}\left[\partial_\omega\ell_{out}\left(\omega, \hat{h}_\omega(x), x, y\right) - \partial_\omega\ell_{out}\left(\omega, h_\omega^\star(x), x, y\right)\right]}_{A_1}$$

$$+ \mathbb{E}_{\mathbb{P}}\left[\partial_{\omega,v}\ell_{in}\left(\omega, \hat{h}_\omega(x), x, y\right)\hat{a}_\omega(x) - \partial_{\omega,v}\ell_{in}\left(\omega, h_\omega^\star(x), x, y\right)a_\omega^\star(x)\right]$$

$$= A_1 + \underbrace{\mathbb{E}_{\mathbb{P}}\left[\left(\partial_{\omega,v}\ell_{in}\left(\omega, \hat{h}_\omega(x), x, y\right) - \partial_{\omega,v}\ell_{in}\left(\omega, h_\omega^\star(x), x, y\right)\right)a_\omega^\star(x)\right]}_{A_2}$$

$$+ \underbrace{\mathbb{E}_{\mathbb{P}}\left[\partial_{\omega,v}\ell_{in}\left(\omega, \hat{h}_\omega(x), x, y\right)(\hat{a}_\omega(x) - a_\omega^\star(x))\right]}_{A_3}.$$

We have the following:

$$\|A_1\| \leq MC_3\left\|\hat{h}_\omega - h_\omega^\star\right\|_{\mathcal{H}}$$

$$\|A_2\| \leq C_2\|a_\omega^\star\|_{\mathcal{H}}\left\|\hat{h}_\omega - h_\omega^\star\right\|_{\mathcal{H}} \leq C_2 B_3\mu^{-1}\left\|\hat{h}_\omega - h_\omega^\star\right\|_{\mathcal{H}}$$

$$\|A_3\| \leq B_2\mathbb{E}_{\mathbb{P}}\left[\|\hat{a}_\omega(x) - a_\omega^\star(x)\|\right]$$

$$\leq B_2\left(\mathbb{E}_{\mathbb{P}}\left[\|\hat{a}_\omega(x) - \tilde{a}_\omega(x)\|\right] + \mathbb{E}_{\mathbb{P}}\left[\|a_\omega^\star(x) - \tilde{a}_\omega(x)\|\right]\right)$$

$$\leq B_2\left(\|\hat{a}_\omega - \tilde{a}_\omega\|_{\mathcal{H}} + \|\tilde{a}_\omega - a_\omega^\star\|_{L_1}\right).$$

The first inequality holds since $v \mapsto \partial_\omega\ell_{out}(\omega, v, x, y)$ is $C_3$-Lipschitz by Assumption (g) and $d\mathbb{Q}(\cdot, \mathcal{Y})$ admits a density w.r.t $d\mathbb{P}(\cdot, \mathcal{Y})$ bounded by a positive constant $M$ by Assumption (B). The second inequality holds since $v \mapsto \partial_{\omega,v}\ell_{in}(\omega, v, x, y)$ is $C_2$-Lipschitz by Assumption (d) and $\|a_\omega^\star\|_{\mathcal{H}}$ is upper-bounded by $\mu^{-1}B_3$ as a consequence of Lemma E.6. Finally, the inequality on $\|A_3\|$ holds since $\partial_{\omega,v}\ell_{in}(\omega, v, x, y)$ is bounded by a constant $B_2$ by Assumption (d). Therefore, it holds that the difference between $G(\omega)$ and $\nabla\mathcal{F}$ satisfies:

$$\|G(\omega) - \nabla\mathcal{F}(\omega)\|^2 \leq 3\left(MC_3 + \frac{C_2 B_3}{\mu}\right)^2\left\|\hat{h}_\omega - h_\omega^\star\right\|_{\mathcal{H}}^2 + 3B_2^2\|\hat{a}_\omega - \tilde{a}_\omega\|_{\mathcal{H}}^2 + 3B_2^2\|\tilde{a}_\omega - a_\omega^\star\|_{L_1}^2.$$

Taking the expectation over $\hat{h}_\omega$ and $\hat{a}_\omega$ and using the bounds in Lemma E.6 yields:

$$\mathbb{E}\left[\|G(\omega) - \nabla\mathcal{F}(\omega)\|^2\right] \leq 6\left(MC_3 + C_2 B_3\mu^{-1}\right)^2\mu^{-1}\epsilon_{in} + 6B_2^2\mu^{-1}\epsilon_{adj}$$

$$+ 6B_2^2\mu^{-1}\left(\mu^{-1}C_4 M + \mu^{-2}C_1 B_3\right)^2\epsilon_{in}.$$

Finally, the upper bound on the bias holds with $c_1$ and $c_2$ defined as:

$$c_1 := 6\left(MC_3 + C_2 B_3\mu^{-1}\right)^2\mu^{-1} + 6B_2^2\mu^{-1}\left(\mu^{-1}C_4 M + \mu^{-2}B_3 C_1\right)^2, \qquad c_2 = 6B_2^2\mu^{-1}. \tag{50}$$

$\square$

**Lemma E.5** (Variance control). *Let Assumptions (a) to (h) and Assumptions (A) to (L) hold, then the variance is upper-bounded as follows:*

$$\mathbb{E}\left[\|\hat{g}(\omega) - G(\omega)\|^2\right] \leq \sigma_0^2,$$

*where $\sigma_0$ is a positive constant given by:*

$$\sigma_0^2 := \frac{2}{|\mathcal{B}_{out}|}\left(2C_3^2\mu^{-1}\epsilon_{in} + \sigma_{out}^2\right) + \frac{4B_2^2}{|\mathcal{B}_{in}|}\left(\mu^{-1}\epsilon_{adj} + 2\mu^{-3}C_4^2 M^2\epsilon_{in} + \mu^{-2}B_3^2\right) \tag{51}$$

*Proof.* By definition of $\hat{g}(\omega)$ and $G(\omega)$ we have that:

$$\mathbb{E}\left[\|\hat{g}(\omega) - G(\omega)\|^2\right] = \frac{1}{|\mathcal{B}_{out}|} \underbrace{\mathbb{E}\left[\left\|\partial_\omega \ell_{out}\left(\omega, \hat{h}_\omega(x), x, y\right) - \mathbb{E}_\mathbb{Q}\left[\partial_\omega \ell_{out}\left(\omega, \hat{h}_\omega(x'), x', y'\right)\Big|\hat{h}_\omega\right]\right\|^2\right]}_{V_1}$$

$$+ \frac{1}{|\mathcal{B}_{in}|} \underbrace{\mathbb{E}\left[\left\|\partial_{\omega,v}\ell_{in}\left(\omega, \hat{h}_\omega(x), x, y\right)\hat{a}_\omega(x)\right\|^2\right]}_{V_2}$$

$$- \frac{1}{|\mathcal{B}_{in}|} \underbrace{\mathbb{E}\left[\left\|\mathbb{E}_\mathbb{P}\left[\partial_{\omega,v}\ell_{in}\left(\omega, \hat{h}_\omega(x), x, y\right)\hat{a}_\omega(x)\Big|\hat{h}_\omega, \hat{a}_\omega\right]\right\|^2\right]}_{V_3}$$

$$\leq \frac{V_1}{|\mathcal{B}_{out}|} + \frac{V_2}{|\mathcal{B}_{in}|}.$$

Where the first line is a direct consequence of the independence of $\mathcal{B}_{in}$ and $\mathcal{B}_{out}$. Moreover, we can bound $V_1$ and $V_2$ as follows:

$$V_1 \leq 2\mathbb{E}\left[\left\|\partial_\omega\ell_{out}\left(\omega, \hat{h}_\omega(x), x, y\right) - \partial_\omega\ell_{out}\left(\omega, h_\omega^\star(x), x, y\right)\right\|^2\right]$$

$$+ 2\mathbb{E}_\mathbb{Q}\left[\|\partial_\omega\ell_{out}\left(\omega, h_\omega^\star(x), x, y\right) - \mathbb{E}_\mathbb{Q}\left[\partial_\omega\ell_{out}\left(\omega, h_\omega^\star(x'), x', y'\right)\right]\|^2\right]$$

$$\leq 2C_3^2\mathbb{E}\left[\left\|\hat{h}_\omega - h_\omega^\star\right\|^2\right] + 2\sigma_{out}^2 \leq 4C_3^2\mu^{-1}\epsilon_{in} + 2\sigma_{out}^2$$

$$V_2 \leq B_2^2\mathbb{E}\left[\|\hat{a}_\omega\|_\mathcal{H}^2\right] \leq 2B_2^2\left(\mathbb{E}\left[\|\hat{a}_\omega - \tilde{a}_\omega\|_\mathcal{H}^2\right] + \mathbb{E}\left[\|\tilde{a}_\omega\|_\mathcal{H}^2\right]\right)$$

$$\leq 4B_2^2\left(\mu^{-1}\epsilon_{adj} + 2\mu^{-3}C_4^2M^2\epsilon_{in} + \mu^{-2}B_3^2\right).$$

For $V_1$, we used that $v \mapsto \partial_\omega\ell_{out}(\omega, v, x, y)$ is $C_3$-Lipschitz uniformly in $\omega$, $x$ and $y$ by Assumption (**g**), and that $(x, y) \mapsto \partial_\omega\ell_{out}(\omega, h_\omega^\star(x), x, y)$ has a variance uniformly bounded by $\sigma_{out}^2$ as a consequence of Assumption (**h**). For $V_2$, we use Assumption (**d**) where we have that $\partial_{\omega,v}\ell_{in}$ is uniformly bounded by a constant $B_2$ and apply the bounds on $\mathbb{E}\left[\|\tilde{a}_\omega\|_\mathcal{H}^2\right]$ and $\mathbb{E}\left[\|\hat{a}_\omega - \tilde{a}_\omega\|_\mathcal{H}^2\right]$ from Lemma E.6. □

**Lemma E.6.** *For the optimal prediction and adjoint functions $h_\omega^\star, a_\omega^\star$ defined in Equations (6) and (7), as well as their estimated versions $\hat{h}_\omega, \hat{a}_\omega$ given in Algorithm 1. Under Assumptions (a) to (f) and Assumptions (A) to (L), the following holds:*

$$\mathbb{E}\left[\left\|\hat{h}_\omega - h_\omega^\star\right\|_\mathcal{H}^2\right] \leq 2\mu^{-1}\epsilon_{in}, \qquad \mathbb{E}\left[\|\hat{a}_\omega - \tilde{a}_\omega\|_\mathcal{H}^2\right] \leq 2\mu^{-1}\epsilon_{adj},$$

$$\mathbb{E}\left[\|\tilde{a}_\omega - a_\omega^\star\|_{L_1}^2\right] \leq 2\left(\mu^{-2}C_1B_3 + \mu^{-1}C_4M\right)^2\mu^{-1}\epsilon_{in},$$

$$\mathbb{E}\left[\|\tilde{a}_\omega\|_\mathcal{H}^2\right] \leq 4\mu^{-3}C_4^2M^2\epsilon_{in} + 2\mu^{-2}B_3^2, \qquad \|a_\omega^\star\|_\mathcal{H} \leq \mu^{-1}B_3.$$

*Proof.* We show each of the upper bounds separately.

**Upper-bound on $\mathbb{E}\left[\left\|\hat{h}_\omega - h_\omega^\star\right\|_\mathcal{H}^2\right]$.** We use the strong-convexity Assumption (**b**) and Assumption (**a**) to show the first bound:

$$\mathbb{E}\left[\left\|\hat{h}_\omega - h_\omega^\star\right\|_\mathcal{H}^2\right] \leq \mathbb{E}\left[\frac{2}{\mu}\left(L_{in}(\omega, \hat{h}_\omega) - L_{in}(\omega, h_\omega^\star)\right)\right] \leq \frac{2\epsilon_{in}}{\mu}. \tag{52}$$

**Upper-bound on $\mathbb{E}\left[\|\hat{a}_\omega - \tilde{a}_\omega\|_\mathcal{H}^2\right]$.** The second bound can be proven in the same way as the first, using that, by definition, $\tilde{L}_{adj}$ is continuous and $\mu-$strongly convex in $a$ together with Assumption (**a**).

**Upper-bound on** $\mathbb{E}\left[\|\tilde{a}_w - a_\omega^\star\|_{L_1}^2\right]$. We exploit the closed form expressions of $\tilde{a}_\omega$ and $a_\omega^\star$:

$$\tilde{a}_\omega = -\left(\underbrace{\partial_h^2 L_{in}\left(\omega, \hat{h}_\omega\right)}_{\hat{H}_\omega}\right)^{-1}\underbrace{\partial_h L_{out}\left(\omega, \hat{h}_\omega\right)}_{\hat{b}_\omega}, \qquad a_\omega^\star = -\left(\underbrace{\partial_h^2 L_{in}\left(\omega, h_\omega^\star\right)}_{H_\omega}\right)^{-1}\underbrace{\partial_h L_{out}\left(\omega, h_\omega^\star\right)}_{b_\omega}.$$

By standard linear algebra, the difference $\tilde{a}_\omega - a_\omega^\star$ can be expressed as:

$$\tilde{a}_\omega - a_\omega^\star = H_\omega^{-1}\left(\hat{H}_\omega - H_\omega\right)\hat{H}_\omega^{-1}b_\omega + \hat{H}_\omega^{-1}\left(b_\omega - \hat{b}_\omega\right).$$

By taking the $L_1(\mathbb{P})$ norm of the above and using the upper-bounds from Lemma E.7, we can write:

$$
\begin{aligned}
\|\tilde{a}_\omega - a_\omega^\star\|_{L_1} &\leq \left\|H_\omega^{-1}\left(\hat{H}_\omega - H_\omega\right)\hat{H}_\omega^{-1}b_\omega\right\|_{L_1} + \left\|\hat{H}_\omega^{-1}\left(b_\omega - \hat{b}_\omega\right)\right\|_{L_1} \\
&\leq \mu^{-1}\left\|\left(\hat{H}_\omega - H_\omega\right)\hat{H}_\omega^{-1}b_\omega\right\|_{L_1} + \mu^{-1}\left\|b_\omega - \hat{b}_\omega\right\|_{L_1} \\
&\leq \mu^{-1}C_1\left\|\hat{h}_\omega - h_\omega^\star\right\|_{\mathcal{H}}\left\|\hat{H}_\omega^{-1}b_\omega\right\|_{\mathcal{H}} + \mu^{-1}\left\|b_\omega - \hat{b}_\omega\right\|_{\mathcal{H}} \\
&\leq \mu^{-2}C_1\left\|\hat{h}_\omega - h_\omega^\star\right\|_{\mathcal{H}}\|b_\omega\|_{\mathcal{H}} + \mu^{-1}C_4 M\left\|\hat{h}_\omega - h_\omega^\star\right\|_{\mathcal{H}} \\
&\leq \left(\mu^{-2}C_1 B_3 + \mu^{-1}C_4 M\right)\left\|\hat{h}_\omega - h_\omega^\star\right\|_{\mathcal{H}}.
\end{aligned}
$$

The first line we used the triangular inequality, the second line follows from Equation (53) from Lemma E.7. The third line applies Equation (54) from Lemma E.7 to the first term and uses that $\left\|b_\omega - \hat{b}_\omega\right\|_{L_1} \leq \left\|b_\omega - \hat{b}_\omega\right\|_{\mathcal{H}}$ by Cauchy-Schwarz inequality. The fourth line uses that $\left\|\hat{H}_\omega^{-1}b_\omega\right\|_{\mathcal{H}} \leq \mu^{-1}\|b_\omega\|_{\mathcal{H}}$ for the first term since $\left\|\hat{H}_\omega^{-1}\right\|_{op} \leq \mu^{-1}$ by Assumption **(b)** and uses Equation (55) from Lemma E.7 for the second term. The final bound is obtained using Assumption **(e)** to upper-bound $\|b_\omega\|_{\mathcal{H}}$ by $B_3$. By taking the expectation w.r.t. $\hat{h}_\omega$ and using Equation (52), we get:

$$
\begin{aligned}
\mathbb{E}\left[\|\tilde{a}_\omega - a_\omega^\star\|_{L_1}^2\right] &\leq \left(\mu^{-2}C_1 B_3 + \mu^{-1}C_4 M\right)^2 \mathbb{E}\left[\left\|\hat{h}_\omega - h_\omega^\star\right\|_{\mathcal{H}}^2\right] \\
&\leq 2\left(\mu^{-2}C_1 B_3 + \mu^{-1}C_4 M\right)^2 \mu^{-1}\epsilon_{in}.
\end{aligned}
$$

**Upper-bound on** $\mathbb{E}\left[\|\tilde{a}_\omega\|_{\mathcal{H}}^2\right]$. We use the closed-form expression of $\tilde{a}_\omega$:

$$
\begin{aligned}
\|\tilde{a}_\omega\|_{\mathcal{H}} &= \left\|\hat{H}_\omega^{-1}\hat{b}_\omega\right\|_{\mathcal{H}} \leq \mu^{-1}\left\|\hat{b}_\omega\right\|_{\mathcal{H}} \\
&\leq \mu^{-1}\left(\left\|\hat{b}_\omega - b_\omega\right\|_{\mathcal{H}} + \|b_\omega\|_{\mathcal{H}}\right) \\
&\leq \mu^{-1}\left(C_4 M\left\|\hat{h}_\omega - h_\omega^\star\right\|_{\mathcal{H}} + B_3\right),
\end{aligned}
$$

where the first line uses that $\left\|\hat{H}_\omega^{-1}\right\|_{op} \leq \mu^{-1}$ by Assumption **(b)**, the second line follows by triangular inequality while the last line uses Equation (55) from Lemma E.7 for the first term and Assumption **(e)** for the second terms. By squaring the above bound and taking expectation w.r.t. $\hat{h}_\omega$, we get:

$$
\begin{aligned}
\mathbb{E}\left[\|\tilde{a}_\omega\|_{\mathcal{H}}^2\right] &\leq \mathbb{E}\left[\left(\mu^{-1}C_4 M\left\|\hat{h}_\omega - h_\omega^\star\right\| + \mu^{-1}B_3\right)^2\right] \\
&\leq 2\mu^{-2}C_4^2 M^2 \mathbb{E}\left[\left\|\hat{h}_\omega - h_\omega^\star\right\|^2\right] + 2\mu^{-2}B_3^2 \\
&\leq 4\mu^{-3}C_4^2 M^2 \epsilon_{in} + 2\mu^{-2}B_3^2,
\end{aligned}
$$

where the last line uses Equation (52).

**Upper-bound on** $\|a_\omega^\star\|_{\mathcal{H}}$. Using the closed-form expression of $a^\star$, it holds that:

$$\|a_\omega^\star\|_{\mathcal{H}} = \left\|H_\omega^{-1} b_\omega\right\|_{\mathcal{H}} \leq \mu^{-1} B_3,$$

where we used that $\left\|H_\omega^{-1}\right\|_{op} \leq \mu^{-1}$ by Assumption (b) and that $\|b_\omega\|_{\mathcal{H}} \leq B_2$ by Assumption (d). $\square$

**Lemma E.7.** *Consider the operators $H_\omega, \hat{H}_\omega, b_\omega, \hat{b}_\omega$ defined in Lemma E.6. Under Assumptions (b), (c) and (f) and Assumptions (A) to (L), the following holds for any $(\omega, s) \in \Omega \times \mathcal{H}$:*

$$\left\|H_\omega^{-1} s\right\|_{L_1} \leq \mu^{-1} \|s\|_{L_1}, \qquad \left\|\hat{H}_\omega^{-1} s\right\|_{L_1} \leq \mu^{-1} \|s\|_{L_1} \tag{53}$$

$$\left\|\left(\hat{H}_\omega - H_\omega\right) s\right\|_{L_1} \leq C_1 \left\|\hat{h}_\omega - h_\omega^\star\right\|_{\mathcal{H}} \|s\|_{\mathcal{H}}, \tag{54}$$

$$\left\|b_\omega - \hat{b}_\omega\right\|_{\mathcal{H}} \leq C_4 M \left\|\hat{h}_\omega - h_\omega^\star\right\|_{\mathcal{H}}. \tag{55}$$

*Proof.* We show each of the upper bounds separately.

**Upper-bound on** $\left\|H_\omega^{-1} s\right\|_{L_1}$ **and** $\left\|\hat{H}_\omega^{-1} s\right\|_{L_1}$. Using strong convexity Assumption (b), we have the following inequality for the Hessian operator $H_\omega$ acting on a function $c \in \mathcal{H} \subset L_1(\mathbb{P})$:

$$\|H_\omega c\|_{L_1} = \mathbb{E}_{\mathbb{P}} \left[\left\|\mathbb{E}_{\mathbb{P}} \left[\partial_v^2 \ell_{in}(\omega, h_\omega^\star(x), x, y)\big| x\right] c(x)\right\|\right] \geq \mu \|c\|_{L_1}.$$

by positive-definiteness of $H_\omega$, we take $c = H_\omega^{-1} s$ for some $s \in \mathcal{H}$:

$$\left\|H_\omega^{-1} s\right\|_{L_1} \leq \mu^{-1} \|s\|_{L_1}.$$

By the same arguments, the above bound applies to $\left\|\hat{H}_\omega^{-1} s\right\|_{L_1}$.

**Upper-bound on** $\left\|\left(\hat{H}_\omega - H_\omega\right) c\right\|_{L_1}$. We can express the operator $\left(\hat{H}_\omega - H_\omega\right)$ acting on some $s \in \mathcal{H}$ as follows:

$$\left(\hat{H}_\omega - H_\omega\right) s = \mathbb{E}_{\mathbb{P}} \left[\left(\mathbb{E}_{\mathbb{P}} \left[\partial_v^2 \ell_{in}(\omega, \hat{h}_\omega(x), x, y) - \partial_v^2 \ell_{in}(\omega, h_\omega^\star(x), x, y)\big| x\right]\right) s(x)\right].$$

Using Assumption (c) we upper-bound the $L_1$ norm of the above quantity as follows:

$$\left\|\left(\hat{H}_\omega - H_\omega\right) s\right\|_{L_1} \leq C_1 \mathbb{E}_{\mathbb{P}} \left[\left\|\hat{h}_\omega(x)) - h_\omega^\star(x)\right\| \|s(x)\|\right] \leq C_1 \left\|\hat{h}_\omega - h_\omega^\star\right\|_{\mathcal{H}} \|s\|_{\mathcal{H}},$$

where we used Cauchy-Schwarz inequality to get the last inequality.

**Upper-bound on** $\left\|b_\omega - \hat{b}_\omega\right\|_{\mathcal{H}}$. Using Lemma D.3 to get an expression of $b_\omega$ and $\hat{b}_\omega$, we obtain the following upper-bound:

$$\left\|b_\omega - \hat{b}_\omega\right\|_{\mathcal{H}}^2 = \mathbb{E}_{\mathbb{P}} \left[\left\|\mathbb{E}_{\mathbb{Q}} \left[\left(\partial_v \ell_{out}(\omega, h_\omega^\star(x), x, y) - \partial_v \ell_{out}(\omega, \hat{h}_\omega(x), x, y)\right) | x\right] r(x)\right\|^2\right]$$

$$\leq C_4^2 M^2 \left\|\hat{h}_\omega - h_\omega^\star\right\|_{\mathcal{H}}^2,$$

where we used that the density $r(x)$ is upper-bounded by a positive constant $M$ by Assumption (B) and that $\partial_v \ell_{out}$ is $C_4$ Lipschitz in its second argument by Assumption (f). $\square$

# F  Connection with Parametric Implicit Differentiation

## F.1  Parametric approximation of the functional bilevel problem

In this section, we approximate the functional problem in Equation (FBO) with a parametric bilevel problem where inner-level functions are parametrized as $h(x) = \tau(\theta)(x)$ with parameters $\theta$. Here, the inner-level variable is $\theta$ instead of the function $h$. Standard bilevel optimization algorithms like AID can be applied, which involve differentiating twice with respect to the parametric model. However, for models like deep neural networks, the inner objective may not be strongly convex in $\theta$, leading to a non-positive or degenerate Hessian (Proposition F.1). This can cause numerical instabilities and divergence from the gradient in Equation (4) (Proposition F.2), especially when using AID, which relies on solving a quadratic problem defined by the Hessian.

If the model has multiple solutions, the Hessian may be degenerate, making the implicit function theorem inapplicable. In contrast, functional implicit differentiation requires solving a positive definite quadratic problem in $\mathcal{H}$ to find an adjoint function $a_\omega^\star$, ensuring a solution even when $h_\omega^\star$ is sub-optimal, due to the strong convexity of $L_{in}(\omega, h)$. This stability with sub-optimal solutions is crucial for practical algorithms like the one in Section 3, where the optimal prediction function is approximated within a parametric family, such as neural networks.

Formally, to establish a connection with parametric implicit differentiation, let us consider $\tau : \Theta \mapsto \mathcal{H}$ to be a map from a finite dimensional set of parameters $\Theta$ to the functional Hilbert space $\mathcal{H}$ and define a parametric version of the outer and inner objectives in Equation (FBO) restricted to functions in $\mathcal{H}_\Theta := \{\tau(\theta) \mid \theta \in \Theta\}$:

$$G_{out}(\omega, \theta) := L_{out}(\omega, \tau(\theta)) \qquad G_{in}(\omega, \theta) := L_{in}(\omega, \tau(\theta)). \tag{56}$$

The map $\tau$ can typically be a neural network parameterization and allows to obtain a "more tractable" approximation to the abstract solution $h_\omega^\star$ in $\mathcal{H}$ where the function space $\mathcal{H}$ is often too large to perform optimization. This is typically the case when $\mathcal{H}$ is an $L_2$-space of functions as we discuss in more details in Section 3. When $\mathcal{H}$ is a Reproducing Kernel Hilbert Space (RKHS), $\tau$ may also correspond to the Nyström approximation [Williams and Seeger, 2000], which performs the optimization on a finite-dimensional subspace of an RKHS spanned by a few data points.

The corresponding parametric version of the problem (FBO) is then formally defined as:

$$\min_{\omega \in \Omega} \; G_{tot}(\omega) := G_{out}(\omega, \theta_\omega^\star)$$
$$\text{s.t. } \theta_\omega^\star \in \arg\min_{\theta \in \Theta} \; G_{in}(\omega, \theta). \tag{PBO}$$

The resulting bilevel problem in Equation (PBO) often arises in machine learning but is generally ambiguously defined without further assumptions on the map $\tau$ as the inner-level problem might admit multiple solutions [Arbel and Mairal, 2022b]. Under the assumption that $\tau$ is twice continuously differentiable and the rather strong assumption that the parametric Hessian $\partial_\theta^2 G_{in}(\omega, \theta_\omega^\star)$ is invertible for a given $\omega$, the expression for the total gradient $\nabla_\omega G_{tot}(\omega)$ follows by direct application of the parametric implicit function theorem [Pedregosa, 2016]:

$$\nabla_\omega G_{tot}(\omega) = \partial_\omega G_{out}(\omega, \theta_\omega^\star) + \partial_{\omega,\theta} G_{in}(\omega, \theta_\omega^\star) u_\omega^\star$$
$$u_\omega^\star = -\partial_\theta^2 G_{in}(\omega, \theta_\omega^\star)^{-1} \partial_\theta G_{out}(\omega, \theta_\omega^\star), \tag{57}$$

where $u_\omega^\star$ is the adjoint vector in $\Theta$. Without further assumptions, the expression of the gradient in Equation (57) is generally different from the one obtained in Proposition 2.2 using the functional point of view. Nevertheless, a precise connection between the functional and parametric implicit gradients can be obtained under expressiveness assumptions on the parameterization $\tau$, as discussed in the next two propositions.

**Proposition F.1.** *Under the same assumptions as in Proposition 2.2 and assuming that $\tau$ is twice continuously differentiable, the following expression holds for any $(\omega, \theta) \in \Omega \times \Theta$:*

$$\partial_\theta^2 G_{in}(\omega, \theta) := \partial_\theta \tau(\theta) \partial_h^2 L_{in}(\omega, \tau(\theta)) \partial_\theta \tau(\theta)^\top + \partial_\theta^2 \tau(\theta) \left[ \partial_h L_{in}(\omega, \tau(\theta)) \right], \tag{58}$$

*where $\partial_\theta^2 \tau(\theta)$ is a linear operator measuring the* distortion *induced by the parameterization and acts on functions in $\mathcal{H}$ by mapping them to a matrix $p \times p$ where $p$ is the dimension of the parameter space $\Theta$. If, in addition, $\tau$ is expressive enough so that $\tau(\theta_\omega^\star) = h_\omega^\star$, then the above expression simplifies to:*

$$\partial_\theta^2 G_{in}(\omega, \theta_\omega^\star) := \partial_\theta \tau(\theta_\omega^\star) C_\omega \partial_\theta \tau(\theta_\omega^\star)^\top. \tag{59}$$

Proposition F.1 follows by direct application of the chain rule, noting that the distortion term on the right of (58) vanishes when $\theta = \theta^\star_\omega$ since $\partial_h L_{in}(\omega, \tau(\theta^\star_\omega)) = \partial_h L_{in}(\omega, h^\star_\omega) = 0$ by optimality of $h^\star_\omega$. A consequence is that, for an optimal parameter $\theta^\star_\omega$, the parametric Hessian is necessarily symmetric positive semi-definite. However, for an arbitrary parameter $\theta$, the distortion does not vanish in general, making the Hessian possibly non-positive. This can result in numerical instability when using algorithms such as AID for which an adjoint vector is obtained by solving a quadratic problem defined by the Hessian matrix $\partial^2_\theta G_{in}$ evaluated on approximate minimizers of the inner-level problem. Moreover, if the model admits multiple solutions $\theta^\star_\omega$, the Hessian is likely to be degenerate making the implicit function theorem inapplicable and the bilevel problem in Equation (PBO) ambiguously defined[1]. On the other hand, the functional implicit differentiation requires finding an adjoint function $a^\star_\omega$ by solving a positive definite quadratic problem in $\mathcal{H}$ which is always guaranteed to have a solution even when the inner-level prediction function is only approximately optimal.

**Proposition F.2.** *Assuming that $\tau$ is twice continuously differentiable and that for a fixed $\omega \in \Omega$ we have $\tau(\theta^\star_\omega) = h^\star_\omega$, and $J_\omega := \partial_\theta \tau(\theta^\star_\omega)$ has a full rank, then, under the same assumptions as in Proposition 2.2, $\nabla_\omega G_{tot}(\omega)$ is given by:*

$$\nabla_\omega G_{tot}(\omega) = g_\omega + B_\omega P_\omega a^\star_\omega, \tag{60}$$

*where $P_\omega : \mathcal{H} \to \mathcal{H}$ is a projection operator of rank $\dim(\Theta)$. If, in addition, the equality $\tau(\theta^\star_{\omega'}) = h^\star_{\omega'}$ holds for all $\omega'$ in a neighborhood of $\omega$, then $\nabla_\omega G_{tot}(\omega) := \nabla \mathcal{F}(\omega) = g_\omega + B_\omega a^\star_\omega$.*

Proposition F.2, which is proven below, shows that, even when the parametric family is expressive enough to recover the optimal prediction function $h^\star_\omega$ at a single value $\omega$, the expression of the total gradient in Equation (60) using parametric implicit differentiation might generally differ from the one obtained using its functional counterpart. Indeed the projector $P_\omega$, which has a rank equal to $\dim(\Theta)$, biases the adjoint function by projecting it into a finite dimensional space before applying the cross derivative operator. Only under a much stronger assumption on $\tau$, requiring it to recover the optimal prediction function $h^\star_\omega$ in a neighborhood of the outer-level variable $\omega$, both parametric and functional implicit differentiation recover the same expression for the total gradient. In this case, the projector operator aligns with the cross-derivative operator so that $B_\omega P_\omega = B_\omega$. Finally, note that the expressiveness assumptions on $\tau$ made in Propositions F.1 and F.2 are only used here to discuss the connection with the parametric implicit gradient and are not required by the method we introduce in Section 3.

*Proof of Proposition F.2.* Here we want to show the connection between the *parametric* gradient of the outer variable $\nabla_\omega G_{tot}(\omega)$ usually used in approximate differentiation methods and the *functional* gradient of the outer variable $\nabla \mathcal{F}(\omega)$ derived from the functional bilevel problem definition in Equation (FBO). Recall the definition of the *parametric* inner objective $G_{in}(\omega, \theta) := L_{in}(\omega, \tau(\theta))$. According to Proposition F.1, we have the following relation

$$\partial^2_\theta G_{in}(\omega, \theta^\star_\omega) := J_\omega C_\omega J^\top_\omega \quad \text{with} \quad J_\omega := \partial_\theta \tau(\theta^\star_\omega).$$

By assumption, $J_\omega$ has a full rank which matches the dimension of the parameter space $\Theta$. Recall from the assumptions of Theorem 2.1 that the Hessian operator $C_\omega$ is positive definite by the strong convexity of the inner-objective $L_{in}$ in the second argument. We deduce that $\partial^2_\theta G_{in}(\omega, \theta^\star_\omega)$ must be invertible, since, by construction, the dimension of $\Theta$ is smaller than that of the Hilbert space $\mathcal{H}$ which has possibly infinite dimension. Recall from Theorem 2.1, $B_\omega := \partial_{\omega,h} L_{in}(\omega, h^\star_\omega)$ and the assumption that $\tau(\theta^\star_\omega) = h^\star_\omega$. We apply the parametric implicit function theorem to get the following expression of the Jacobian $\partial_\omega \theta^\star_\omega$:

$$\partial_\omega \theta^\star_\omega := -B_\omega J^\top_\omega \left( J_\omega C_\omega J^\top_\omega \right)^{-1}.$$

Hence, differentiating the total objective $G_{tot}(\omega) := G_{out}(\omega, \theta^\star_\omega) = L_{out}(\omega, \tau(\theta^\star_\omega))$ and applying the chain rule directly results in the following expression:

$$\nabla_\omega G_{tot}(\omega) = g_\omega - B_\omega J^\top_\omega \left( J_\omega C_\omega J^\top_\omega \right)^{-1} J_\omega d_\omega, \tag{61}$$

with previously defined $g_\omega := \partial_\omega L_{out}(\omega, h^\star_\omega)$ and $d_\omega := \partial_h L_{out}(\omega, h^\star_\omega)$.

---

[1] although a generalized version of such a theorem was recently provided under restrictive assumptions [Arbel and Mairal, 2022b].

We now introduce the operator $P_\omega := J_\omega^\top \left( J_\omega C_\omega J_\omega^\top \right)^{-1} J_\omega C_\omega$. The operator $P_\omega$ is a projector as it satisfies $P_\omega^2 = P_\omega$. Hence, using the fact that the Hessian operator is invertible, and recalling that the adjoint function is given by $a_\omega^\star = -C_\omega^{-1} d_\omega$, we directly get form Equation (61) that:

$$\nabla_\omega G_{tot}(\omega) := g_\omega + B_\omega P_\omega a_\omega^\star.$$

If we further assume that $\tau(\theta_{\omega'}^\star) = h_{\omega'}^\star$ holds for all $\omega'$ in a neighborhood of $\omega$, then differentiating with respect to $\omega$ results in the following identity:

$$\partial_\omega \theta_\omega^\star J_\omega = \partial_\omega h_\omega^\star.$$

Using the expression of $\partial_\omega h_\omega^\star$ from Equation (3), we have the following identity:

$$-\partial_\omega \theta_\omega^\star J_\omega C_\omega = B_\omega.$$

In other words, $B_\omega$ is of the form $B_\omega := D J_\omega C_\omega$ for some finite dimensional matrix $D$ of size $\dim(\Omega) \times \dim(\Theta)$. Recalling the expression of the total gradient, we can deduce the equality between *parametric* and *functional* gradients:

$$\begin{aligned}
\nabla_\omega G_{tot}(\omega) &= g_\omega - B_\omega J_\omega^\top \left( J_\omega C_\omega J_\omega^\top \right)^{-1} J_\omega d_\omega \\
&= g_\omega - D J_\omega C_\omega J_\omega^\top \left( J_\omega C_\omega J_\omega^\top \right)^{-1} J_\omega d_\omega \\
&= g_\omega - D J_\omega d_\omega \\
&= g_\omega - D J_\omega C_\omega C_\omega^{-1} d_\omega \\
&= g_\omega + B_\omega a_\omega^\star = \nabla \mathcal{F}(\omega).
\end{aligned}$$

The first equality follows from the general expression of the total gradient $\nabla_\omega G_{tot}(\omega)$. In the second line we use the expression of $B_\omega$ which then allows to simplify the expression in the third line. Then, recalling that the Hessian operator $C_\omega$ is invertible, we get the fourth line. Finally, the result follows by using again the expression of $B_\omega$ and recalling the definition of the adjoint function $a_\omega^\star$. $\qquad\square$

### F.2 Computational Cost and Scalability

The optimization of the prediction function $\hat{h}_\omega$ in the inner-level optimization loop is similar to AID, although the total gradient computation differs significantly. Unlike AID, Algorithm 1 does not require differentiating through the parameters of the prediction model when estimating the total gradient $\nabla \mathcal{F}(\omega)$. This property results in an improved cost in time and memory in most practical cases as shown in Table 1 and Figure 4. More precisely, AID requires computing Hessian-vector products of size $p_{in}$, which corresponds to the number of hidden layer weights of the neural network $\hat{h}_\omega$. While *FuncID* only requires Hessian-vector products of size $d_v$, i.e. the output dimension of $\hat{h}_\omega$. In many practical cases, the network's parameter dimension $p_{in}$ is much larger than its output size $d_v$, which results in considerable benefits in terms of memory when using *FuncID* rather than AID, as shown in Figure 4 (left). Furthermore, unlike AID, the overhead of evaluating Hessian-vector products in *FuncID* is not affected by the time cost for evaluating the prediction network. When $\hat{h}_\omega$ is a deep network, such an overhead increases significantly with the network size, making AID significantly slower (Figure 4 (right)).

| Method | Time cost | Memory cost |
|--------|-----------|-------------|
| AID | $\gamma(T_{L_{in}} + T_h)$ | $\beta p_{in} + M_h$ |
| *FuncID* | $\gamma T_{L_{in}} + (2 + \delta)T_a + T_h$ | $\beta d_v + M_a$ |

Table 1: Cost in time and memory for performing a single total gradient estimation using either AID or *FuncID* and assuming the prediction model is learned. **Time cost**: $T_h$ and $T_a$ represent the time cost of evaluating both prediction and adjoint models $h$ and $a$, while $T_{in}$ is the time cost for evaluating the inner objective once the outputs of $h$ are computed. The factors $\gamma$ and $\delta$ are multiplicative overheads for evaluating hessian-vector products and gradient. **Memory cost**: $M_h$ and $M_a$ represent the memory cost of storing the intermediate outputs of $h$ and $a$, $p_{in}$ and $d_v$ are the memory costs of storing the Hessian-vector product for AID and *FuncID* respectively and $\beta$ is a multiplicative constant that depends on a particular implementation.

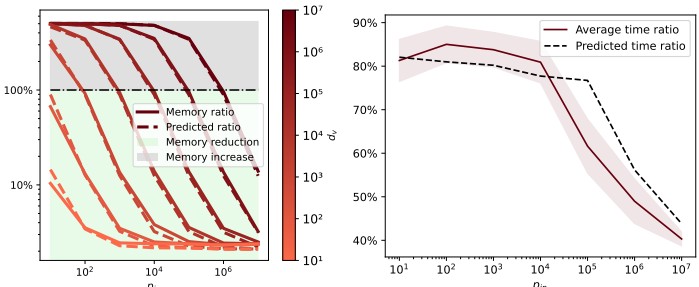

Figure 4: Memory and time comparison of a single total gradient approximation using *FuncID* vs AID. (**Left**) Memory usage ratio of *FuncID* over AID vs inner model parameter dimension $p_{in}$, for various values of the output dimension $d_v$. (**Right**) Time ratio of *FuncID* over AID vs inner model parameter dimension $p_{in}$ averaged over several values of $d_v$ and $10^4$ evaluations. The continuous lines are experimental results obtained using a JAX implementation [Bradbury et al., 2018] running on a GPU. The dashed lines correspond to theoretical estimates obtained using the algorithmic costs given in Table 1 with $\gamma = 12, \delta = 2$ for time, and the constant factors in the memory cost fitted to the data.

# G   Additional Details about 2SLS Experiments

We closely follow the experimental setting of the state-of-the-art method DFIV [Xu et al., 2021a]. The goal of this experiment is to learn a model $f_\omega$ approximating the structural function $f_{struct}$ that accurately describes the effect of the treatment $t$ on the outcome $o$ with the help of an instrument $x$, as illustrated in Figure 5.

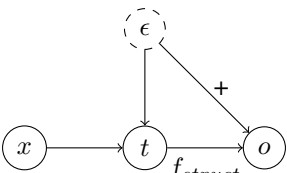

Figure 5: The causal relationships between all variables in an Instrumental Variable (IV) causal graph, where $t$ is the treatment variable (*dsprites* image), $o$ is the outcome (label in $\mathbb{R}$), $x$ is the instrument and $\epsilon$ is the unobserved confounder

## G.1   Dsprites data.

We follow the exact same data generation procedure as in Xu et al. [2021a, Appendix E.3]. From the *dsprites* dataset [Matthey et al., 2017], we generate the treatment $t$ and outcome $o$ as follows:

1. Uniformly sample latent parameters *scale, rotation, posX, posY* from *dsprites*.

2. Generate treatment variable $t$ as

$$t = Fig(scale, rotation, posX, posY) + \lambda.$$

3. Generate outcome variable $o$ as

$$o = \frac{\|At\|_2^2 - 5000}{1000} + 32(posY - 0.5) + \varepsilon.$$

Here, function *Fig* returns the corresponding image to the latent parameters, and $\lambda, \varepsilon$ are noise variables generated from $\lambda \sim \mathcal{N}(0.0, 0.1I)$ and $\varepsilon \sim \mathcal{N}(0.0, 0.5)$. Each element of the matrix $A \in \mathbb{R}^{10 \times 4096}$ is generated from Unif$(0.0, 1.0)$ and fixed throughout the experiment. From the data generation process, we can see that $t$ and $o$ are confounded by *posY*. We use the instrumental variable $x = (scale, rotation, posX) \in \mathbb{R}^3$, and figures with random noise as treatment variable $t$. The variable

*posY* is not revealed to the model, and there is no observable confounder. The structural function for this setting is

$$f_{struct}(t) = \frac{\|At\|_2^2 - 5000}{1000}.$$

Test data points are generated from grid points of latent variables. The grid consist of 7 evenly spaced values for *posX, posY*, 3 evenly spaced values for *scale*, and 4 evenly spaced values for *orientation*.

## G.2   Experimental details

All results are reported over an average of 20 runs with different seeds on *24GB NVIDIA RTX A5000* GPUs.

**Feature maps.**   As in the DFIV setting, we approximate the true structural function $f_{struct}$ with $f_\omega = u^\top \psi_\chi(t)$ where $\psi_\chi$ is a feature map of the treatment $t$, $u$ is a vector in $\mathbb{R}^{d_2}$, and $f_\omega$ is parameterized by $\omega = (u, \chi)$. To solve the inner-problem of the bilevel formulation in Section 4.1, the inner prediction function $h_\omega$ is optimized over functions of the form $h(x) = V\phi(x)$ where we denote $\phi$ the feature map of the instrument $x$ and $V$ is a matrix in $\mathbb{R}^{d_1 \times d_1}$. The feature maps $\psi_\chi$ and $\phi$ are neural networks (Table 2) that are optimized using empirical objectives from Section 3.1 and synthetic *dsprites* data, the linear weights $V$ and $u$ are fitted exactly at each iteration.

**Choice of the adjoint function in *FuncID*.**   In the *dsprites* experiment, we call *linear* FuncID the functional implicit diff. method with a linear choice of the adjoint function. *Linear* FuncID uses an adjoint function of the form $a_\omega^\star(x) = W\phi(x)$ with $W \in \mathbb{R}^{d_1 \times d_1}$. In other words, to find $a_\omega^\star$, the features $\phi$ are fixed and only the optimal linear weight $W$ is computed in closed-form. In the *FuncID* method, the adjoint function lives in the same function space as $h_\omega$. This is achieved by approximating $a_\omega^\star$ with a separate neural network with the same architecture as $h_\omega$.

| **Layer** | instrument feature map $\phi$ | **Layer** | treatment feature map $\psi_\chi$ |
|:---:|:---:|:---:|:---:|
| 1 | Input($x$) | 1 | Input($t$) |
| 2 | FC(3, 256), SN, ReLU | 2 | FC(4096, 1024), SN, ReLU |
| 3 | FC(256, 128), SN, ReLU, LN | 3 | FC(1024, 512), SN, ReLU, LN |
| 4 | FC(128, 128), SN, ReLU, LN | 4 | FC(512, 128), SN, ReLU |
| 5 | FC(128, 32), SN, LN, ReLU | 5 | FC(128, 32), SN, LN, Tanh |

Table 2: Neural network architectures used in the *dsprites* experiment for all models. The *FuncID* model has an extra fully-connected layer FC(32, 1) in both networks. LN corresponds to *LayerNorm* and SN to *SpectralNorm*.

**Hyper-parameter tuning.**   As in the setup of DFIV, for training all methods, we use 100 outer iterations ($N$ in Algorithm 1), and 20 inner iterations ($M$ in Algorithm 1) per outer iteration with full-batch. We select the hyper-parameters based on the best validation loss, which we obtain using a validation set with instances of all three variables $(t, o, x)$ [Xu et al., 2021a, Appendix A]. Because of the number of linear solvers, the grid search performed for AID is very large, so we only run it with one seed. For other methods, we run the grid search on 4 different seeds and take the ones with the highest average validation loss. Additionally, for the hyper-parameters that are not tuned, we take the ones reported in Xu et al. [2021a].

- **Deep Feature Instrumental Variable Regression:** All DFIV hyper-parameters are set based on the best ones reported in Xu et al. [2021a].

- **Approximate Implicit Differentiation**: We perform a grid search over 5 linear solvers (two variants of gradient descent, two variants of conjugate gradient and an identity heuristic solver), linear solver learning rate $10^{-n}$ with $n \in \{3, 4, 5\}$, linear solver number of iterations $\{2, 10, 20\}$, inner optimizer learning rate $10^{-n}$ with $n \in \{2, 3, 4\}$, inner optimizer weight decay $10^{-n}$ with $n \in \{1, 2, 3\}$ and outer optimizer learning rate $10^{-n}$ with $n \in \{2, 3, 4\}$.

- **Iterative Differentiation**: We perform a grid search over number of "unrolled" inner iterations $\{2, 5\}$ (this is chosen because of memory constraints since "unrolling" an iteration is memory-heavy), number of warm-start inner iterations $\{18, 15\}$, inner optimizer learning

rate $10^{-n}$ with $n \in \{2, 3, 4\}$, inner optimizer weight decay $10^{-n}$ with $n \in \{1, 2, 3\}$ and outer optimizer learning rate $10^{-n}$ with $n \in \{2, 3, 4\}$.

- **Gradient Penalty**: The method is based on Eq. 5.1 in Shen and Chen [2023], for this single-level method we perform a grid search on the learning rate $10^{-n}$ with $n \in \{2, 3, 4, 5, 6\}$, weight decay $10^{-n}$ with $n \in \{1, 2, 3\}$, and the penalty weight $10^{-n}$ with $n \in \{0, 1, 2, 3\}$. Since the method has only a single optimization loop, we increase the number of total iterations to 2000 compared to the other methods (100 outer-iterations and 20 inner iterations).

- **Value Function Penalty**: The method is based on Eq. 3.2 in Shen and Chen [2023], for this method we perform the same grid search as for the Gradient Penalty method. However, since this method has an inner loop, we perform 100 outer iterations and perform a grid search on the number of inner iterations with 10 and 20.

- *FuncID*: We perform a grid search over the number of iterations for learning the adjoint network $\{10, 20\}$, adjoint optimizer learning rate $10^{-n}$ with $n \in \{2, 3, 4, 5, 6\}$ and adjoint optimizer weight decay $10^{-n}$ with $n \in \{1, 2, 3\}$. The rest of the parameters are the same as for DFIV since the inner and outer models are almost equivalent to the treatment and instrumental networks used in their experiments.

### G.3 Additional results

We run an additional experiment with $10k$ training points using the same setting described above to illustrate the effect of the sample size on the methods. Figure 6 shows that a similar conclusion can be drawn when increasing the training sample size from $5k$ to $10k$, thus illustrating the robustness of the obtained results.

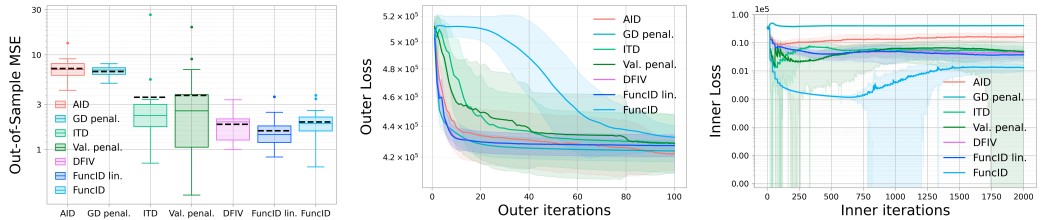

Figure 6: Performance metrics for Instrumental Variable (IV) regression. **(Left)** final test loss. **(Middle)** outer loss vs training iterations, **(Right)** inner loss vs training iterations. All results are averaged over 20 runs with 10000 training samples and 588 test samples.

## H  Additional Details about Model-Based RL Experiments

### H.1  Closed-form expression for the adjoint function

For the *FuncID* method, we exploit the structure of the adjoint objective to obtain a closed-form expression of the adjoint function $a_\omega^\star$. In the model-based RL setting, the unregularized adjoint objective has a simple expression of the form:

$$\hat{L}_{adj}(\omega, a, \hat{h}_\omega, \mathcal{B}) = \frac{1}{2|\mathcal{B}_{in}|} \sum_{(x,y) \in \mathcal{B}_{in}} \|a(x)\|^2 \tag{62}$$

$$+ \frac{1}{|\mathcal{B}_{out}|} a(x)^\top \partial_v f(\hat{h}_\omega(x), y). \tag{63}$$

The key observation here is that the same batches of data are used for both the inner and outer problems, i.e. $\mathcal{B}_{in} = \mathcal{B}_{out}$. Therefore, we only need to evaluate the function $a$ on a finite set of points $x$ where $(x, y) \in \mathcal{B}_{in}$. Without restricting the solution set of $a$ or adding regularization to $\hat{L}_{adj}$, the optimal solution $a_\omega^\star$ simply matches $-\partial_v f(\hat{h}_\omega(x), y)$ on the set of points $x$ s.t. $(x, y) \in \mathcal{B}_{in}$. Our implementation directly exploits this observation and uses the following expression for the total

gradient estimation:

$$g_{out} = - \sum_{(x,y) \in \mathcal{B}_{in}} \partial_{\omega,v} f(\hat{h}_{\omega}(x), r_{\omega}(x), s_{\omega}(x)) \partial_v f(\hat{h}_{\omega}(x), y). \tag{64}$$

## H.2 Experimental details

As in the experiments of Nikishin et al. [2022], we use the *CartPole* environment with 2 actions, 4-dimensional continuous state space, and optimal returns of 500. For evaluation, we use a separate copy of the environment. The reported return is an average of 10 runs with different seeds.

**Networks.** We us the same neural network architectures that are used in the *CartPole* experiment of Nikishin et al. [2022, Appendix D]. All networks have two hidden layers and *ReLU* activations. Both hidden layers in all networks have dimension 32. In the misspecified setting with the limited model class capacity, we set the hidden layer dimension to 3 for the dynamics and reward networks.

**Hyper-parameters.** We perform 200000 environment steps (outer-level steps) and set the number of inner-level iterations to $M = 1$ for both OMD and *FuncID*. for MLE, we perform a single update to the state-value function for each update to the model. For training, we use a replay buffer with a batch size of 256, and set the discount factor $\gamma$ to 0.99. When sampling actions, we use a temperature parameter $\alpha = 0.01$ as in Nikishin et al. [2022]. The learning rate for outer parameters $\omega$ is set to $10^{-3}$. For the learning rate of the inner neural network and the moving average coefficient $\tau$, we perform a grid search over $\{10^{-4}, 10^{-3}, 3 \cdot 10^{-3}\}$ and $\{5 \cdot 10^{-3}, 10^{-2}\}$ as in Nikishin et al. [2022].

## H.3 Additional results

**Time comparison.** Figure 7 shows the average reward on the evaluation environment as a function of training time in seconds. We observe that our model is the fastest to reach best performance both in the well-specified and misspecified settings.

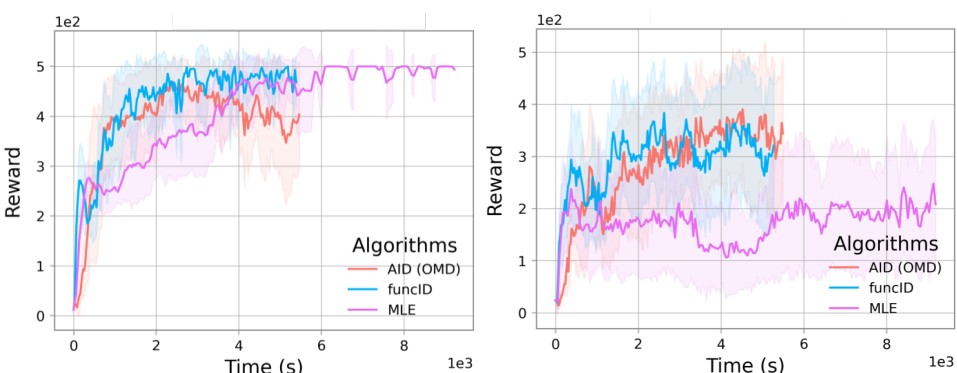

Figure 7: Average Reward on an evaluation environment vs. time in seconds on the *CartPole* task. (**Left**) Well-specified predictive model with 32 hidden units to capture the variability in the states dynamics. (**Right**) misspecified predictive model with only 3 hidden states.

**MDP model comparison.** Figure 8 shows the average prediction error of different methods during training. The differences in average prediction error between the bilevel approaches (OMD, *FuncID*) and MLE reflect their distinct optimization objectives and trade-offs. OMD and *FuncID* focus on maximizing performance in the task environment, while MLE emphasizes accurate representation of all aspects of the environment, which can lead to smaller prediction errors but may not necessarily correlate with superior evaluation performance. We also observe that *FuncID* has a stable prediction error in both settings meanwhile OMD and MLE exhibit some instability.

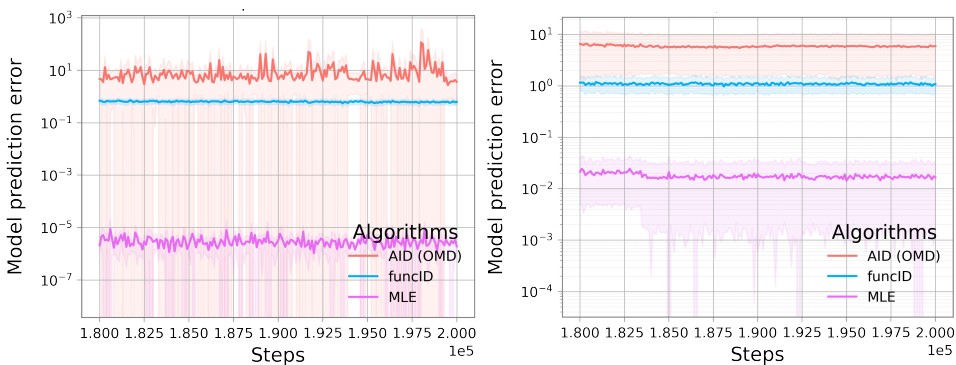

Figure 8: Average MDP model prediction error in the training environment vs. inner optimization steps on the *CartPole* task. (**Left**) Well-specified predictive model with 32 hidden units to capture the variability in the states dynamics. (**Right**) misspecified predictive model with only 3 hidden states.

