# OpenReview forum: "Functional Bilevel Optimization for Machine Learning"
_NeurIPS.cc/2024/Conference — NeurIPS 2024 spotlight_

### Official Review · Reviewer_QfTd · 2024-07-06

**Soundness:** 3
**Presentation:** 3
**Contribution:** 3
**Rating:** 7
**Confidence:** 3

**Summary:**

The authors propose a functional view of bilevel optimization in machine learning, in which the inner objective is often strongly convex for many problems of interest. The authors prove the resulting optimization algorithm's convergence and benchmark their approach for regression and model-based RL tasks.

**Strengths:**

I'm borderline on this submission as the authors have invested substantial effort, but I'm not fully convinced. I am interested to see what other reviewers think.

1. The authors address an important problem with a range of applications. I'd additionally suggest adversarial robustness / adversarial games in general as another important bilevel optimization problem area.
2. The authors have a good balance of theory and experiments, with particularly extensive theoretical contributions (although I haven't had the bandwidth to check the proofs).
3. The writing and logical flow of the paper are solid, and the authors do a good job of making complicated theory comprehensible.

**Weaknesses:**

1. The authors seem to do a "bait-and-switch" when going from function space to parameter space. Namely, the authors repeatedly emphasize that in function space, the inner objective is generally strongly convex. However, as the authors point out, this is no longer true when moving to function parameterizations -- and all the experiments concern concrete parameterizations. I'm fairly lost here as to how the authors theoretically handle this jump, as all the theory seems to assume that we are operating in function space.
2. The assumptions in Theorem 3.1 seem very strong. Namely, it is assumed that the inner and adjoint problems are solved close to optimality; however, the inner optimization is nonconvex in the parameters (as discussed above).
3. The authors repeatedly scatter in comparisons to AID throughout the text. I think it would be good to have these all summarized in one table in the appendix (runtime, convergence guarantees, assumptions, etc.)
4. The experimental results are quite weak. Namely, buried in H.3 the authors experiment with boosting the training sample size by a factor of two, and in this setting FuncID seems to underperform DFIV (although FuncID linear is now better). It seems that the improvement over the state of the art is not very robust.
5. FuncID seems to require more outer iterations to converge than baselines (Figure 1b), which I don't see being discussed by the authors.

**Questions:**

1. Can you elaborate on $\epsilon_{in}$ and $\epsilon_{adjoint}$ in Theorem 3.1? I'm not sure what the expectation is taken over in the definition of line 1100.
2. Why is the prediction error of FuncID worse than MLE in Figure 7b?

Notes:
1. In (FBO), there's a space between the colon and the equals (use \coloneqq).
2. In line 94, link to where the assumption is discussed.
3. Font size in the figures is very small and hard to read.

**Limitations:**

I'd like to see an explicit limitations section. The authors' response to the limitations question in the checklist is that they state theorem assumptions -- this is not a comprehensive discussion of limitations, which should include practical experimental considerations as well.

---

> ### Author Rebuttal · Authors · 2024-08-07
>
> Thank you for your detailed feedback. We will correct the typos, increase the font size in the figure captions, and add a link to the discussion of the assumptions as suggested. Additionally, we will include a discussion section in the main paper (provided in the general response) to address limitations and perspectives. We address other points below.
>
> ### From functional to parametric
>
> Our approach addresses one of the two main challenges below (C1 and C2) in using deep networks for bilevel optimization. We outline these challenges and show how our method overcomes C2 using strong convexity in function space. These clarifications will be added to the text.
>
> **A. Challenges of bilevel optimization with deep networks**
>
> - **C1- Non-convexity of the lower-level optimization problem** is unavoidable with deep networks. However, several studies show this non-convexity to be 'benign' and ensure global convergence when optimising deep networks using gradient methods in the overparameterized regime (see general comments).
> - **C2- Ambiguity of the bilevel problem (Arbel & Mairal 2022)**. Exact solutions at the lower level can be multiple due to over-parameterization. Thus, no 'implicit function' links the inner-level solution to the outer-level parameter, making it impossible to use the implicit function theorem to compute the total gradient.
>
>
> **B. Algorithm derivation**. From a functional bilevel problem, our approach is to "Differentiate implicitly first, then parameterize," leading to funcID. The alternative, "Parameterize first, then differentiate implicitly," results in AID, as described below.
>
> - **Differentiate implicitly first, then parameterize (FuncID)**. Functional strong convexity is used to apply the implicit function theorem in function space and derive the implicit gradient. This gradient is then estimated by approximating the lower-level solution and adjoint function using neural networks. While optimising these networks involves a non-convex problem (Challenge C1), the approach avoids Challenge C2 since the neural networks merely estimate quantities in the well-defined implicit gradient expression.
>
> - **Parametrize first, then differentiate implicitly (AID)**. The inner-level function is constrained to be a NN, converting the problem into a non-convex ‘parametric’ bilevel problem where the lower-level variables are the NN parameters. Computing the implicit gradient requires viewing the optimal network parameter as an implicit function of the outer parameter. However, this implicit function does not exist due to multiple global solutions to the inner problem for a given upper variable $\omega$. Thus, this approach faces both challenges C1 and C2.
>
> ### Other comments
>
> **Optimality assumption in Thm 3.1**. The assumptions in Thm 3.1 align with recent findings that non-convexity is 'benign' when optimizing over-parametrized NN ensuring that gradient methods linearly converge to global solutions. Thus, *in such a regime*, one can reduce the errors $\varepsilon_{\text{in}}$​ and $\varepsilon_{\text{adj}}$ by optimising the inner and adjoint functions using gradient descent. We will clarify these points (see proposed limitation section).
>
> **Comparison table between AID and FuncID**. For more clarity, we propose to merge section D and F of the appendix which both compare AID and FuncID from different aspects.
> We will also summarize the comparisons already present throughout the paper between AID and *FuncID* into a single table that will be included in the paper.
>
> **Experimental results**:
>
> *FuncID vs. DFIV for instrumental regression*: to reach a more rigorous conclusion, we have performed statistical tests. For the 5K dataset, funcID outperforms DFIV (p-value=0.003, one-sided paired t-test), but for the 10K dataset in H.5, the difference between both approaches was not statistically significant. Overall, FuncID performs in the same ballpark as the state-of-the-art approach DFIV, which is specifically designed for the IV problem. We will add these observations in the discussion of the results.
>
> *Robust/consistent improvement over AID/ITD*: all results, including those in Appendix H.3, show that FuncID outperforms commonly used bilevel optimization algorithms in ML (namely AID and ITD). These results were obtained by fairly allocating a budget for selecting the best hyperparameters for AID and ITD.
>
> *Additional comparisons*: we have compared our method with a recent approach that handles non-convexity by considering an optimistic version of the bilevel problem and turning it to a penalised single-level problem (see general response for more details).
>
> **Number of outer iterations**. Convergence in Fig 1 can be assessed by monitoring both inner-level and outer-level losses. Outer-level loss alone does not indeed indicate convergence unless the outer loss is evaluated at the ‘exact’ inner-level solution, which is generally inaccessible before convergence. We agree that this is a source of confusion. The part of Fig 1 from which it is easier to draw conclusion is the out-of-sample MSE, which reflects generalization (see comment above).
>
> **Q 1**. $\mathbb{E}$ denotes the expectation with respect to the random data samples. The quantities $\epsilon_{in}$ and $\epsilon_{adj}$ represent a bound on the optimality error of the approximate solutions $\hat{h}$ and $\hat{a}$.  For instance, $\mathbb{E}[L_{in}(w,\hat{h_w})- L_{in}(w,h_w^{\star})]  \leq \epsilon_{in}$. The expectation accounts for the fact that these approximations are based on random samples. We agree that this deserves some clarifications.
>
> **Q 2**. It is expected that MLE has a smaller prediction error because it explicitly minimizes the prediction error in its objective. In contrast, the bilevel formulations (FuncID and AID) learn an MDP whose state-value function aligns with the true MDP.

---

> > ### Comment · Reviewer_QfTd · 2024-08-09
> >
> > Thank you to the authors for their clarification. The comparison in points A and B above is really important, and it gets lost in the technical details of the paper. I strongly suggest featuring this prominently in the main paper body, and perhaps cutting from section 3.
> >
> > I've raised my score.

---

### Official Review · Reviewer_98ub · 2024-07-13

**Soundness:** 3
**Presentation:** 3
**Contribution:** 3
**Rating:** 6
**Confidence:** 3

**Summary:**

The paper proposes a functional approach to bilevel optimization for machine learning, focusing on inner-level problems defined over function spaces rather than traditional parametric settings. This allows the application of the proposed method to machine learning tasks without requiring the strong convexity typically assumed in bilevel optimization.

**Strengths:**

- The paper introduces a functional perspective to bilevel optimization, extending its applicability to settings where traditional assumptions (like strong convexity) do not hold.

**Weaknesses:**

- The paper does not sufficiently compare the proposed methods against a broad spectrum of existing algorithms, particularly the latest advancements in the field of bilevel optimization problem. This lack of comprehensive benchmarking restricts the ability to fully evaluate the performance enhancements or potential drawbacks of the proposed methods relative to the state-of-the-art.

- The paper lacks a thorough analysis of how the proposed methods perform across varied settings and parameter configurations. It is not clear if there are specific scenarios where the methods might underperform or fail to converge. Additional details on the robustness of the methods in diverse operational environments would be beneficial.

- The assumptions necessary for the theoretical framework might not be easily verifiable in practical scenarios, potentially limiting the method's applicability.

**Questions:**

1. Can the proposed functional framework be generalized to other types of bilevel problems that involve constraints on the lower-level problem? How would you address potential non-smoothness or non-convexity in these extended settings?

2. Please explain how does the choice of function space impact the stability and convergence of the FuncID method?

**Limitations:**

Authors addressed their work limitation.

---

> ### Author Rebuttal · Authors · 2024-08-07
>
> **Additional comparisons**. As suggested by reviewer *MXrv* and in addition to the comparisons already made with most widely-used bilevel algorithms (AID, ITD, and variants) and SoTA methods for each problem (DFIV for the IV problem and MLE for model-based RL), we now additionally include a comparison with a recent approach for solving non-convex bilevel problems based on penalty methods (see general response). The new results are, on average, consistent with the previous ones and are still in favour of *FuncID* which exploits functional strong convexity. We would be happy to include additional methods if you think they would be relevant.
>
> **Relevant settings for FuncID**. We expect *FuncID* to outperform AID in settings where the bilevel problem has a ‘hidden’ strong convexity in functional space. That is simply because AID does not exploit such strong convexity, while *FuncID* does. While this setting covers many practical scenarios (such as those considered in the paper (two-stage least squares regression and model-based RL), we do not expect particular improvements in the absence of such a structure. We will make that clear in the text and limitation section.
>
> **Verifiable assumptions**. We agree that the assumptions in Prop. 2.3 might seem complex, but they are easily verifiable through standard calculations. In Proposition E.1 of the appendix, we verify that these assumptions hold for regression problems involving feedforward networks when using quadratic objectives. The verification only requires computing derivatives and upper-bounding them, and could be applied to other problems similarly. We will make sure this is clear in the text.
>
> **Extensions**. Thank you for raising these points, we discuss them in the future work section (presented in the general response). Extending the framework to a constrained inner problem or non-smooth setting should be possible, if the uniqueness of the solutions in functional space is preserved. However, this would require introducing additional tools to handle non-smoothness/constraints such as those from the recent works on non-smooth bilevel optimization [Bolte et al., 2022] and would be an interesting future work direction.
>
> **Choice of the function space**. The function space we consider is motivated by existing bilevel problems that are already formulated in such spaces (of which we consider 2 examples in the applications). Using different spaces would require a different analysis which is beyond the scope of this work but would certainly be interesting for future work as discussed in the new limitation/future work section (provided in the general response).

---

> > ### Comment · Reviewer_98ub · 2024-08-09
> >
> > Appreciate authors for their complete response and clarification. I am satisfied with their response and maintain my score.

---

### Official Review · Reviewer_MXrv · 2024-07-13

**Soundness:** 3
**Presentation:** 3
**Contribution:** 3
**Rating:** 7
**Confidence:** 4

**Summary:**

This paper introduces a novel functional perspective on bilevel optimization, where the inner objective is defined over a function space. The authors developed functional implicit differentiation and functional adjoint sensitivity, which together facilitate the establishment of a gradient-based algorithm in the functional space. They also analyze the convergence rate of the proposed algorithm and apply it to two-stage least squares regression and model-based reinforcement learning. Experimental results validate the effectiveness of the proposed method.

**Strengths:**

1. The proposed method offers a new insight into solving bilevel optimization problems with nonconvex lower-level objectives by leveraging their strong convexity in the functional space. This is particularly noteworthy because, although neural networks are nonconvex, the loss function in model training can be convex or strongly convex.
2. This paper provides a heuristic approach with both theoretical and practical impact. The proposed method not only has a convergence guarantee but is also implementable in real-world applications. The two applications chosen in this paper are also novel: the first has potential impacts on causal representation learning, while the second provides a new perspective on model-based reinforcement learning.

**Weaknesses:**

1. The convergence analysis in this paper is based on the stochastic biased gradient descent framework, making the results explicitly dependent on the sub-optimality constant $\epsilon_{in}$ and $\epsilon_{adj}$. However, it is unclear how these errors relate to the inner loop $M$ and $K$. It might be beneficial to leverage the strong convexity of the inner and adjusted objective functions to clarify these dependencies. See similar techniques used in [1]-[3].

[1] K. Ji, J. Yang, and Y. Liang. Bilevel optimization: Convergence analysis and enhanced design. ICML, 2021.

[2] T. Chen, Y. Sun, and W. Yin. Closing the Gap: Tighter Analysis of Alternating Stochastic Gradient Methods for Bilevel Problems. NeurIPS, 2021.

[3] M. Dagréou, P. Ablin, S. Vaiter, T. Moreau. A framework for bilevel optimization that enables stochastic and global variance reduction algorithms. NeurIPS, 2022.


2. Minor issues: experimental baselines. It might be better to also compare with those bilevel methods that are capable to solve nonconvex lower-level problem [4]-[5] as they can also potentially solve the two-stage least squares regression and model-based reinforcement learning problem.

[4] J. Kwon, D. Kwon, S. Wright, and R. D. Nowak. On penalty methods for nonconvex bilevel optimization and first-order stochastic approximation. ICLR 2024.

[5] H. Shen, and T. Chen. On Penalty-based Bilevel Gradient Descent Method. ICML 2023.

**Questions:**

Same as weakness.

**Limitations:**

Same as weakness.

---

> ### Author Rebuttal · Authors · 2024-08-07
>
> **Error analysis**. Thank you for pointing out these references, we will make sure to discuss them. We agree that the result of Thm. 3.1 does not provide an explicit dependence of the errors on the inner-level optimization. Providing such dependence would require introducing another level of technical complexity, beyond the techniques used in *[1]-[3]* which are tailored for the strongly convex case in a parametric setting. In our case, one would instead need to use quantitative approximation results of functions in $L_2$ spaces by NNs [Bach 2017], as well as global convergence results for NNs [Allen-Zhu et al., 2019, Liu et al., 2022]. Such analysis would require substantial effort that is best suited for a separate future work. We discuss this in the future work section (see the general response).
>
> **Additional comparison**. Thank you for suggesting these methods. We performed an additional comparison on the Instrumental Variable (IV) problem (see Fig. 1 in the pdf file). These methods handle non-convexity by considering an optimistic version of the bilevel problem and turning it into a penalized single-level problem. However, they do not exploit the functional strong convexity of the IV problem. Consequently, the new results, on average, still favor *FuncID*, which exploits functional strong convexity as shown in the general response.
>
> [Bach 2017] Bach, F. Breaking the curse of dimensionality with convex neural networks. Journal of Machine Learning Research, 18(19), 1-53 2017.
>
> [Allen-Zhu et al. 2019] Zeyuan Allen-Zhu. Yuanzhi Li. Zhao Song. A Convergence Theory for Deep Learning via Over-Parameterization. ICML 2019.
>
> [Liu et al. 2022] Chaoyue Liu, Libin Zhu, Mikhail Belkin. Loss landscapes and optimization in over-parameterized non-linear systems and neural networks. Applied and Computational Harmonic Analysis. 2022

---

> > ### Comment · Area_Chair_TR5g · 2024-08-12
> > **Reviewer response?**
> >
> > Reviewer MXrv, could you please review the authors' response and see whether it addresses your questions? Please acknowledge having done so in a comment. Thanks.

---

> > ### Comment · Reviewer_MXrv · 2024-08-12
> >
> > I thank the authors for their detailed responses. It solves all of my concerns so that I will raise my score.

---

### Official Review · Reviewer_p8LC · 2024-07-15

**Soundness:** 4
**Presentation:** 3
**Contribution:** 4
**Rating:** 7
**Confidence:** 3

**Summary:**

This paper offers a new functional point of view for bilevel optimization problems in machine learning. This functional approach allows the use of an overparameterized neural network as inner prediction function while previous works have used an inner objective that is strongly convex with respect to the parameters of the prediction function. For the inner problem, the prediction function is a function that lies in a Hilbert space of square integrable functions ($L^2$). The authors develop the theory of Functional Implicit Differentiation, which is a flexible class of algorithms to do functional bilevel optimization over $L^2$ spaces. First, they show that strong convexity assumption with respect to the prediction function as opposed to the model parameters ensures the existence and uniqueness of the optimal prediction function $h_\omega^\ast$ for the inner-level objective. Second, they show that differentiability assumptions on the inner-level objective and its Fr\'echet derivative with respect to prediction function $h$ ensure the differentiability of the map $\omega \rightarrow h_\omega^\ast$. Given further assumption about joint differentiability of outer objective, they show that it is possible to compute total objective $\mathcal{F}$ using the adjoint function that minimizes a quadratic objective over a Hilbert space. Finally, assuming that the inner objective and outer objective are defined over the distributions and that a batch of data samples from each distribution can be sampled, an iterative algorithm is proposed that has the following three steps: (1) approximation of inner objective, outer objective and quadratic objective for adjoint function using the batch samples, (2) do a gradient-based update for the parameters of the prediction function and adjoint function assuming they have a fixed parametric form and (3) total gradient approximation and gradient-based update of the functional parameter $\omega$ obtained by solving the outer objective. They show that these class of algorithms converge to a stationary point at $\mathcal{O}(1/N)$ rate. Experiments are performed using Two-stage least squares regression and Model-based reinforcement learning as use cases.

**Strengths:**

The paper makes a substantial contribution by developing the theory of functional bilevel optimization with less restrictive assumptions of strong convexity with respect to the prediction function as opposed to model parameters. This is useful because it allows the prediction function $h_\omega$ to be modeled by deep neural network, which has a non-convex training objective with respect to the model parameters. This paper could lead to more research into optimization over function spaces as strong convexity over model parameters is a restrictive assumption in practice.

**Weaknesses:**

The paper is very technical and requires a good understanding of monotone operator theory and theory of Fr\'echet and Hadamard differentiability to understand it fully. Still, the technical details about the existence and uniqueness of the prediction function and the map $\omega \rightarrow h_\omega^\ast$ are deferred to the appendix for interested readers. I would recommend the authors simplify the notation in the main paper a bit to reduce the clutter. For instance, $h_\omega^\ast(x)$ can be represented simply as $h_\omega$. It may help to have a table that shows the functional arguments (e.g. $h_\omega$, $a_\omega$, etc.), functional parameters $\theta$, $\xi$ and $\omega$ and then function variables $x$ and $y$.

**Questions:**

1. On Line 126, there is a typo: "elemen" should be replaced by "element".
2. On Line 629, capitalize the word "euclidean".
3. What purpose does the variable $y$ and space $\mathcal{Y}$ serve in general? Is it a variable or a parameter? Your prediction function $h_\omega$ is a function of $x$. In the case of 2SLS, you have $y = (t, o)$ and $x$ is the instrumental variable. The function you are interested in is $f_\omega(t)$ and is a function of $t$, which is a subvariable of $y$ and $h_\omega(x) = \mathbb{E}[f_\omega(t) | x]$. Maybe it will help to clarify what $y$ and $\mathcal{Y}$ are in the theoretical section.
4. What is the rationale for the label "FuncID linear" in Figure 1? Why do you refer to it as "linear"?
5. Why do you feel "FuncID linear" converge faster in terms of outer iterations than "FuncID" in Figure 1?

**Limitations:**

The authors didn't go into a discussion about the limitations nor did they provide a conclusion section and simply concluded with a results section.

---

> ### Author Rebuttal · Authors · 2024-08-07
>
> Thank you for thoroughly reading our work and giving us helpful feedback. Taking your feedback into account, we will simplify the notation and include a notation table. We agree that this could help the reader get a quick grasp of the mathematical objects considered. We will also include a discussion section (presented in the general response) with a paragraph on the limitations of our work. Below, we address your comments in detail.
>
> **Typos**. We will fix the typos in the final version.
>
> **Purpose of $y$ and $Y$**. Data is represented by pairs $(x,y)$, noting that the function $h$ takes only $x$ as input. Thus, $x$ and $y$ are both random variables. In the simplest supervised learning setup (see *Eq. 1* for instance), $y$ is a label living in a space $Y$ and $h(x)$ tries to predict $y$. The theory of Section 3 is however more general, and $y$ can serve other purposes. For instance, in 2SLS, $y$ is made of two variables $t$ and $o$, whose causal relationship is described in Fig. 4 (App. H). We admit that the setup of 2SLS Instrumental Variable regression is a bit particular and can be confusing. We will clarify these points in the final version of the paper.
>
> **Difference between *FuncID* and *FuncID linear***. *FuncID linear* uses a linear model to approximate the adjoint function, while *FuncID* uses a trainable neural network. The linear model is obtained by learning a linear combination of the frozen penultimate layer of the current prediction function $h$ (modelled as a neural network). We will clarify this in the text.
>
> **Why *FuncID linear* converges faster**. Following the previous explanation, *FuncID* optimizes all adjoint network parameters, while *FuncID linear* learns only the last layer in closed form. "FuncID linear" converges faster because solving a linear system is computationally faster than iteratively approximating the full adjoint function using a neural network. However, *FuncID linear* is less expressive for approximating the adjoint, which may result in suboptimal solutions that could explain the performance gap in terms of test error.

---

> ### Comment · Reviewer_p8LC · 2024-08-12
>
> Thank you for your responses to my questions and for the clarifications! Good to hear that the typos will be fixed in the final draft and that the missing details will be added to the final draft! I am also glad that you are adding a detailed discussion/conclusion at the end as it is important to leave the reader with some take-away points from your paper. Good luck!

---

### Author Rebuttal · Authors · 2024-08-07

# General comments

We thank the reviewers for their useful feedback. We now list the main changes made to the paper.

## Discussion section (limitations and perspectives).

We agree that such a section is important, and propose to include the following discussion:

### Discussion and concluding remarks
This paper introduces a functional paradigm for bilevel optimization in machine learning, shifting the optimization focus from the parameter space to the function space. This new approach addresses the limitations of traditional bilevel optimization methods when dealing with over-parameterized neural networks. Specifically, the proposed method exploits the functional strong convexity of certain bilevel problems to derive an abstract, yet approximable, expression for the implicit gradient that requires solving both an inner and an adjoint optimization problem in functional space. Approximation is achieved by restricting both problems to a flexible class of functions, such as neural networks. The paper establishes the validity of this approach by developing a theory of functional implicit differentiation and providing a general convergence result for the proposed method. Despite these contributions, we discuss several limitations of our work and highlight potential research directions.

**Hyperparameter selection**. One notable limitation is the presence of multiple hyperparameters in the proposed algorithms. This is a common challenge shared by all bilevel optimization methods, complicating the practical implementation and tuning of these algorithms. Selecting and optimizing these hyperparameters can be time-consuming and may require extensive experimentation to achieve optimal performance.

**Convergence guarantees:** The result in Theorem 3.1 relies on the assumption that both the inner and adjoint optimization problems are solved up to some optimality errors. This assumption is motivated by recent global convergence results for over-parameterized networks [Allen-Zhu et al., 2019, Liu et al., 2022]. Although over-parameterized networks are ubiquitous in the machine learning literature, it is unclear to what extent this optimality assumption remains realistic beyond such settings. Moreover, the result in Theorem 3.1 does not explicitly relate these optimality errors to the optimization procedure used for the inner and adjoint problems. A precise quantification of these errors would be valuable to strengthen the theoretical foundations of the proposed methods and provide principled guidelines for the choice of hyperparameters. These theoretical considerations do not prevent from applying the method even beyond the setting where the convergence results hold, much like with popular bilevel algorithms such as AID or ITD.

**Choice of the function space**. Another important consideration is the choice of the function space in the functional bilevel optimization framework. While we primarily focus on $L_2$ spaces, there are numerous other function spaces that could be explored, such as Reproducing Kernel Hilbert Spaces and Sobolev spaces. Investigating these alternative spaces may reveal additional advantages and open the way for a broader class of machine learning applications where higher-order derivatives of a prediction function appear naturally in the objectives.

**Non-smooth/constrained setting**. The proposed method primarily focuses on smooth and unconstrained problems, but many practical machine learning applications involve non-smooth objectives or constraints. Extending the proposed framework to handle these scenarios would significantly broaden its applicability. Notably, the works [Bolte et al.] on non-smooth implicit differentiation could perhaps be leveraged to adapt our methods to non-smooth settings. Future work should explore these opportunities to further enhance the flexibility and applicability of the functional bilevel optimization approach.

## Additional experiment (see attached PDF).
As suggested by reviewer *MXrv*, we have compared to a bilevel penalty-based method for the Instrumental Variable application. We use two variants and perform an extensive grid search to adjust their hyper-parameters:

1.  **Gradient penalty** (Eq. 5.1 in [5] (Shen et. al. 2023)): we perform a grid search with the following hyper-parameters: learning rate [0.01, 0.001, 1e-4, 1e-5, 1e-6]; weight decay [0., 0.1, 0.01, 0.001]; penalty constant [1, 0.1, 0.01, 0.001]. Since the method has only a single optimization loop, we increase the number of total iterations to 2000 compared to the other methods (100 outer-iterations). The rest of the parameters are the same for all methods.

2. **Value function penalty** (Eq. 3.2a in [5]). We use the same grid search and test number of inner steps [10,20]. Since the method has a double optimization  loop, we then use 100 outer iterations. The rest of the parameters are the same for all methods.

This method handles non-convexity by considering an optimistic version of the bilevel problem and turning it into a penalised single-level problem. *funcID* performs significantly better than the gradient penalty, whereas it is in the same ballpark as the value function penalty method or better (lower mean on 5k/10k, but higher median for 5k). Notably, the value function penalty seems to have a high variance, with some particularly bad outliers, despite the extensive grid search for tuning hyper-parameters. These conclusions will be added to the paper.

[Allen-Zhu et al. 2019] Zeyuan Allen-Zhu. Yuanzhi Li. Zhao Song. A Convergence Theory for Deep Learning via Over-Parameterization. ICML 2019.

[Liu et al. 2022] Chaoyue Liu, Libin Zhu, Mikhail Belkin. Loss landscapes and optimization in over-parameterized non-linear systems and neural networks. Applied and Computational Harmonic Analysis. 2022

[Bolte et al., 2022]  Jérôme Bolte, Edouard Pauwels, Samuel Vaiter.  Automatic differentiation of nonsmooth iterative algorithms. Neurips. 2022.

---

### Comment · Area_Chair_TR5g · 2024-08-08
**Discussion period**

Dear Reviewers,

Thanks for the contributions in the review period and looking forward to a fruitful discussion on Submission 6898 in the next few days.

Kind regards,

Your AC

---

### Author Response · Authors · 2024-08-14
**Discussion period authors' comment**

Thank you to all reviewers for responding to our rebuttal, and to the area chair for encouraging the discussion on our submission. We will ensure that the changes mentioned in the rebuttal and confirmed by the reviewers are integrated into the final version of the paper, if it is accepted.

---

### Decision · Program_Chairs · 2024-09-25

**Decision:**

Accept (spotlight)

**Comment:**

This paper develops a new functional point of view for bilevel optimization problems in machine learning that provides a new insight into solving bilevel optimization problems with nonconvex lower-level objectives by leveraging their strong convexity in the functional space. This development is noteworthy because, although neural networks are nonconvex, the loss function in model training can be convex or strongly convex. The authors have developed a complete paper with theory, practical insights, and plenty of computational tests. Bilevel optimization is a tough topic, but the authors have also made the paper notably accessible.